# Distinctive molecular features of regenerative stem cells in the damaged male germline

Hue M. La [1,2,3], Jinyue Liao[4,5], Julien M. D. Legrand [1,2,3], Fernando J. Rossello[3,6], Ai-Leen Chan[1,2,3], Vijesh Vaghjiani[2,7], Jason E. Cain [2,7], Antonella Papa [8], Tin Lap Lee [4✉] & Robin M. Hobbs [1,2,3✉]

Maintenance of male fertility requires spermatogonial stem cells (SSCs) that self-renew and generate differentiating germ cells for production of spermatozoa. Germline cells are sensitive to genotoxic drugs and patients receiving chemotherapy can become infertile. SSCs surviving treatment mediate germline recovery but pathways driving SSC regenerative responses remain poorly understood. Using models of chemotherapy-induced germline damage and recovery, here we identify unique molecular features of regenerative SSCs and characterise changes in composition of the undifferentiated spermatogonial pool during germline recovery by single-cell analysis. Increased mitotic activity of SSCs mediating regeneration is accompanied by alterations in growth factor signalling including PI3K/AKT and mTORC1 pathways. While sustained mTORC1 signalling is detrimental for SSC maintenance, transient mTORC1 activation is critical for the regenerative response. Concerted inhibition of growth factor signalling disrupts core features of the regenerative state and limits germline recovery. We also demonstrate that the FOXM1 transcription factor is a target of growth factor signalling in undifferentiated spermatogonia and provide evidence for a role in regeneration. Our data confirm dynamic changes in SSC functional properties following damage and support an essential role for microenvironmental growth factors in promoting a regenerative state.

[1] Centre for Reproductive Health, Hudson Institute of Medical Research, Melbourne, VIC 3168, Australia. [2] Department of Molecular and Translational Sciences, Monash University, Melbourne, VIC 3800, Australia. [3] Australian Regenerative Medicine Institute, Monash University, Melbourne, VIC 3800, Australia. [4] Developmental and Regenerative Biology Program, School of Biomedical Sciences, Faculty of Medicine, The Chinese University of Hong Kong, Shatin, Hong Kong SAR, China. [5] Department of Chemical Pathology, Faculty of Medicine, The Chinese University of Hong Kong, Shatin, New Territories, Hong Kong SAR, China. [6] University of Melbourne Centre for Cancer Research, University of Melbourne, Melbourne, VIC 3000, Australia. [7] Centre for Cancer Research, Hudson Institute of Medical Research, Melbourne, VIC 3168, Australia. [8] Cancer Program, Monash Biomedicine Discovery Institute and Department of Biochemistry and Molecular Biology, Monash University, Melbourne, VIC 3800, Australia. ✉email: leetl@cuhk.edu.hk; robin.hobbs@monash.edu

Male fertility is sustained by spermatogonial stem cells (SSCs) within the testis that self-renew and produce differentiating germ cells for spermatogenesis[1,2]. Germ cells are sensitive to chemotherapeutic agents and radiation, placing cancer patients at risk of treatment-induced infertility. Therapies that primarily affect cycling differentiating spermatogonia cause temporary infertility while those targeting slower-cycling SSCs, e.g., alkylating agents, can cause permanent infertility[3]. Although therapy-resistant SSCs may restore fertility, outcomes are variable due to differences in treatment regimens and unknown factors underlying a patient's sensitivity to therapy and susceptibility to SSC loss[3]. Given that pathways regulating regenerative responses in SSCs are poorly characterised, the prediction, prevention, and treatment of infertility for these patients are not possible. Although assisted reproductive technologies such as sperm banking can allow post-pubertal men to have children following treatment, these options are not available to prepubertal boys[4].

SSCs in adult mice are contained within a population of Type A undifferentiated spermatogonia ($A_{undiff}$) (Fig. 1a)[1,2]. The $A_{undiff}$ pool consists of singly isolated cells ($A_s$), pairs of interconnected cells ($A_{pr}$) and chains of 2n cells ($A_{al}$) formed because of incomplete cytokinesis. $A_{undiff}$ are functionally and molecularly heterogenous and a fraction act as SSCs in homoeostatic tissue, while the bulk act as transit-amplifying progenitors. $A_{undiff}$ positive for cell surface receptor GFRα1 (a majority of $A_s$ and $A_{pr}$) represent an SSC-enriched population, whereas $A_{undiff}$ expressing Ngn3 (Neurog3) or Rarg (predominantly $A_{al}$) are differentiation-destined progenitors (Fig. 1a)[1]. SSCs are also marked by Id4 expression and progenitors by Oct4 (Pou5f1) while transcription factors PDX1 and EOMES mark a primitive GFRα1 + fraction[1,5]. In response to retinoic acid, progenitors initiate differentiation and undergo a series of mitotic divisions before generating meiotic spermatocytes and ultimately spermatids. Markers such as PLZF, SALL4 and E-Cadherin are expressed throughout the $A_{undiff}$ population and at early differentiation stages while c-KIT is induced upon differentiation (Fig. 1a)[1].

Spermatogenesis recovery following germ cell depletion is dependent on surviving SSCs[6,7]. Effects of the alkylating agent busulfan (BU) have been studied in rodents where it causes apoptosis of $A_{undiff}$ and differentiating spermatogonia. Few spermatogonia remain after the highest non-lethal BU dose (~40 mg/kg) resulting in infertility[8]. However, lower BU doses (e.g. 10 mg/kg) induce spermatogonial depletion while sparing some $A_{undiff}$ to restore the germline, providing a model of regeneration[8,9]. While molecular mechanisms underpinning male germline regeneration are poorly appreciated, morphological studies of testis seminiferous tubules have defined kinetics of $A_{undiff}$ recovery following BU treatment[9,10]. Spermatogonia are substantially depleted by day (D) 2 to 4 after BU and the lowest density of $A_{undiff}$ is observed between D6 and 8 (Fig. 1b)[9]. Transplantable SSCs are depleted by D3 post-BU[6,7]. After low-dose BU, regeneration is initiated by D10 and remaining $A_{undiff}$ actively proliferate between D10 and D15, resulting in 'overshoot' of $A_{undiff}$ populations between D16 and D18 accompanied by delayed differentiation commitment (Fig. 1b)[9].

Identification of markers associated with $A_{undiff}$ subsets has provided insight into the contribution of distinct populations during regeneration. PAX7+ and EOMES + spermatogonia, which function as homoeostatic SSCs, are resistant to BU and involved in germline regeneration[11,12]. $A_{undiff}$ marked by NGN3 or MIWI2 typically act as differentiation-destined progenitors in undisturbed tissue but following germ cell depletion contribute to regeneration, albeit to a lesser extent than PAX7 + or EOMES + cells[11–14]. Therefore, regeneration is driven not only by homoeostatic stem cells but by differentiation-primed cells

that revert to a stem cell fate upon tissue damage[13,14]. While these studies have contributed to our understanding of populations mediating regeneration, cellular pathways involved in the $A_{undiff}$ regenerative response have not been well-characterised. Furthermore, effects of germline depletion and induction of regeneration on $A_{undiff}$ heterogeneity and dynamics are unclear.

Growth factors produced within the seminiferous tubule microenvironment play key roles in SSC function[15]. Glial cell-derived neurotrophic factor (GDNF) is essential for SSC self-renewal and is produced by Sertoli and other somatic cell types within the testis[15]. GDNF binds the GFRα1/c-RET receptor present on $A_{undiff}$ subsets and activates downstream pathways including phosphoinositide 3-kinase (PI3K)/AKT and ERK MAPK to promote self-renewal[1]. Germ and somatic cells produce basic fibroblast growth factor (bFGF), which promotes spermatogonial self-renewal and proliferation[16]. Studies support a role for ERK MAPK downstream bFGF and GDNF in $A_{undiff}$ self-renewal[16,17]. However, while GDNF and bFGF are both self-renewal factors and synergistically promote expansion of cultured $A_{undiff}$, they have distinct effects on $A_{undiff}$ function[1,18]. FGF5 produced by lymphatic endothelial cells also promotes self-renewal and proliferative activity of GFRα1+ spermatogonia and acts as a limiting factor to control GFRα1+ cell density[19].

Tissue regeneration involves remodelling of the niche microenvironment that influences stem cell behaviour[20]. Gdnf expression in the testis increases after BU treatment, and levels peak during initiation of regeneration, suggesting a role in germline recovery[6,21]. Oscillatory changes in density of GFRα1 spermatogonia in BU-treated animals during recovery can be observed and result from competition for niche factors including FGF5[19]. These studies indicate that the niche is distinct during testis regeneration, and GDNF and FGFs play roles in SSC-driven regeneration.

Appropriate activation of signalling pathways in response to growth factors is essential for SSC maintenance. For instance, loss of PTEN, a negative regulator of AKT signalling, drives SSC exhaustion[22]. FOXO transcription factors are required for SSC maintenance and inhibited by AKT[22]. Further, aberrant activation of mTORC1, a growth-regulatory pathway downstream AKT and ERK MAPK, results in SSC exhaustion[23,24]. Physiological mTORC1 activation is also required for $A_{undiff}$ differentiation commitment[25–27]. Despite the role played by growth factor signalling in homoeostatic SSC function, involvement of these pathways in regenerative responses remains unstudied.

In this study, we characterise $A_{undiff}$ of homoeostatic tissue and from mice treated with BU to induce germline depletion and regeneration. We uncover a switch in gene expression of regenerative $A_{undiff}$ towards an SSC state and characterise regenerative $A_{undiff}$ markers through single cell analysis. We observe increased growth factor signalling in regenerative $A_{undiff}$ and confirm an essential role for mTORC1 in the regenerative response. Further, we identify the cell cycle regulator FOXM1 as a target of growth factor signalling in $A_{undiff}$ and provide evidence for a role of this transcription factor in regeneration. These findings increase our understanding of germline regeneration and suggest therapeutic approaches for stimulating regenerative $A_{undiff}$ activity and restoring male fertility following cancer treatment.

## Results

**Molecular features of the regenerative $A_{undiff}$ population.** To study stem cell-driven regeneration in the male germline, adult mice were treated with a low dose of BU (10 mg/kg). The $A_{undiff}$ surviving BU, here termed regenerative $A_{undiff}$, initiate a regenerative response by D8-10 post-treatment (Fig. 1b)[28,29]. Whole-mount immunofluorescence (IF) analysis of seminiferous tubules

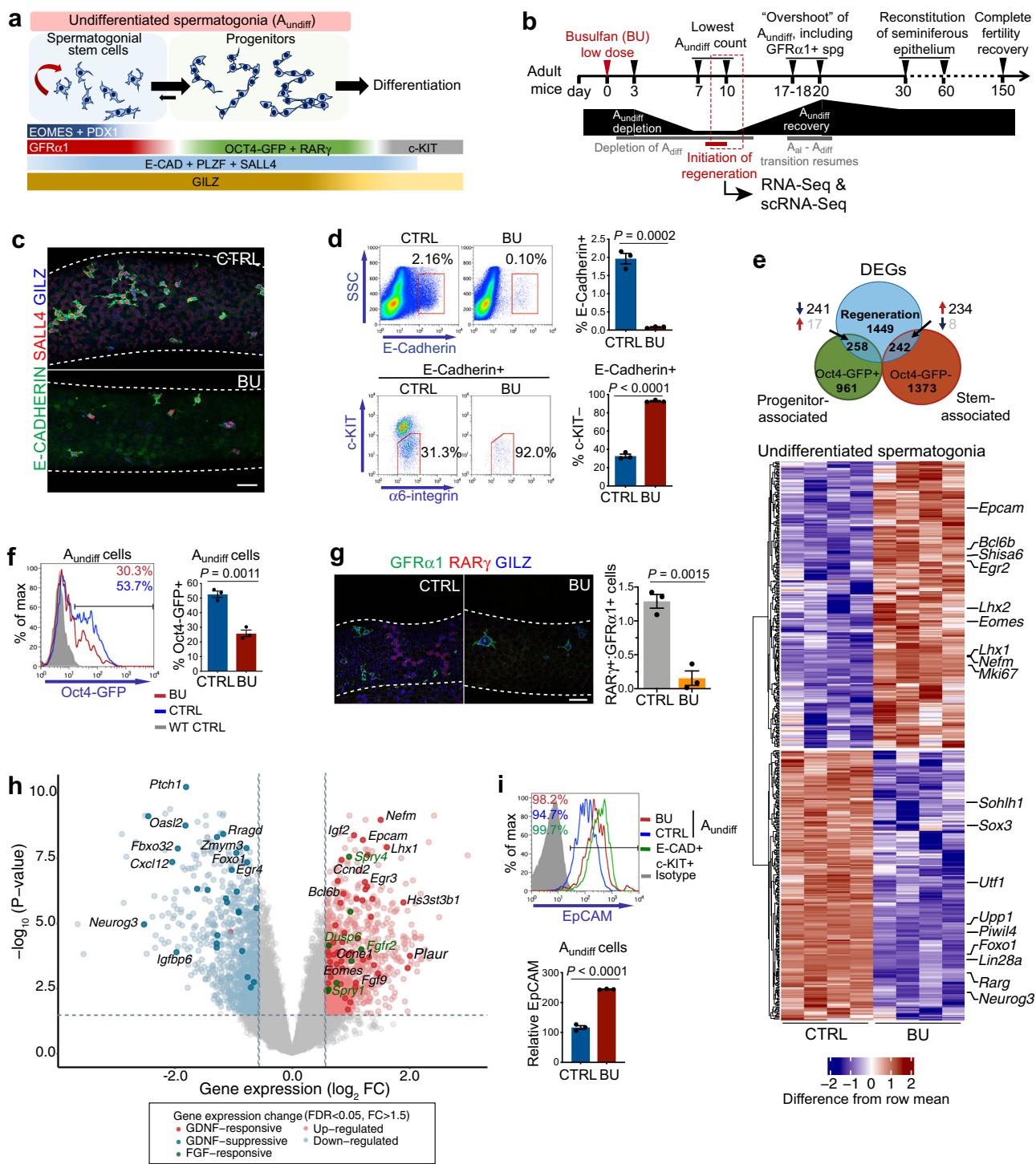

from wild-type control (CTRL) and BU-treated mice at D10 confirmed depletion of spermatogonia and persistence of low numbers of E-Cadherin+ SALL4 + $A_{undiff}$ (primarily $A_s$ and $A_{pr}$) (Fig. 1c). $A_{undiff}$ can be identified according to cell surface markers E-Cadherin + α6-integrin+ c-KIT− while spermatogonia at early differentiation stages co-express E-Cadherin and c-KIT[24]. By flow cytometry, few E-Cadherin+ spermatogonia remained and numbers of identified $A_{undiff}$ were significantly reduced at D10 after BU compared to controls, confirming spermatogonial depletion (Fig. 1d and Supplementary Fig. 1a). >90% of E-Cadherin+ cells were c-KIT− in BU-treated samples versus ~30% in

controls, indicating that essentially all spermatogonia at D10 post-BU are $A_{undiff}$ (Fig. 1d).

To characterise features of $A_{undiff}$ during germline regeneration, we isolated $A_{undiff}$ from control and BU-treated mice at D10 and analysed by RNA-Seq (Fig. 1e). 1949 genes were differentially expressed in regenerative $A_{undiff}$ compared to homoeostatic $A_{undiff}$ (false discovery rate < 0.05 and fold change >1.5) (Supplementary Data 1). Comparison of differentially expressed genes (DEGs) with a dataset of genes enriched in SSC and progenitor-enriched $A_{undiff}$ fractions isolated according to Oct4-GFP expression revealed a shift in gene expression programme (Fig. 1e)[5]. DEGs associated

**Fig. 1 Molecular characteristics of spermatogonia remaining in adult testis following BU-induced damage. a** Spermatogonial hierarchy and marker expression associated with distinct populations of A$_{undiff}$. **b** Kinetics of adult male germline recovery in response to treatment with low dose BU. Initiation of regenerative response occurs between D8 and 10. D10 post-BU was selected for subsequent analysis. **c** Representative wholemount IF of tubules D10 post-BU ($n = 2$ mice per group). **d** Flow-sorting strategy for isolation of A$_{undiff}$ (E-Cadherin+ c-KIT− α6-integrin+) for RNA-seq. Percentages of cells within gates are indicated ($n = 4$ mice per group). **e** Venn-diagram comparing differentially expressed genes (DEGs) in regenerative versus control A$_{undiff}$ (false discovery rate < 0.05 and absolute fold change > 1.5) and genes marking progenitor (Oct4-GFP+) or stem-associated (Oct4-GFP−) A$_{undiff}$ fractions. Heatmap shows selected DEGs in regenerative vs. control A$_{undiff}$ associated with stem and progenitor A$_{undiff}$. Genes of interest are highlighted. **f** Flow cytometry of adult Oct4-GFP testis D10 post-BU. Oct4-GFP expression within A$_{undiff}$ population is shown (E-Cadherin+ c-KIT− α6-integrin+). Percentages of A$_{undiff}$ Oct4-GFP + are included ($n = 3$ mice per group). **g** Representative wholemount IF of tubules D10 post-BU. Graph shows ratio of RARγ + (progenitor-enriched) to GFRα1+ (stem-enriched) spermatogonia ($n = 3$ mice per condition, 31–50 mm tubule length scored per animal). **h** Volcano plot of DEGs from **e** (analysis by Limma-voom and empirical Bayes method). Genes regulated by GDNF and FGF are highlighted, and relevant genes are indicated. **i** Representative flow cytometry analysis of adult WT testis D10 post-BU. EpCAM expression in A$_{undiff}$ from control versus BU-treated mice and A$_{diff}$ (E-CAD + c-KIT+) of controls is shown in histogram. Percentages of cells EpCAM+ are indicated. EpCAM is upregulated in E-Cadherin+ c-KIT + differentiating cells compared to A$_{undiff}$. Graph shows relative levels of EpCAM (median fluorescent intensity) on A$_{undiff}$ of control and BU-treated mice ($n = 3$ per group). Scale bars: 50 μm. Dashed lines in wholemount IF indicate seminiferous tubule profile. Data are presented as mean values ± SEM in **d**, **f**, **g**, **i**. Significance determined by two-tailed unpaired student $t$ test in **d**, **f**, **g**, **i**. Source data are provided as a Source Data file.

with the Oct4-GFP− SSC fraction (e.g., *Eomes*, *Lhx1*, *Nefm*, *Shisa6*, *Egr2*) were typically upregulated in regenerative A$_{undiff}$ (234 out of 242) while genes enriched in Oct4-GFP + progenitors (e.g., *Upp1*, *Rarg*, *Neurog3*, *Sox3*, *Piwil4*) were mostly downregulated (241 out of 258). Expression of SSC-associated genes *Gfra1* and *Id4* was not significantly altered. Given that the A$_{undiff}$ population is heterogenous and contains stem and progenitor fractions, our data indicated a switch in predominant fate towards a self-renewing state at the initiation of regeneration. Accordingly, when Oct4-GFP mice were treated with low-dose BU, the proportion of A$_{undiff}$ GFP + was substantially reduced at D10 (Fig. 1f). Wholemount IF confirmed that A$_{undiff}$ at D10 post-BU were predominantly GFRα1+ A$_s$ and A$_{pr}$ and relatively few spermatogonia were positive for progenitor marker RARγ (Fig. 1g).

Comparison of identified DEGs to GDNF-responsive genes in cultured A$_{undiff}$[30] revealed that regenerative A$_{undiff}$ typically upregulated genes stimulated by GDNF (54 out of 189, 29% of GDNF-responsive; $P < 2.02E-13$) and downregulated genes suppressed by GDNF (28 out of 71, 39% of GDNF-repressive; $P < 2.99E-11$) (Fig. 1h and Supplementary Data 1). While FGF-responsive genes in spermatogonia are poorly characterised, genes induced by FGF in other systems (*Spry4*, *Dusp6*, *Fgfr2*, *Spry1*)[31] were upregulated in regenerative A$_{undiff}$ (Fig. 1h). Both GDNF and bFGF stimulate the expression of SSC-associated genes *Lhx1*, *Bcl6b* and *Etv5* in cultured A$_{undiff}$[16,30], indicating that these factors regulate overlapping sets of genes. Analysis of DEGs in regenerative vs. homoeostatic A$_{undiff}$ revealed increased expression of adhesion molecules EpCAM and MCAM, which we confirmed by flow cytometry (Fig. 1i and Supplementary Fig. 1b). Both EpCAM and MCAM are induced by GDNF and/or bFGF stimulation of A$_{undiff}$ in vitro, suggesting they provide readout of growth factor stimulation[32,33]. Our data indicate that A$_{undiff}$ at initial regeneration stages are responding to increased levels of niche growth factors than under homoeostatic conditions, which may promote adoption of a self-renewing state.

We also assessed whether treatment was damaging for cells contributing to the SSC niche. In agreement with previous studies[8,34], we found no evident disruption of SOX9 + Sertoli or KIT + interstitial Leydig cell populations post-BU (Supplementary Fig. 1c). IF for smooth muscle actin revealed an intact layer of peritubular myoid cells surrounding the seminiferous tubules post-BU (Supplementary Fig. 1d). However, potential effects of BU on functional activity of these niche cells await further study.

**Heterogeneity and cellular dynamics of regenerating A$_{undiff}$.**
Gene expression analysis of A$_{undiff}$ suggested a switch in predominant fate during regeneration to an SSC state (Fig. 1e). To

characterise changes in composition of the A$_{undiff}$ pool during regeneration we performed single-cell RNA-Seq on A$_{undiff}$ sorted from pooled control and BU-treated mice at D10 (data from two independent experiments). After quality control, analysis was performed on 3798 control cells and 2669 cells from BU-treated mice (see Methods)[35]. Data were processed with Seurat, which revealed six cell clusters in control and BU samples (Fig. 2a; clusters 0–5). All clusters expressed general A$_{undiff}$ and spermatogonial markers (*Plzf/Zbtb16*, *Sall4*, *Foxo1*, *Ddx4*) (Supplementary Fig. 2a). Cluster 1 cells showed strong expression of *Neurog3*, *Rarg*, *Sox3*, *Upp1* and *Ddit4*, indicating it represented the progenitor population (Fig. 2b, c and Supplementary Fig. 2b)[5]. Cells in cluster 1 were substantially depleted from the regenerative A$_{undiff}$ pool but constituted a major fraction of homoeostatic A$_{undiff}$ (Fig. 2a), consistent with shift in A$_{undiff}$ composition towards self-renewing states during regeneration.

Cluster 2 cells co-expressed stem and progenitor-associated genes, suggesting they were transitioning between these states, and were more abundant within the regenerative A$_{undiff}$ pool (Fig. 2a–c and Supplementary Fig. 2a, b). These transitional cells were also identified by expression of *Stra8*, consistent with our previous study (Supplementary Fig. 2a)[5]. Cluster 3 cells were characterised by detectable expression of genes involved at late spermatogenic stages, including *Ldhc* and *Meig1* (Fig. 2c)[36]. The physiological relevance of cells in cluster 3 was unclear although expressed spermatogonial markers and were termed undefined A$_{undiff}$ (Supplementary Fig. 2a).

We observed 3 cell clusters (0, 4 and 5) expressing high levels of SSC-associated genes *Gfra1*, *Nefm*, *Lhx1*, *Lhx2*, *Egr2*, *Etv5*, *Foxc2*, and *Id4* (Fig. 2a–c, Supplementary Fig. 2a, b and Supplementary Data 2)[5]. A feature distinguishing these potential SSC clusters included expression of cell cycle-related genes (e.g., *Cdc20*, *Ube2c*, *Aurka*, *Top2a*, *Aspm*) (Fig. 2b and Supplementary Fig. 2b). Cluster 4 and 5 cells had the most pronounced expression of cell cycle genes and were termed proliferative SSC-1 and SSC-2 respectively, while cluster 0 cells expressed lower levels of these genes and were termed primitive SSCs in part due to this quiescent signature (see below). Interestingly, cluster 4 and 5 cells were characterised by high levels of *Epcam* expression (Fig. 2c and Supplementary Fig. 2a). Proliferative SSC clusters were more prominent in the regenerative A$_{undiff}$ population while the proportion of primitive/quiescent SSCs remained comparable to controls, consistent with expansion of mitotically active SSCs within the regenerative A$_{undiff}$ pool (Fig. 2a). Gene ontology (GO) analysis of DEGs in each cluster supported distinct cell cycle status of SSC subsets and highlighted importance of signalling pathway control (protein kinase inhibition) for the primitive SSC

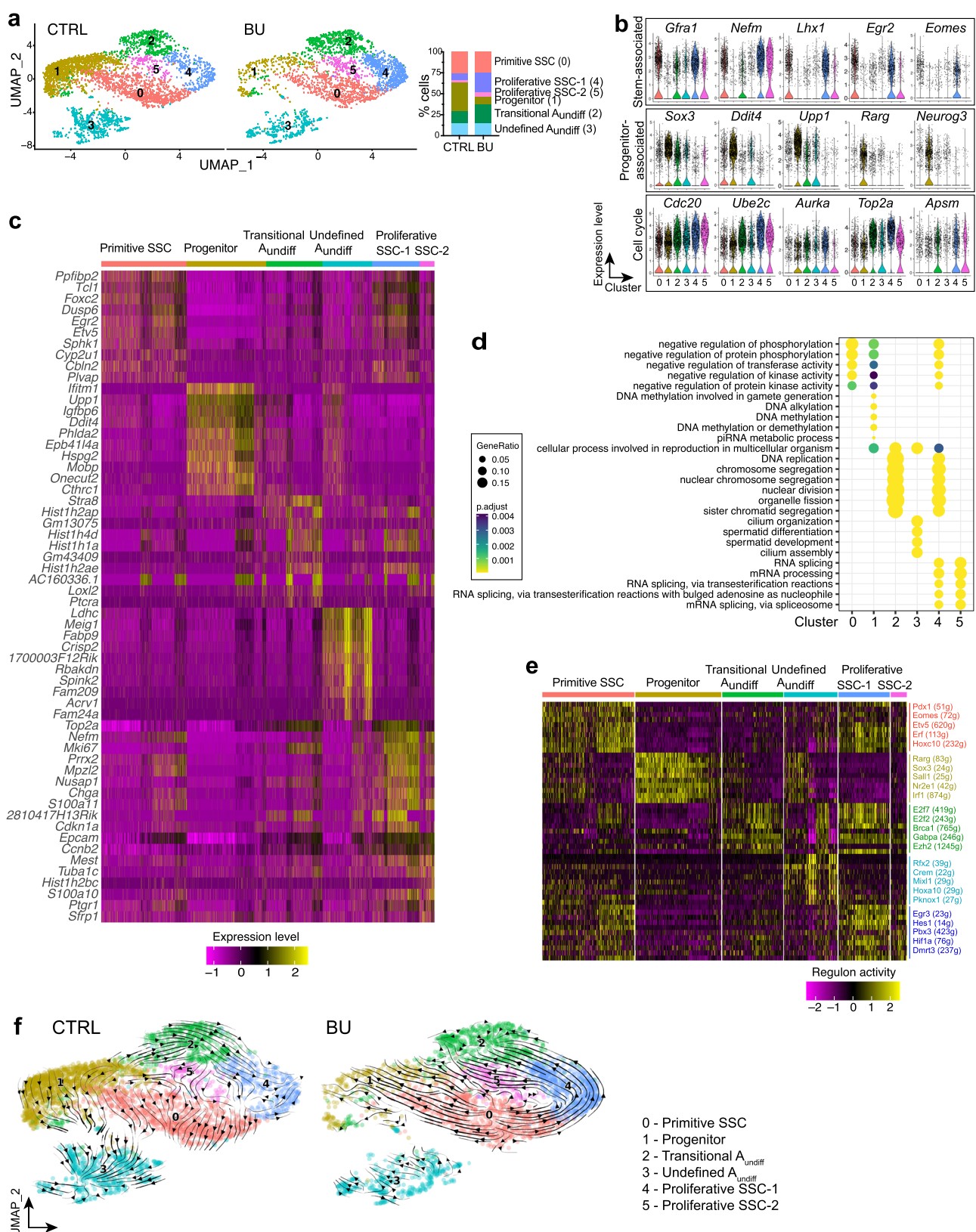

state and RNA splicing/processing for proliferative SSC states (Fig. 2d). Our analysis indicated that SSCs exist in a series of functional states with distinct cell-cycle status and regulatory pathways and that proliferative SSC states are more evident during regeneration. Given that $A_{undiff}$ proliferation varies according to periodic cycling of the seminiferous epithelium

and function can be affected by proximity to sources of growth factors[2,19], these clusters may represent SSCs from different cycle stages and/or distinct locations within the seminiferous tubule.

To gain insight into identity of $A_{undiff}$ clusters and associated regulatory mechanisms, we analysed gene regulatory networks using SCENIC, which calculates activity of regulons

**Fig. 2 Dynamic heterogeneity of the A_undiff population during initiation of regeneration. a** Visualisation of single-cell RNA-Seq data from A_undiff (E-Cadherin + α6-integrin+ c-KIT−) isolated from CTRL and BU mice at D10 by UMAP. Bar chart illustrates distribution of A_undiff from CTRL and BU-treated samples into different cell clusters and identity of each cluster. **b** Violin plots comparing expression of selected markers associated with stem and progenitor populations in cell clusters from **a**. Cell cycle-associated genes that delineate cell clusters are shown in bottom panels. **c** Heatmap of top 10 DEGs in cell clusters identified from analysis of **a**. **d** GO terms associated with gene expression profiles of cell clusters identified in **a**. GO terms enrichment identified by one-sided Fisher's exact test with Benjamini–Hochberg correction for multiple comparisons. **e** SCENIC analysis identifying cluster-enriched regulons of control samples from **a**. Heatmap of top ten regulons that define each cluster or cell state are shown and selected regulons highlighted. Number of genes (g) in each regulon is indicated. **f** scVelo plot embedding in the UMAP space reveals directional progression of transcriptional states.

(transcription factors and putative targets) in individual cells, allowing identification of transcription factors controlling cell states (Fig. 2e, Supplementary Fig. 2c and Supplementary Data 2)[37]. Primitive SSCs were predicted to be driven by transcription factors associated with SSC function, including PDX1, EOMES and ETV5[5,12,16]. Progenitors exhibited activity of RARγ, SOX3 and SALL1. SOX3 and RARγ play key roles in SSC differentiation priming, validating cluster identity[38,39]. Proliferative SSC clusters were regulated by similar transcription factors including EGR3, HES1, HIF1A and DMRT3 while transitional A_undiff were distinguished by activity of E2F7, E2F2 and EZH2 regulons.

To compare A_undiff dynamics under homoeostatic and regenerative conditions, we performed RNA-velocity analysis (scVelo)[40]. RNA-velocity was visualised by projection onto UMAP plots where direction and magnitude of arrows indicates cell state progression based on mRNA splicing analysis (Fig. 2f). In control A_undiff, transition of cells between different clusters or states was generally limited, consistent with a relatively static cell population during homoeostasis. In contrast, regenerative A_undiff displayed striking velocity streamlines between clusters, indicating rapid transitions between subsets that correlated with increased cell cycle (Fig. 2f and Supplementary Fig. 3a). Within regenerative A_undiff, velocity-inferred directionality indicated that primitive SSCs transitioned into both proliferative SSCs and progenitors. This suggests that during regeneration, SSCs are triggered to expand the SSC pool through proliferation and generate progenitors for spermatogenic recovery. Moreover, that quiescent SSCs can form progenitors without transiting through a mitotically active state. Prediction of genes driving transitions indicated that *Cdk1*, *Mki67*, *Cenpe* and *Usp26* promoted transition of primitive SSCs to the proliferative state, supporting cell cycle involvement, whereas *Foxo1*, *Tmtc4*, *Rbms2* and *Dnmt3b* promoted transition of primitive SSCs to progenitors (Supplementary Fig. 3b). *Foxo1* and *Dnmt3b* are linked with A_undiff differentiation[22,41]. Combined, RNA-velocity analysis indicates that A_undiff undergo more dynamic cellular transitions during regeneration than under homoeostatic conditions and highlights the distinct fates available to regenerative SSCs.

**Distinct cellular state of regenerative GFRα1 + spermatogonia.** Clustering analysis of scRNA-seq data indicated that an increased proportion of regenerative A_undiff adopted an SSC state compared to controls. However, SSCs mediating germline regeneration may be functionally distinct from those of homoeostatic tissue and we therefore characterised DEGs between SSCs of homoeostatic and regenerative testis to gain insight into unique features. As the A_undiff population likely comprises a continuum of cell states rather than discrete subsets as predicted from clustering analysis and the GFRα1+ fraction forms the homoeostatic SSC pool[5,42], we initially selected SSCs from CTRL and BU scRNA-Seq datasets based on *Gfra1* expression (normalised expression level >2; 755 and 1331 cells respectively) and identified DEGs (adjusted $P < 0.05$) (Fig. 3a and Supplementary Data 3). GO analysis revealed terms associated with cell division, response to stimulus

plus stress, metabolic processes, chromosome organisation and modification (Fig. 3b). *Gfra1* + cells from BU-treated samples upregulated genes involved in cell cycle (*Mki67*, *Kif4*, *Top2a*, *Smarca5*, *Hells*), epigenetic regulation (*Hist1h1a*, *Suz12*, *Dnmt1*, *Ezh2*) and genes of interest including *Igf2* and *Plaur* (Fig. 3a). Consistent with suppression of stem-progenitor transition, regenerative *Gfra1* + cells downregulated progenitor-associated genes (*Sohlh1*, *Sox3*, *Upp1*). Regenerative *Gfra1* + cells also downregulated *Pdx1*, a marker of homoeostatic SSCs[5]. To corroborate our analysis with an alternative SSC marker, we identified DEGs in cells expressing *Eomes* (345 cells in CTRL and 667 in BU samples)[5,12]. Similar genes and GO terms were found when comparing DEGs in *Eomes* and *Gfra1*-positive populations from regenerative vs. homoeostatic testis (Fig. 3b and Supplementary Fig. 4a).

As the proportion of cells in distinct SSC states changes during regeneration (Fig. 2a), DEGs identified in regenerative *Gfra1*/*Eomes* + cells may be reflective of composition of the SSC pool rather than features of regenerative SSCs. Projection of selected DEGs from analysis of regenerative vs. homoeostatic *Gfra1*/*Eomes* + cells onto UMAP plots indicated that genes associated with proliferation (*Mki67*, *Kif4*) and regeneration (*Eomes*, *Plaur*, *Igf2*, *Epcam*) were induced within SSC clusters of regenerative A_undiff (Fig. 3c), supporting changes in SSC behaviour during regeneration. To confirm this observation, we defined DEGs between cells of the same SSC clusters from regenerative and homoeostatic samples (cluster 0 primitive and cluster 4 proliferative). Similar sets of DEGs were identified in SSC clusters as in *Gfra1*/*Eomes* + cells, validating our approach and revealing the distinct functional status of regenerative A_undiff (Supplementary Fig. 4b, c and Supplementary Data 3).

Interestingly, regenerative *Gfra1*/*Eomes* + cells and SSC clusters upregulate *Plaur*, encoding urokinase-type plasminogen activator receptor (uPAR), a regulator of cell migration and invasion[43]. A uPAR+ spermatogonial population with increased migratory potential and expression of SSC-associated genes is present in neonatal testis[44]. While uPAR+ spermatogonia were largely absent in adult control tubules, uPAR was detected on a substantial fraction of regenerative A_undiff (Fig. 3d and Supplementary Fig. 4d). By flow cytometry, ~30% of E-Cadherin+ c-KIT− A_undiff were uPAR+ in BU-treated samples while <5% were uPAR+ in controls (Fig. 3e). Our data reveal a unique gene expression signature of SSCs in a regenerative environment and demonstrate that regenerative A_undiff are marked by uPAR expression that may contribute to their functional properties[43].

A_undiff exist in a series of functional states and physiological condition of the niche (e.g. during development, homoeostasis or regeneration) may promote adoption of distinct states[5]. By comparing our data with scRNA-Seq of neonatal A_undiff[36], we noticed similarities in identified DEGs (Fig. 3f and Supplementary Data 4). A significant fraction of genes upregulated in regenerating *Gfra1* + cells were also upregulated in neonatal *Gfra1* + cells when compared to *Gfra1* + cells from homoeostatic adults (e.g., *Usp26*, *Top2a*, *Txn1*, *Bcl6b*). Conversely, a significant number of genes downregulated in *Gfra1* + cells of regenerative samples

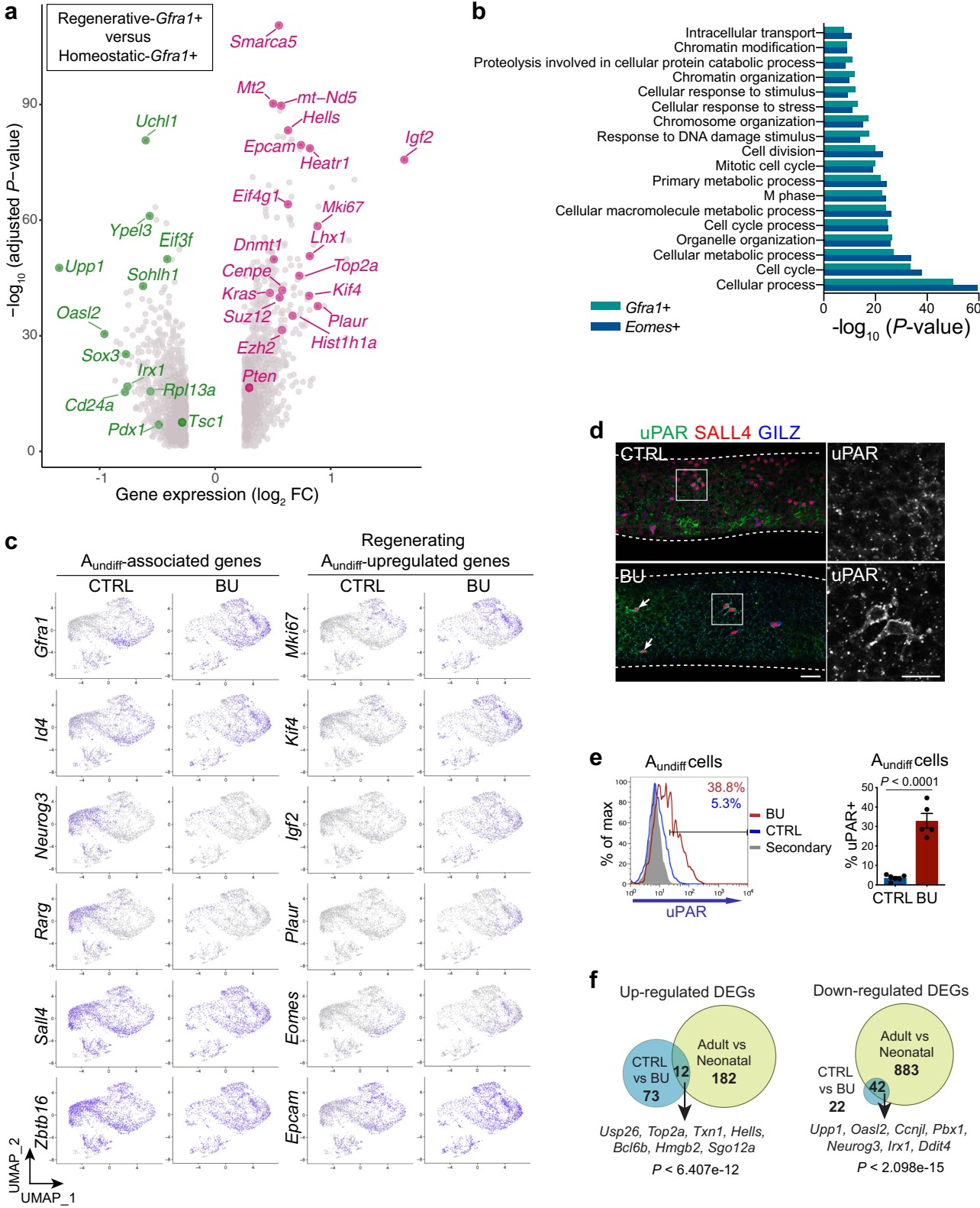

were also downregulated in neonatal samples (e.g., *Oasl2*, *Upp1*, *Neurog3, Ddit4*). Few common DEGs in *Gfra1* + cells with opposite trends in expression were found (Supplementary Fig. 5a). Equivalent results were observed when examining all A$_{undiff}$ rather than *Gfra1* + fractions (Supplementary Fig. 5b, c). We concluded that regenerating A$_{undiff}$ share features with neonatal A$_{undiff}$, particularly in terms of cell cycle-related genes

(Supplementary Data 4), reflecting similar demands on these cells during development and regeneration.

**Enhanced growth factor signalling in regenerative A$_{undiff}$.** SSC function is dependent on growth factors produced within the niche and balanced activation of downstream signalling pathways[1]. However, consequences of germline damage on

**Fig. 3 Comparative analysis of regenerative and homoeostatic A$_{undiff}$ by single-cell RNA-seq. a** Volcano plot of DEGs (MAST differential expression test with Bonferroni correction, adjusted $P$ value < 0.05) within *Gfra1*+ cells (normalised expression level > 2) from scRNA-seq analysis of sorted A$_{undiff}$ (E-Cadherin + α6-integrin+ c-KIT−) from CTRL and BU-treated mice at D10 (755 *Gfra1*+ cells in CTRL and 1331 in BU). Genes of interest are highlighted. **b** GO of DEGs from comparison of *Eomes*+ (345 cells in CTRL and 667 in BU) or *Gfra1*+ cells from scRNA-seq analysis of **a**. $P$ values from one-sided Fisher's Exact test. **c** UMAP plots showing clustering analysis of scRNA-Seq data of CTRL and D10 BU A$_{undiff}$ of **a** (3798 cells in CTRL and 2669 in BU-treated). Expression of selected genes associated with A$_{undiff}$ (left panels), and genes upregulated in regenerative A$_{undiff}$ (right panels) are shown. **d** Representative wholemount IF of tubules D10 post-BU ($n = 3$ per group). Arrows: uPAR+ A$_s$ cells. Scale bar: 50 μm. Dashed lines indicate seminiferous tubule profiles. **e** Representative flow cytometry analysis of adult mice treated with BU vs. CTRL at D10. Graph shows mean percentage of A$_{undiff}$ (E-Cadherin+ c-KIT−) uPAR+ ± SEM ($n = 5$ per group). Significance determined by two-tailed unpaired student $t$ test. **f** Venn diagrams illustrating overlap of up and downregulated DEGs (Fold change > 1.5 and adjusted $P$ value < 0.05) within *Gfra1*+ spermatogonia from CTRL vs. BU-treated A$_{undiff}$ from analysis of **a** and CTRL (adult) vs. neonatal ID4$^{bright}$ spermatogonia[36]. $P$ values derived from hypergeometric tests are shown. Examples of concordantly regulated DEGs are indicated. Source data are provided as a Source Data file.

growth factor signalling within SSCs and the role of these pathways in regeneration remain unclear. Gene expression profiling of A$_{undiff}$ at initial stages of regeneration indicated enhanced stimulation by growth factors including GDNF and FGFs (Fig. 1h). To confirm this prediction and dissect mechanisms underlying the regenerative response, we analysed activity of pathways regulated by GDNF and FGFs (Fig. 4a). As readout of PI3K/AKT activity we analysed FOXO1 localisation, which is exported from the nucleus following AKT-dependent phosphorylation (Supplementary Fig. 6a, b)[22,24]. By wholemount analysis of controls, we found that FOXO1 is predominantly localised to the nucleus (N) or both nucleus and cytosol (C + N) of GFRα1+ spermatogonia, indicating low levels of AKT activity in homoeostatic SSCs (Fig. 4b). In contrast, in GFRα1+ cells D10 post-BU, there was a significant shift in FOXO1 to the cytosol, indicating that regenerative A$_{undiff}$ hyperactivate AKT (Fig. 4b).

An additional effector downstream PI3K/AKT is mTORC1 (Fig. 4a). Chronic mTORC1 activation is detrimental for SSC maintenance but mTORC1 signalling is linked to regeneration of multiple tissues[45,46]. To assess whether mTORC1 is activated in regenerative A$_{undiff}$, we analysed phosphorylated ribosomal protein S6 (P-S6) by wholemount IF of BU-treated samples. GFRα1+ spermatogonia are infrequently P-S6 + during homoeostasis, indicating low mTORC1 activity (Fig. 4c)[25]. However, a significantly increased proportion of GFRα1+ cells (3.9-fold) were P-S6 + in D10 BU-treated samples (Fig. 4c), indicating mTORC1 activation in regenerative A$_{undiff}$. Consistent with scRNA-Seq analysis and cellular role of AKT/mTORC1 signalling, the proportion of GFRα1+ cells positive for KI67 increased ~4-fold during regeneration, indicating enhanced mitotic activity (Fig. 4c). Our data suggest that regenerative A$_{undiff}$ are subjected to growth factor stimulation that activates AKT/mTORC1 signalling and proliferation.

**mTORC1 activation is required for initiation of regeneration.** Given mTORC1 activation in regenerative A$_{undiff}$ and role of this pathway in tissue regeneration[45,46], we assessed whether mTORC1 was required for germline regeneration. Mice were treated with BU and 3 days later (prior to initiation of regeneration) treated daily with the mTORC1 inhibitor rapamycin or vehicle for 1 week before analysis at D10 and later points (Fig. 4d). Rapamycin blocked mTORC1 activation occurring in GFRα1+ spermatogonia at D10 post-BU as indicated by lack of P-S6 (Fig. 4d). Flow cytometry analysis demonstrated that rapamycin did not affect depletion of A$_{undiff}$ (PLZF + c-KIT−) or early differentiating cells (PLZF + c-KIT+) post-BU (Fig. 4e). However, rapamycin suppressed increase of KI67 + cells within the PLZF + c-KIT− population, indicating that mTORC1 plays an important role in promoting proliferation of regenerative A$_{undiff}$ (Fig. 4e). Rapamycin did not disrupt increased frequency of EOMES + cells in the PLZF + pool after BU but inhibited

elevated mitotic activity of EOMES + cells (Supplementary Fig. 6c). While cytostatic effects of rapamycin might be due to apoptosis of proliferating cells, treatment of cultured A$_{undiff}$ with rapamycin induces cell cycle arrest without increasing numbers of apoptotic cells (Supplementary Fig. 6d)[23].

In BU plus vehicle-treated samples, abundance of GFRα1+ spermatogonia was substantially reduced at D10 compared to controls but recovered to normal levels by D17 (Fig. 4f, g). GFRα1+ A$_{al}$ are infrequently observed in homoeostasis but evident at D17 post-BU during regeneration (Fig. 4f)[29]. KI67 staining indicated that GFRα1+ spermatogonia of vehicle-treated samples were more mitotically active than untreated controls at D10 and D17 post-BU but by D30 the proliferation rate had returned to normal, coincident with restoration of the spermatogonial pool (Fig. 4f, h). RARγ + progenitors were rarely found D10 post-BU but reappeared by D17 in vehicle-treated samples and accumulated over time (Figs. 1g and 4i). Our data indicate that GFRα1+ cells transiently increase proliferation following germ cell depletion to restore the SSC pool and then generate progenitors for spermatogenic recovery.

Importantly, inhibiting mTORC1 with rapamycin during the initiation of regeneration substantially delayed germline recovery. At D17 post-BU, abundance of GFRα1+ cells in rapamycin-treated mice had not recovered from levels at D10 and GFRα1+ A$_{al}$ were not observed (Fig. 4f, g). However, by D30 the GFRα1+ population of rapamycin-treated mice had recovered and exhibited an overshoot in numbers that normalised by D90 (Fig. 4g). Rapamycin blocked increased proliferation of GFRα1+ spermatogonia at D10 and D17 post-BU as indicated by KI67 (Fig. 4f, h). Delayed recovery of the GFRα1+ population in rapamycin-treated mice was accompanied by delayed reappearance of RARγ + progenitors that became evident by D30 (Fig. 4i). By D90, the seminiferous epithelium of rapamycin-treated mice had recovered and appeared comparable to vehicle-treated controls (Supplementary Fig. 6e). These data confirmed that mTORC1 plays a critical role in induction of A$_{undiff}$ regenerative responses but that the suppressive effects of temporary mTORC1 inhibition on regeneration did not compromise long-term regenerative capacity.

**Inhibition of growth factor signalling disrupts regeneration.** A$_{undiff}$ are evidently triggered to transition into a regenerative state through stimulation by niche factors including GDNF and FGFs. Growth factor stimulation of A$_{undiff}$ will activate multiple signalling pathways besides mTORC1 that can be required for regeneration. We therefore assessed whether broader inhibition of growth factor signalling would lead to profound disruption of regeneration.

To achieve this, mice were treated with the multikinase inhibitor AD80 or vehicle daily from D5 to D9 following BU and testes analysed at D10 and D30 (Fig. 5a). AD80 was developed

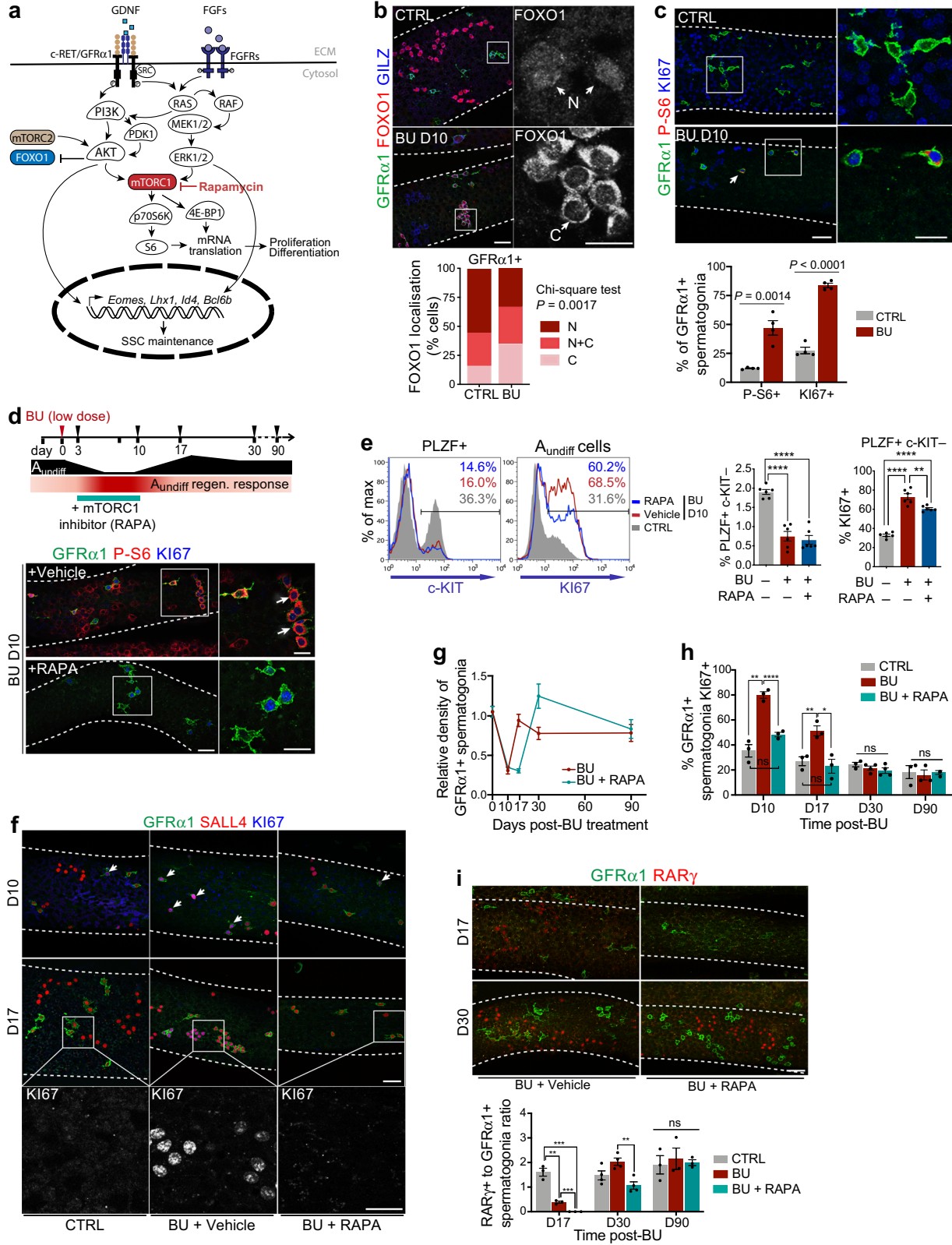

from a screen for inhibitors of RET-dependent signalling and targets RET plus downstream kinases SRC, RAF and S6K (Supplementary Fig. 7a)[47,48]. Given broad specificity of AD80 and the role of RET as GDNF receptor component, AD80 could inhibit AKT, ERK MAPK and mTORC1 activation by GDNF in cultured $A_{undiff}$ and partially supress bFGF-dependent signalling (Supplementary Fig. 7b). Strikingly, GFRα1 expression in

surviving SALL4 + spermatogonia at D10 post-BU was substantially reduced by AD80 treatment (Supplementary Fig. 7c). However, persisting $A_s$ and $A_{pr}$ could be identified by IF for E-Cadherin (Fig. 5a, b). Whereas E-Cadherin+ $A_s$/$A_{pr}$ of D10 vehicle-treated controls exhibited mostly cytosolic FOXO1 as anticipated, AD80 treatment caused re-localisation of FOXO1 to the nucleus, indicating that AD80 inhibited AKT signalling in

**Fig. 4 Growth factor signalling in regenerative $A_{undiff}$ and role of mTORC1 in regeneration. a** Pathways activated by niche factors to regulate SSCs. **b** Representative IF of wholemount tubules D10 post-BU. Right panels: Higher magnification images of FOXO1 in highlighted regions. Arrows: FOXO1 localisation (C, predominantly cytosolic; N, predominantly nuclear; N + C, nuclear and cytosolic). Graph shows FOXO1 distribution within GFRα1+ spermatogonia and $P$-value ($\chi^2$ test, two-sided) ($n = 4$ control, $n = 3$ BU-treated; >47 mm tubule length analysed per sample). **c** Representative wholemount IF of tubules D10 post-BU. Higher magnification images of indicated regions are shown. Arrow: selected P-S6 + $A_{undiff}$. Graph shows percentage of GFRα1+ spermatogonia P-S6 + and KI67 + in CTRL and BU-treated samples ($n = 4$ per group). Significance by two-tailed unpaired $t$ test. **d** Timeline illustrating $A_{undiff}$ recovery from BU and the RAPA regimen to block mTORC1. Right panels: Representative wholemount IF of tubules from D10 mice treated with BU then RAPA or vehicle daily according to timeline. Higher magnification images of indicated regions are shown. Arrows: P-S6 + $A_{undiff}$. **e** Representative flow cytometry of testis from D10 mice treated as in **d**. Graphs indicate percentage of testis cells $A_{undiff}$ (PLZF + c-KIT−) and percentage of $A_{undiff}$ KI67 + ($n = 6$ mice per group). **f** Representative wholemount IF of tubules from D10 and D17 mice treated as in **d**. Arrows: KI67 + GFRα1+ cells. Higher magnification images of KI67 from indicated regions are shown. **g** Relative density of GFRα1+ spermatogonia along tubules of mice treated as in **d** ($n = 3$ per group and time point; $n = 4$ per group at D30; 40–60 mm of tubule length scored per mouse). **h** Graph shows percentage of GFRα1+ spermatogonia KI67 + from analysis of **f** ($n = 3$ per group per time point; $n = 4$ per group at D30). **i** Representative wholemount IF of tubules from mice treated as in **d**. Graph shows ratio of RARγ+ to GFRα1+ spermatogonia ($n = 3$ mice per condition, minimum 50 mm tubule length scored per animal). Scale bars: 50 μm (main panels) and 25 μm (magnified areas). Dashed lines: seminiferous tubule profiles. Data presented as mean values ± SEM in **c**, **e**, **g–i**. Significance by one-way ANOVA with Tukey's multiple comparisons test in **e**, **h**, **i** ($^*P < 0.05$; $^{**}P < 0.01$; $^{***}P < 0.001$; $^{****}P < 0.0001$; not significant (ns) $P > 0.05$). Source data are provided as a Source Data file.

regenerative $A_{undiff}$ (Fig. 5a). In contrast, rapamycin did not suppress AKT in regenerative GFRα1+ spermatogonia as indicated by cytosolic FOXO1 (Supplementary Fig. 7d). Like rapamycin, AD80 inhibited mTORC1 in regenerative $A_{undiff}$ as indicated by reduction in the proportion of E-Cadherin+ $A_s/A_{pr}$ positive for P-S6 (Fig. 5b). Our data demonstrate that AD80 inhibits multiple signalling pathways in regenerative $A_{undiff}$.

To assess effects of AD80 on the regenerative $A_{undiff}$ state, we analysed D10 post-BU testis cells by flow cytometry. Both vehicle and AD80-treated samples contained similarly low numbers of PLZF + c-KIT− $A_{undiff}$ (Supplementary Fig. 7e). However, the proportion of regenerative $A_{undiff}$ that were KI67+ was reduced by AD80, indicating inhibition of proliferation (Fig. 5c). Importantly, the proportion of $A_{undiff}$ that were EOMES+, normally enhanced during regeneration, was substantially lower (Fig. 5c). AD80 also blocked uPAR induction in E-Cadherin+ c-KIT− $A_{undiff}$ after BU and reduced levels of EpCAM (Fig. 5d), key features of the regenerative state. Our data demonstrate that AD80 inhibits gene expression signature, cellular signalling, and proliferation of regenerative $A_{undiff}$.

Given striking effects of AD80 on regenerative $A_{undiff}$ at D10 post-BU, we assessed effects on germline recovery at D30 following treatment cessation (Fig. 5e). In vehicle-treated controls, ~90% of the tubule length contained SALL4 + spermatogonia and active spermatogenesis at D30, confirming germline recovery (Fig. 5e, f). However, multiple tubule regions of AD80-treated samples were devoid of SALL4 + spermatogonia and spermatogenesis (~40% of length), indicating depletion and/or functional disruption of regenerative $A_{undiff}$ (Fig. 5e, f). Within spermatogenic patches of AD80-treated samples, GFRα1+ spermatogonia were at higher densities than in vehicle controls and an increased proportion was KI67 + (Fig. 5f and Supplementary Fig. 7f), consistent with a delayed and on-going regenerative response. Enhanced proliferative status of $A_{undiff}$ and EOMES + cells in AD80-treated samples were evident by flow cytometry (Supplementary Fig. 7g, h). Recovery of GFRα1/EOMES + populations by D30 demonstrated that the negative effects of AD80 on *Gfra1* and *Eomes* expression in $A_{undiff}$ at D10 were reversible, highlighting plasticity of $A_{undiff}$ transcriptional states. Our data indicate that broad inhibition of growth factor signalling during initiation of regeneration results in depletion of regenerative $A_{undiff}$ and disruption in germline recovery.

**Role of FOXM1 in induction of germline regeneration.** While the importance of growth factor signalling in induction of regeneration is clear, cellular effectors that underlie the switch in

$A_{undiff}$ behaviour remain incompletely understood. Pathway analysis of DEGs in regenerative vs. homoeostatic $A_{undiff}$ indicated positive regulators of the regenerative state (Fig. 6a and Supplementary Data 5). One factor predicted to be activated in regenerative $A_{undiff}$ was the Forkhead box M1 (FOXM1) transcription factor, a cell cycle regulator involved in liver regeneration and pancreatic β-cell proliferation[49,50]. Given the role of FOXM1 in proliferation and tissue repair plus modulation by growth factor signalling[49–51], we considered that FOXM1 could be a regulator of regenerative $A_{undiff}$.

From RNA-Seq data, FOXM1 targets involved in cell cycle were upregulated in regenerative $A_{undiff}$ (e.g., *Ccnb1*, *Ccnd1*, *Ccne1*, *Ccna2*, *Cdc25a*, *Top2a*, *Birc5*, *Plk1*) (Fig. 6b). *Plaur* (uPAR) is a FOXM1 target[52] and induced in regenerative $A_{undiff}$ (Figs. 3d, e, 6b). Within adult testis FOXM1 was detected in spermatocytes plus subsets of PLZF + spermatogonia and upregulated within the GFRα1+ population after BU (Fig. 6c and Supplementary Fig. 8a, b). FOXM1 targets were also significantly elevated within *Gfra1* + cells and SSC clusters from scRNA-Seq data, confirming FOXM1 activation during regeneration (Supplementary Fig. 8c, d and Supplementary Data 3).

As roles of FOXM1 in $A_{undiff}$ are undefined, we assessed effects of FOXM1 inhibition on cultured $A_{undiff}$. Interest in FOXM1 as a target for cancer therapy led to identification of novel compounds that directly inhibit FOXM1, including the antibiotic thiostrepton[53,54]. Cultured $A_{undiff}$ were treated with thiostrepton and expression of FOXM1 targets analysed (Fig. 6d). As FOXM1 phosphorylation by cyclin D-CDK4/6 promotes transcriptional activity and stability[55], we treated cells with CDK4/6 inhibitor palbociclib as positive control for reduced FOXM1 function. By RT-qPCR, thiostrepton and palbociclib strongly reduced expression of cell cycle-associated FOXM1 targets (*Ccnb1*, *Cdc25b*, *Ccne1*, *Ccna2*, *Birc5*, *Plk1*, *Kif4*) and *Foxm1* itself (Fig. 6d), consistent with previous studies and FOXM1 auto-regulation[56,57]. By western blot, treating cultured $A_{undiff}$ with thiostrepton or palbociclib led to substantial reduction in FOXM1 plus target CCNB1 (Supplementary Fig. 8e). An alternative FOXM1 inhibitor (FDI-6) was less effective at suppressing FOXM1 and targets in $A_{undiff}$ (Supplementary Fig. 8e)[58]. Cell cycle analysis of cultured $A_{undiff}$ treated with thiostrepton showed G1 arrest (Fig. 6e), supporting the role of FOXM1 in cell cycle and effects of thiostrepton on expression of cell cycle-related genes (Fig. 6d and Supplementary Fig. 8e).

To confirm involvement of FOXM1 in $A_{undiff}$ proliferation, we transduced cultures with *Foxm1* constructs and assessed effects on growth. Different *Foxm1* isoforms can have distinct

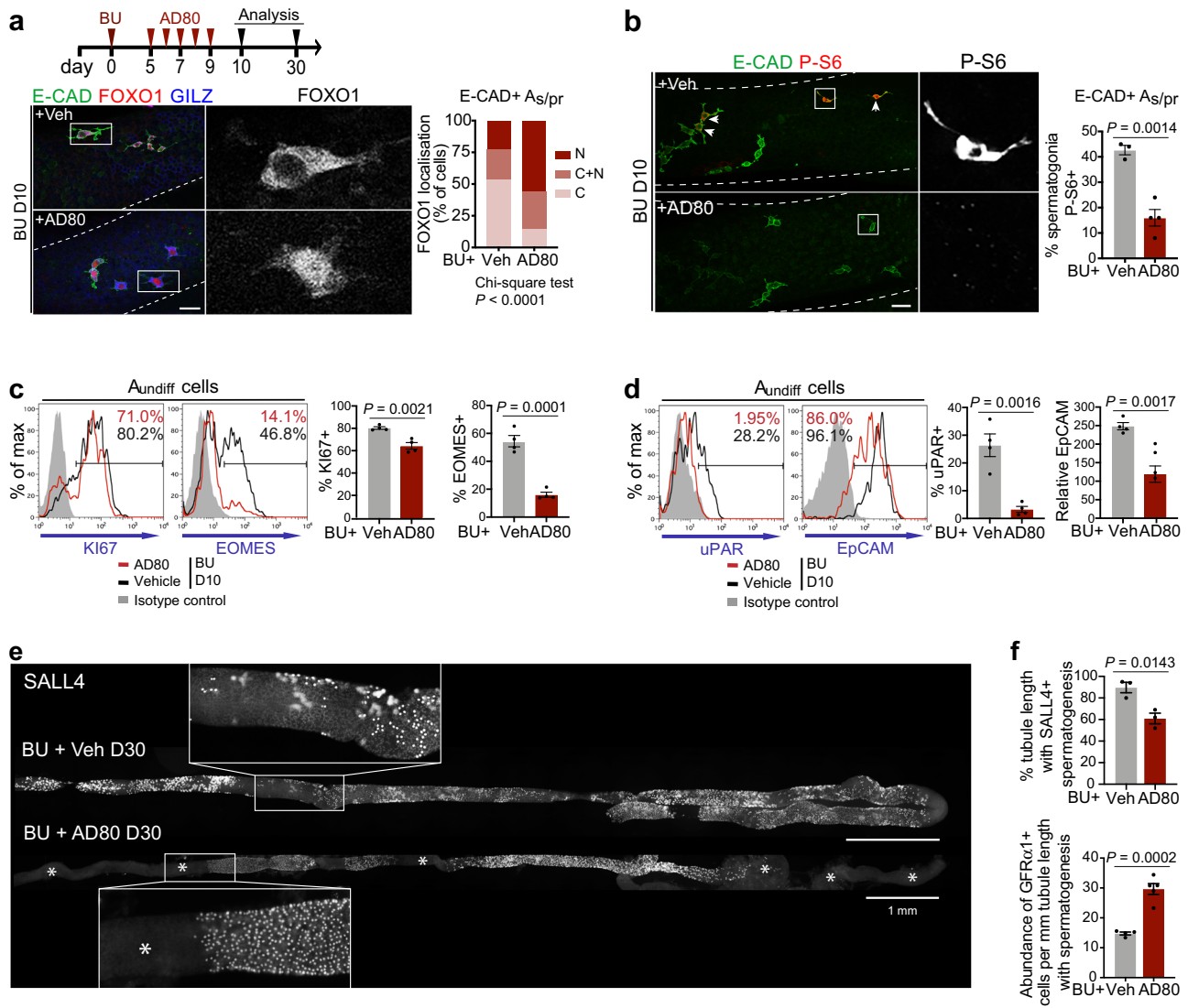

**Fig. 5 Molecular features and function of regenerative A_undiff are dependent on growth factor signalling. a** Upper panel: treatment regimen and analysis timeline for mice receiving multikinase inhibitor AD80 after BU. Lower panels: Representative wholemount IF of adult mice treated with BU then AD80 or vehicle daily for 5 days from D5 post-BU. Higher magnification grayscale images illustrating FOXO1 localisation in highlighted cells are shown. Graph shows FOXO1 distribution within E-CADHERIN + A_s/A_pr at D10 (N, predominantly nuclear; N + C, nuclear and cytosolic; C, predominantly cytosolic) and P-value ($\chi^2$ test, two-sided) (n = 3 vehicle and n = 4 AD80 treated; >100 mm tubule length analysed per sample). **b** Representative wholemount IF of mice treated as in **a**. Higher magnification grayscale images of P-S6 in highlighted regions are shown. Graph indicates percentage of E-CADHERIN + A_s/A_pr P-S6 + at D10 (n = 3 vehicle and n = 4 AD80). **c, d** Representative flow cytometry of testis from mice treated as in **a** at D10. Graphs indicate percentage of A_undiff (PLZF + c-KIT–) KI67 + and EOMES + in **c** and percentage of A_undiff (E-Cadherin+ c-KIT–) uPAR+ plus relative levels of EpCAM on A_undiff (median fluorescent intensity) in **d** (n = 4 mice per group). **e** Representative wholemount IF grayscale images illustrating SALL4 + populations in tubules from mice treated as in **a** at D30. Images taken along tubule length (n = 3 mice per group, >8 cm tubule length analysed per vehicle-treated control and >14 cm per AD80-treated animal). Insets show magnified regions of highlighted areas. Asterisks: tubule regions lacking SALL4 + germ cells and spermatogenesis. **f** Upper graph: Percentage of tubule length containing SALL4 + cells and spermatogenesis (n = 3 mice per group). Lower graph: Abundance of GFRα1+ spermatogonia per mm tubule length within spermatogenic regions (n = 4 mice per group). Dashed lines in **a**, **b**: tubule outline. Scale bars: 50 μm in **a**, **b** and 1 mm in **e**. Data presented as mean values ± SEM in **b**, **c**, **d**, **f**. Significance by two-tailed unpaired t test in **b**–**d**, **f**. Source data are provided as a Source Data file.

functions[59] and *Foxm1c* was the predominant isoform in cultured A_undiff (Supplementary Fig. 8f). Cultures were transduced with lentivirus containing tagged *Foxm1c* or dominant-negative *Foxm1c* lacking the transactivation domain (ΔC597) (Supplementary Fig. 8g)[51]. Growth of cells overexpressing *Foxm1c* was significantly increased compared to TdTomato-transduced controls while cultures expressing the ΔC597 mutant expanded more slowly (Fig. 6f), supporting a role for FOXM1 in A_undiff proliferation.

To validate a role for FOXM1 in germline regeneration, mice were treated daily with thiostrepton or vehicle from D3 to D9 post-BU and analysed at D10 (Fig. 6g)[53,60]. Flow cytometry revealed a modest but significant reduction in fraction of A_undiff (PLZF + c-KIT–) that were KI67+ in thiostrepton vs. control samples (Fig. 6g), indicating a role for FOXM1 in promoting regenerative A_undiff proliferation. A_undiff abundance was unaffected by thiostrepton at this timepoint (Supplementary Fig. 8h). Strikingly, from wholemount IF, thiostrepton significantly reduced the proportion

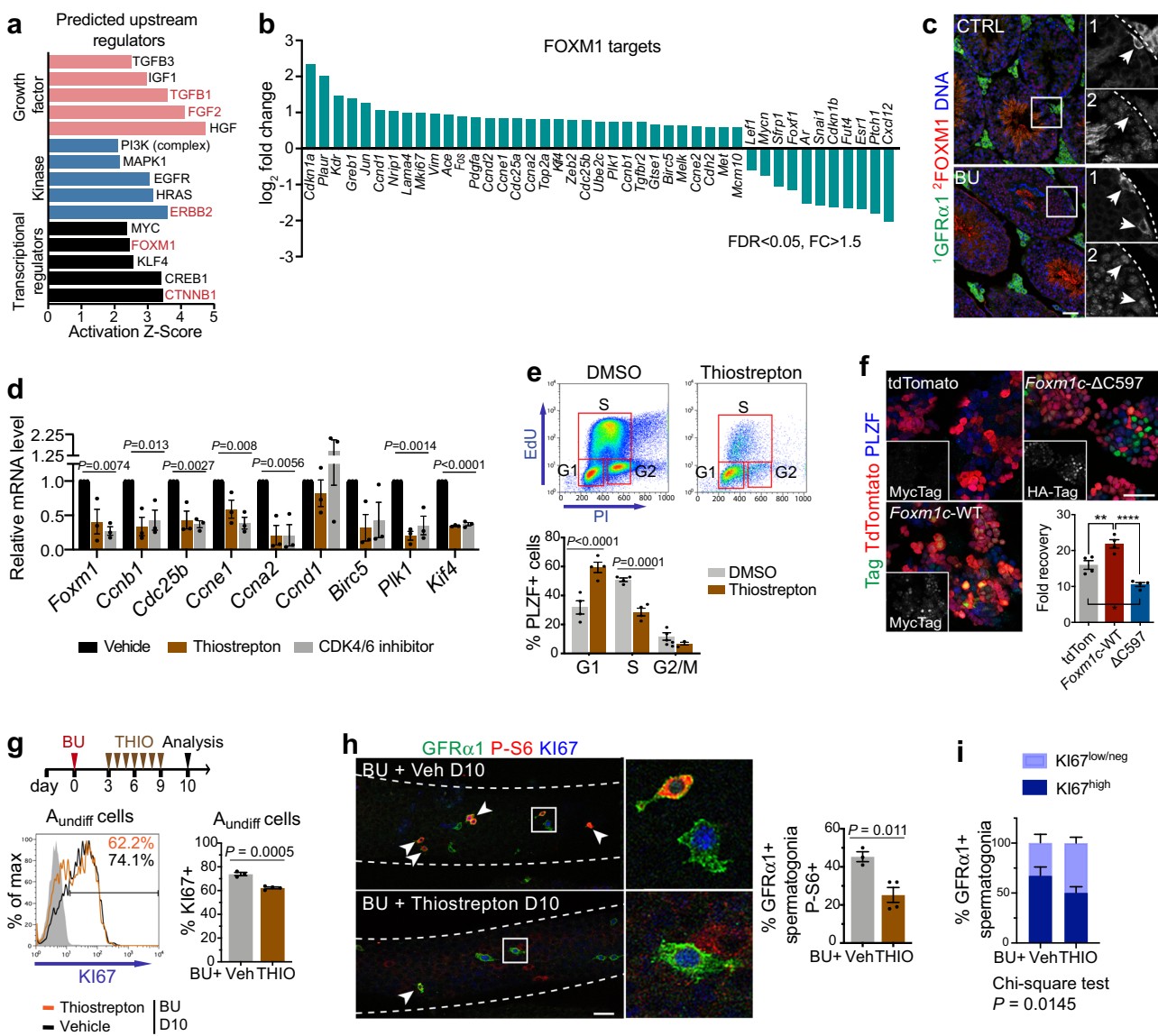

**Fig. 6 Role of transcription factor FOXM1 in the regenerative response. a** Predicted upstream regulator by IPA of DEGs from $A_{undiff}$ of CTRL and D10 BU-treated mice from Fig. 1e. Top 5 significant regulators in red. **b** FOXM1 targets identified from a differentially expressed in BU-treated samples. **c** Representative IF of testis cross-sections from CTRL and D10 post-BU mice ($n = 3$ mice per group). Single channels of indicated regions are shown at higher magnification. Arrows: GFRα1+ spermatogonia. Dashed lines: Basement membrane. **d** RT-qPCR of cultured WT $A_{undiff}$ treated with inhibitors in complete medium for 20 h ($n = 3$ independent cultures). Thiostrepton used at 10 μM. Significance by one-way ANOVA. **e** Cell cycle analysis of WT $A_{undiff}$ cultures by flow cytometry. Cells treated as in **d**. Graph shows percentage of PLZF + cells in cell cycle phases ($n = 4$ independent cultures) Significance by two-way ANOVA ($^{***}P < 0.001$; $^{****}P < 0.0001$). **f** Representative IF of cultured $A_{undiff}$ overexpressing full-length (WT) *Foxm1c*, *Foxm1c*-ΔC597 or tdTomato as control. Graph shows cell recovery at D12 of culture ($n = 4$ cultures). Significance by one-way ANOVA followed by Tukey's multiple comparisons test ($^*P = 0.0114$, $^{**}P = 0.007$, $^{****}P < 0.0001$). **g** Upper panel: treatment regimen and timeline for mice receiving FOXM1 inhibitor thiostrepton (THIO) after BU. Lower panels: representative flow cytometry of testis from D10 mice treated with BU then THIO or vehicle daily for 7 days from D3 post-BU. Graph shows percentage of $A_{undiff}$ KI67 + ($n = 3$ vehicle, $n = 4$ THIO). **h** Representative wholemount IF of tubules from D10 mice treated as in **g**. Higher magnification images of indicated regions are shown. Arrowheads: P-S6 + GFRα1+ spermatogonia. Graph shows percentage of GFRα1+ spermatogonia P-S6 + ($n = 3$ vehicle, $n = 4$ THIO). Dashed lines: tubule outline. **i** Graph shows KI67 intensity within GFRα1+ spermatogonia from **h** grouped into indicated levels using ImageJ (cut-off = 40) ($n = 3$ vehicle, $n = 4$ THIO, >150 cells analysed per sample per group). Significance by two-sided $\chi^2$ test. Scale bars, 50 μm. Data presented as mean values ± SEM in **d**–**i**. Significance by two-tailed unpaired $t$ test in **g**, **h**. Source data are provided as a Source Data file.

of GFRα1+ spermatogonia that were P-S6 + (Fig. 6h), indicating that FOXM1 promotes mTORC1 signalling in regenerative $A_{undiff}$, a feature of the regenerative response. Suppression of mTORC1 by thiostrepton correlated with reduction in percentage of GFRα1+ cells with high KI67 levels although did not affect abundance of GFRα1+ cells (Fig. 6i and Supplementary Fig. 8i). Our data indicate that FOXM1 induction in $A_{undiff}$ following damage is

required for effective initiation of the regenerative response. However, genetic studies will be required to fully characterise the role of FOXM1 in germline regeneration.

**Growth factor-dependent signalling induces FOXM1 in $A_{undiff}$.**
As our data demonstrated roles for growth factor signalling in regeneration, we assessed whether these pathways were involved

in FOXM1 induction in regenerative A$_{undiff}$. Incubation of cultured A$_{undiff}$ with multikinase inhibitors AD80 or ponatinib suppressed activity of PI3K/AKT, ERK MAPK and mTORC1 and resulted in substantial downregulation of FOXM1 plus targets BIRC5 and CCNB1 (Supplementary Fig. 9a–d), linking growth factor signalling to FOXM1 induction. EpCAM levels were also reduced by AD80 and ponatinib, consistent with use as readout of growth factor stimulation (Fig. 5d and Supplementary Fig. 9e). Notably, AD80 treatment suppressed expression of FOXM1 targets uPAR and CCND1 in regenerative A$_{undiff}$ at D10 post-BU (Fig. 5d and Supplementary Fig. 9f)[52,61], indicating that AD80 inhibited FOXM1 activity in vivo. However, effects of AD80 on CCNB1 and FOXM1 expression in regenerative A$_{undiff}$ were variable (Supplementary Fig. 9g and not shown), suggesting that AD80 was more effective at inhibiting FOXM1 in cultured A$_{undiff}$.

As control, treatment of cultured A$_{undiff}$ with a dual PI3K/mTOR inhibitor (apitolisib) to suppress key pathways activated during regeneration (AKT and mTORC1) effectively reduced FOXM1 and target expression while selective inhibition of FGF or IGF receptors had more limited effects on signalling and FOXM1 levels (Supplementary Fig. 9a–d). Analysis of transcript levels confirmed differential ability of inhibitors to repress FOXM1 targets although suggested post-transcriptional regulation of some genes (e.g., *Ccnb1*) (Supplementary Fig. 9h).

We concluded that FOXM1 induction in A$_{undiff}$ requires concerted activation of multiple signalling pathways. Accordingly, inhibition of PI3K, ERK MAPK or mTORC1 alone led to modest reduction in FOXM1 and targets when compared to combined AKT/mTOR inhibition (Supplementary Fig. 9i). While differences may exist in the role of signalling pathways in cultured A$_{undiff}$ versus regenerative A$_{undiff}$ in vivo, our data provide support for involvement of PI3K/AKT and mTORC1 pathways plus downstream factor FOXM1 in induction of the A$_{undiff}$ regenerative state.

## Discussion

Male germline recovery following genotoxic damage is dependent on surviving undifferentiated spermatogonia but whether functional characteristics and molecular features of A$_{undiff}$ mediating germline regeneration are distinct from those of A$_{undiff}$ involved in tissue homoeostasis remain unclear. Through analysis of a mouse model of germline depletion and regeneration we define a unique gene expression signature and distinctive functional state of regenerative A$_{undiff}$ (Fig. 7). Based on our data we propose that changes in the A$_{undiff}$ microenvironment are responsible for induction of a regenerative state. Specifically, that increased growth factor abundance following germline damage activates key signalling pathways, including PI3K/AKT and mTORC1, within surviving A$_{undiff}$ to initiate regeneration. Further, we demonstrate that transcription factor FOXM1 is an important effector downstream growth factor-dependent signalling in regenerative A$_{undiff}$.

Stem cells are defined by their ability to initiate endogenous regeneration following tissue damage[20]. A$_{undiff}$ that survive cytotoxic treatment to restore spermatogenesis ('repopulating stem cells') were observed to share morphological features with A$_s$ 'stem' spermatogonia although were more often mitotic[7,10]. Lineage-tracing data indicated that these regenerative A$_{undiff}$ are marked by *Eomes* and *Pax7* expression[11,12]. Our study dissected molecular features and heterogeneity of A$_{undiff}$ in the regenerating germline and identified markers and regulatory mechanisms of regenerative A$_{undiff}$. Based on gene expression, we documented a shift in predominant fate of A$_{undiff}$ from differentiation-primed to self-renewing during initiation of regeneration. Single cell analysis of A$_{undiff}$ demonstrated that GFRα1+ SSC-enriched populations of homoeostatic and regenerative testis are heterogenous and contain subsets distinguished through expression of cell cycle and other stem cell genes. Relative abundance of these cell subsets changed under regenerative conditions such that proliferative states were more prominent. RNA-Velocity indicated that quiescent 'primitive' SSCs undergo increased transition into both proliferative SSCs and differentiation-destined progenitors during regeneration, supporting a central role for the primitive SSC pool in mediating germline recovery. Our analysis demonstrated that cellular dynamics of the A$_{undiff}$ pool are also altered following damage and revealed the distinct fate choices available to regenerative A$_{undiff}$.

Besides changes in relative abundance of A$_{undiff}$ states during regeneration, we identified genes uniquely expressed or upregulated within regenerative A$_{undiff}$. EpCAM expression was significantly increased in regenerative A$_{undiff}$ and has been suggested to mark "activated" SSCs induced by high levels of GDNF, e.g. upon culture[62]. We demonstrate that EpCAM expression provides a readout of growth factor signalling in A$_{undiff}$, and that growth factor stimulation experienced by A$_{undiff}$ during culture induces regenerative features including inactive/cytosolic FOXO1 (Supplementary Figs. 6b, 9d). Transplant studies indicate that SSCs of undisturbed testis express low EpCAM levels[32]. SSCs may therefore be exposed to limiting growth factor levels within

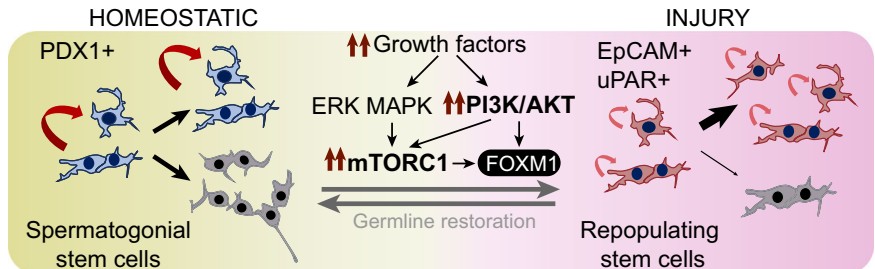

**Fig. 7 Model illustrating distinct states of homoeostatic and regenerative A$_{undiff}$ plus pathways regulating interconversion.** Homoeostatic SSCs are marked by PDX1, undergo balanced self-renewal (curved arrows) and differentiation commitment (grey syncytia) and are optimised for life-long germline maintenance. In response to germ cell loss, increased growth factor stimulation hyperactivates PI3K/AKT and mTORC1 signalling pathways in surviving A$_{undiff}$ and drives adoption of a unique, highly proliferative state marked by EpCAM and uPAR that is biased towards self-renewal (regenerative A$_{undiff}$ or 'repopulating stem cells'). Regenerative A$_{undiff}$ properties are optimised for rapid germline regeneration, but long-term germline maintenance capacity is limited (small, curved arrows). Transcription factor FOXM1 is a downstream effector of growth factor-dependent signalling in A$_{undiff}$ and promotes increased cell cycle progression characteristic of regenerative A$_{undiff}$. Expression of uPAR can underlie an increased migratory capacity. Following recovery of the spermatogonial pool and declining growth factor stimulation, regenerative A$_{undiff}$ transition back to a homoeostatic state to preserve life-long germline integrity.

homoeostatic tissue, in agreement with mitogen competition models that account for SSC abundance[19].

A feature of regenerative $A_{undiff}$ was expression of uPAR (*Plaur*), which has roles in tissue remodelling, signalling, epithelial-mesenchymal transition, and cell migration[43,63,64]. Steady-state tissue has low uPAR expression while levels are elevated during development and upon stress, inflammation, and cancer[43,63,64]. uPAR is expressed at low levels in the adult germline but detected on neonatal undifferentiated spermatogonia with enhanced migration potential, expression of SSC-associated genes and proliferative capacity[44]. Upregulation of uPAR in regenerative $A_{undiff}$ can be reflective of similar functional requirements of stem cells in the developing and regenerating germline. *Plaur* expression in cultured $A_{undiff}$ is reportedly induced by GDNF that promotes $A_{undiff}$ self-renewal via ERK MAPK[17,30]. In turn, uPAR can promote cell migration, proliferation, and survival through ERK MAPK, FAK and c-SRC[43,63,64]. We propose that increased stimulation of surviving $A_{undiff}$ with growth factors such as GDNF following germline damage induces uPAR expression, which promotes proliferation and migration of regenerative $A_{undiff}$ during tissue recovery. Consistently, treatment of mice with AD80, which blocks response of $A_{undiff}$ to GDNF, prevented uPAR induction in regenerative $A_{undiff}$ and inhibited proliferation.

Cellular response to microenvironmental growth factors is mediated by intracellular signalling. Increased production and availability of GDNF and FGFs are suggested to underlie expansion and recovery of spermatogonial populations following germline damage[6,19,21]. Gene expression analysis of $A_{undiff}$ during regeneration supports stimulation by both GDNF and FGFs. These factors promote activation of ERK MAPK and PI3K/AKT pathways in $A_{undiff}$, which stimulate mTORC1 and inactivate FOXO1[1,22]. In vivo analysis confirms hyperactivation of AKT-mTORC1 signalling and FOXO1 inactivation in regenerative $A_{undiff}$ and role of mTORC1 in the regenerative response. While chronic mTORC1 activation is detrimental for stem cell function in many tissues, this growth-promoting pathway is essential for stem cells to contribute to tissue regeneration[25,45,46]. Deletion of the mTORC1 component RAPTOR disrupts regenerative function of haematopoietic stem cells after irradiation[46]. Further, in response to muscle damage and increase in systemic growth factor HGF, quiescent satellite cells become primed for cell cycle and tissue repair through an mTORC1-dependent mechanism[45]. Given detrimental effects of prolonged mTORC1 activation on stem cell self-renewal, the ability to downregulate mTORC1 signalling following tissue restoration is essential. Similarly, while FOXO transcription factors are required for stem cell maintenance in several systems including the male germline[22], FOXO1 became cytosolic and inactive due to AKT activation in regenerative $A_{undiff}$. Increased FOXO1 activity inhibits regeneration of liver and muscle suggesting it might play an equivalent role in the germline[65,66]. Notably, combined inhibition of mTORC1 and re-activation of FOXO1 by treatment with AD80 led to more pronounced inhibition of germline regeneration than inhibition of mTORC1 alone with rapamycin.

In conclusion, while regenerative $A_{undiff}$ exhibit elevated mTORC1 signalling and FOXO1 inactivation, long-term stem cell maintenance requires low mTORC1 activity and FOXO function. This "activated stem cell state" of regenerative $A_{undiff}$ is likely incompatible with life-long germline maintenance but rather suited for short-term expansion and spermatogonial recovery. Once the germline is restored and growth factor levels normalise, mTORC1 and FOXO activity in $A_{undiff}$ will return to levels found in undisturbed conditions. Expression of multiple negative and partially redundant regulators of AKT-mTORC1 signalling in $A_{undiff}$ (e.g., *Pten*, *Tsc1/2*, *Plzf*, *Gilz*, *Nanos2*) might be key for

resumption of an mTORC1-low FOXO-active homoeostatic state[22–25,67].

Regenerative $A_{undiff}$ induced expression of FOXM1, a cell cycle regulator with roles in tissue regeneration, cancer development and senescence[49,52,55]. Treatment of mice with FOXM1 inhibitor thiostrepton following BU inhibited proliferation of surviving $A_{undiff}$ and mTORC1 activation in GFRα1+ cells, core features of the regenerative response. *Foxm1* expression is regulated via RB/E2F1 and an autoregulatory loop while FOXM1 stability, activity and localisation are controlled by CDK and ERK MAPK-dependent phosphorylation[55,57,59,68]. Our data indicated concerted regulation of FOXM1 mRNA and protein in $A_{undiff}$ by PI3K/AKT and mTOR signalling. PI3K/AKT and mTORC1 are linked to FOXM1 induction in other cell types[50,69]. Given that FOXM1 expression and activity are inhibited by FOXO factors[53,70], AKT signalling in regenerative $A_{undiff}$ may promote FOXM1 function via FOXO inactivation. Notably, use of multi-kinase inhibitors led to more efficient FOXM1 downregulation in $A_{undiff}$ than inhibition of single signalling components such as ERK MAPK or PI3K. Increased FOXM1 levels in regenerative $A_{undiff}$ are therefore reflective of combined levels of growth factor stimulation rather than activation of a single signalling pathway. Notably, the negative effect of FOXM1 inhibition on mTORC1 activity in regenerative $A_{undiff}$ suggests a positive feedback loop between FOXM1 and growth factor signalling[71].

Combined, our study provides insight into the molecular characteristics of regenerative $A_{undiff}$ and regulatory pathways that underlie their functional capabilities. Ultimately, this knowledge can allow development of therapeutic approaches that stimulate regenerative capacity of spermatogonia and increase rate and degree of germline recovery following chemotherapy. Given use of targeted therapies for cancer treatment, dissection of effects of clinically relevant signalling inhibitors on activity of regenerative $A_{undiff}$ can also improve our understanding of effects of these drugs on patient fertility and guide selection of treatments that minimise long-term germline damage.

## Methods

**Mouse strains and treatments**. Animal studies were performed in accordance with the Australian Code of Practice for the Care and Use of Animals for Scientific Purposes. Experiments were subject to approval by the Monash University and Medical Centre Animal Ethics Committees (MARP-2015-025 and MMCB-2020-15). Mice were housed at 18–24 °C with 40–70% humidity and a 12-hour day-night cycle. Adult wildtype mice (6–8 weeks old) were of C57BL6 background. BU (Cayman Chemical) was prepared for intraperitoneal (IP) injection as described[72]. Mice were treated with a single dose of BU at 10 mg/kg. BU-treated cohorts were treated daily by IP injection starting at D3 for 7 consecutive days with RAPA (4 mg/kg), thiostrepton (50 mg/kg) or vehicle[24,60]. Mice were treated with AD80 (20 mg/kg) or vehicle by IP injection from D5 after BU for 5 consecutive days as described[73]. For analysis of signalling pathway activity at D10, an additional dose of AD80 or vehicle was administered 3 h before harvest. Oct4-GFP transgenic mice have been described elsewhere[5].

**Flow cytometry**. Single cell suspensions were generated from adult testis by digestion with type II collagenase (Sigma)[24]. Harvested cells were stained for 25 min on ice with antibodies in phosphate-buffered saline (PBS) with 2% foetal bovine serum (FBS) (Supplementary Data 6). Antibody to uPAR was detected with Alexa488-conjugated Donkey anti-Goat antibody (1:500) (Thermo Fisher). DAPI was used for live/dead cell discrimination. Analysis of fixed and permeabilised testis cells for PLZF, c-KIT, KI67 and EOMES is described elsewhere[5]. Cells were sorted at Monash Flowcore with an Influx Cell Sorter (BD Biosciences). An LSR Fortessa X-20 (BD Biosciences) was used for analysis and data processed with FlowJo software.

**RNA-sequencing**. Testes from adult control and BU-treated mice ($n = 4$ per group) were harvested at D10 and $A_{undiff}$ (E-Cadherin+ c-KIT– α6-integrin+) isolated. RNA was extracted using TRIzol LS (Thermo Fisher) and Direct-zol RNA MiniPrep Kits (Zymo Research) including removal of contaminating DNA by in-column DNase I digestion. SPIA amplified cDNA was processed according to Nugen Ovation RNA-Seq system V2 protocol[24]. Libraries were sequenced with an Illumina HiSeq 3000 (100 bp paired end) at the Medical Genomics Facility,

Monash Health Translation Precinct (MHTP). Data were processed by the Monash Bioinformatics Platform using the RNAsik pipeline and a raw counts file uploaded to Degust Webapp, which uses limma-voom for statistical analysis (http://degust.erc.monash.edu). Cut-off for DEG was false discovery rate < 0.05 and fold change > 1.5. DEGs were classified using the DAVID Bioinformatics Resource[74] and Ingenuity Pathway Analysis (Qiagen). DEGs were compared to unique, validated GDNF-regulated genes identified from microarray analysis of cultured $A_{undiff}$[30].

**Single-cell RNA-sequencing and data analysis.** $A_{undiff}$ (E-Cadherin+ c-KIT− α6-integrin+) were isolated from control and BU-treated adult mice at D10 as above. Pooled cells (3 mice per group, 2 independent experimental repeats) were resuspended at 200 viable cells/µl in PBS with 2% FBS. A Chromium Single Cell 3′ Reagent Kit V2 and 10X Chromium controller were used for library construction and libraries from each experiment were sequenced in one high-output lane of an Illumina NextSeq in single-read 150b format. Data were processed using Cell Ranger software (10x Genomics, Inc., Cellranger count v3.0.2) aligned to mm10 genome (refdata-cellranger-mm10–1.2.0) and matrix files were used for subsequent bioinformatics analysis as detailed[5]. Sequencing metrics from the Cell Ranger pipeline are included in Supplementary Data 7.

*Dimensionality reduction and batch-effect correction.* Matrix files from Cell Ranger output were analysed using Seurat v3.2. Seurat objects of Control, BU-treated groups, and ID4-GFP-bright cells (GSE109049) were built by loading matrix files using the 'Read10X' function. Each dataset was filtered and cells with greater than 1000 genes expressed and less than 20% of reads mapped to the mitochondrial genome were retained. Quality control metrics for each experimental repeat are shown in Supplementary Fig. 10a. After these filtering steps, the datasets were normalised using the 'LogNormalize' function and scaled using a scale factor of 100,000. After normalisation, the top 2000 highly variable genes were selected using the 'FindVariableFeatures' function. To account for variations among different batches, the datasets were aligned using canonical correlation analysis (CCA) with 'FindIntegrationAnchors' and 'IntegrateData' functions. Dimensionality of data was reduced by principal component analysis (PCA) (30 components) and visualised with UMAP (Supplementary Fig. 10b).

*Cell cluster identification, annotation and DEG analysis.* Clustering was performed using the Louvain algorithm on 30 principal components. After initial clustering, small clusters representing contaminating somatic cells (*Cd74/Wt1/Aldh1a1+*), spermatids (*Spem+*) and one cluster with high *Kit* expression (differentiating spermatogonia), were excluded from subsequent analysis (Supplementary Fig. 10c, d). Cell numbers from initial clustering are listed in Supplementary Data 7. After splitting the Seurat object, we repeated normalisation, variable gene identification, CCA integration, dimension reduction and cluster identification as described above. Cell clustering and distribution of cells between clusters were comparable between the different experimental repeats (Supplementary Fig. 10e, f). Cluster-specific differentially expressed genes (DEGs) were calculated with the 'FindAllMarkers' function with MAST using default parameters and clusters were assigned based on known stem cell and progenitor markers. Total cell numbers in the respective clusters were as follows: (0) Primitive SSCs - CTRL 983, BU 674; (1) Progenitors - CTRL 1309, BU 234; (2) Transitional - CTRL 517, BU 576; (3) Undefined - CTRL 570, BU 393; (4) Proliferative SSC1 - CTRL 301, BU 619; (5) Proliferative SSC2 - CTRL 118, BU 172. Enriched ontology terms for differentially expressed genes were identified using ClusterProfiler. Differential gene expression analysis on the scRNA-seq datasets among Control, BU-treated groups, and neonatal Id4-GFP-bright dataset (GSE109049) was performed via 'FindMarkers' using MAST with default parameters on all expressed genes. The Seurat FindMarkers function was used to identify DEGs between cells of the same SSC clusters from control and BU-treated samples.

*RNA velocity analysis.* We employed scVelo to infer future states of individual cells using the spliced and unspliced information. The aligned bam file generated by Cell Ranger was recounted with the Velocyto counting pipeline velocyto.py in python. The sample-wise counts of unspliced and spliced reads in loom format were loaded to scVelo. Samples of the same group were combined. Genes with less than 20 spliced and unspliced counts in a cell were filtered and the counts were normalised using normalize_per_cell(). 2000 high variability genes were identified and retained by filter_genes_dispersion(), following which the counts were log-transformed using log1p(). The first and second order moments for each cell across its nearest neighbours were calculated using scvelo.pp.moments(). We used recover_dynamics() to define the splicing kinetics of expressed genes. Subsequently, the velocities were estimated using the scvelo.tl.velocity() with the mode set to 'dynamical' and the velocity graph constructed using scvelo.tl.velocity_graph() function. Velocities were visualised on top of the previously calculated UMAP coordinates obtained from Seurat.

*Gene regulatory network analysis.* SCENIC was employed to identify regulons controlling gene expression. The python implemented SCENIC was run on the raw count matrix combining all samples using GRNboost2 method for gene network

reconstruction. The cisTarget motif dataset (mm9-500bp-upstream-7species.mc9nr.feather, mm9-tss-centered-10kb-7species.mc9nr.feather) was used to construct regulons for each transcription factor. Cellular enrichment of the regulons was subsequently assessed by AUCell. Regulon specificity scores were ranked based on Jensen-Shannon divergence following the SCENIC pipeline and the top 10 regulons of each cluster were visualised on top of Seurat UMAP embedding.

*Packages used.* R v3.6; python 3.7.3; cellRanger v3.1; velocyto v0.17.17; R packages (Seurat v3.2.1; SCENIC v1.1.3; clusterProfiler v3.14.3; tidyverse v1.3.0; AUCell v1.8.0; MAST v1.12.0; Augur v1.0.0); python packages (scvelo 0.2.2; scanpy 1.6.0; scikit-learn v0.23.2; scipy v1.5.1); custom R and python scripts. For the bulk RNA-seq analysis, R packages include limma v3.40.6, edgeR v3.26.8, jsonlite v1.7.2, ggplot2 v3.3.5 and ComplexHeatmap v2.2.0.

**Immunofluorescence.** Mouse testes were fixed with 4% paraformaldehyde (PFA) in PBS overnight at 4 °C, cryoprotected with 30% sucrose in PBS, embedded in OCT compound (Tissue-Tek) and cut into 8 µm sections. Sections were blocked in 2% bovine serum albumin (BSA) (Sigma) and 10% FBS (GE Healthcare) in PBS then incubated overnight with primary antibodies diluted in blocking solution. Slides were washed in PBS before incubation with secondary antibodies plus DAPI nuclear counterstain. For wholemount IF, testes were detunicated and seminiferous tubules teased apart and rinsed in ice-cold PBS. Tubules were fixed with 4% PFA for 5 h at 4 °C and washed in PBS prior to blocking in 0.3% Triton X-100 in PBS (PBSX) supplemented with 10% FBS and 2% BSA. Tubules were incubated with primary antibodies in PBSX containing 1% BSA overnight at 4 °C. Samples were washed in PBSX and primary antibodies detected with Alexa Fluor-conjugated secondary antibodies (ThermoFisher Scientific). Sections and tubules were mounted in Vectashield mounting medium (Vector Labs). Primary antibodies are included in Supplementary Data 6. Image analysis was performed with Zeiss LSM780 FCS and Nikon C1 confocal microscopes and an Olympus Stereologer system at the Monash Micro Imaging facilities within Monash University and the Monash Health Translation Precinct.

**Quantitative RT-PCR.** Isolated $A_{undiff}$ and cultured spermatogonia were lysed in TRIzol LS reagent (Thermo Fisher Scientific), RNA was purified, and DNase treated using a Direct-zol RNA Miniprep kit (Zymo Research). A Tetro cDNA synthesis kit (Bioline) was used for cDNA synthesis and quantitative PCRs were run on Mic qPCR Cycler (Bio Molecular Systems) using Takara Sybr Premix Ex Taq II (Clontech). Primer sequences are in Supplementary Table 1.

**Cell culture.** Undifferentiated spermatogonia were cultured on mitomycin-inactivated mouse embryonic fibroblasts (MEF) in StemPro-34 media (Thermo Fisher) with 10 ng/ml GDNF, 10 ng/ml bFGF, 20 ng/ml EGF, 25 µg/ml insulin and other additives[5,72]. To establish cultures, $A_{undiff}$ were enriched from adult WT testis single-cell suspensions using biotinylated anti-CD9 antibody clone MZ3 (Biolegend, 1:400) and EasySep biotin selection kits (Stem Cell Technologies). To assess effects of inhibitors on cultured $A_{undiff}$, cells were harvested using trypsin/EDTA and feeder cells depleted by plating on tissue culture plates for 2 h in culture media. Non-adherent spermatogonia were removed from adherent MEFs and plated onto 12-well plates coated with Geltrex (Thermo Fisher) at $5 \times 10^5$ cells per well and treated with inhibitors 2–3 days later. Cell dissociation buffer (Thermo Fisher) was used to prepare cell suspensions for flow cytometry. Inhibitors (Selleckchem) were dissolved in DMSO or water (palbociclib) and diluted in media to the following concentrations: AD80, ponatinib, AZD4547, linsitinib, apitolisib, alpelisib and buparlisib 1 µM, PD0325901 and palbociclib 5 µM, torin1 0.5 µM and rapamycin 20 nM. FOXM1 inhibitors thiostrepton (Tocris) and FDI-6 (Merck) were prepared in DMSO and used at the indicated concentrations. To assess effects of growth factors on signalling in cultured $A_{undiff}$, cells were placed in growth-factor reduced medium (lacking GDNF, bFGF, EGF and insulin) for 20 h prior to stimulation[24]. Inhibitor was added to cells 30 min before addition of growth factors at concentrations in complete media. Inhibitor targets and associated $IC_{50}$ values are detailed in Supplementary Data 6.

**Cloning of *Foxm1c* and lentiviral transduction.** Constructs were generated using mouse *Foxm1c* coding sequence (NM_008021.4) synthesised as gBlock gene fragment with 5′ Kozak sequence and 3′ Myc tag (Integrated DNA Technologies). *Foxm1c* constructs were subcloned into pCCL-hPGK vector by PCR[24]. The dominant-negative *Foxm1c* construct was based on previous studies[51,75] and generates an HA-tagged protein with C-terminal truncation from amino acid 597. Lentivirus production and collection was performed using HEK293FT cells. Cultured $A_{undiff}$ were infected with lentiviral-containing supernatant supplemented with 8 µg/ml polybrene (Millipore). Cells were labelled by co-infection with pCCL-hPGK-tdTomato lentivirus (1:4 volume ratio of tdTomato:Foxm1c supernatant). Infected cells were selected by FACS and re-plated for analysis. Cells infected with pCCL-hPGK-tdTomato were used as controls.

**Western blotting.** Western blotting was performed according to standard methods[24]. To prepare lysates, feeder depleted cultured $A_{undiff}$ on Geltrex plates

were washed in PBS and then scraped into RIPA buffer containing protease inhibitors and PhosSTOP (Roche). Primary antibodies are included in Supplementary Data 6[23]. Band intensity was quantified using ImageJ.

**Statistical analysis**. Assessment of statistical significance was performed using two-tailed unpaired $t$ tests, one-way ANOVA with Tukey multiple comparisons tests or Chi-squared tests. Statistical analysis was performed using GraphPad Prism v8. Associated $P$ values are indicated as follows: *$P < 0.05$; **$P < 0.01$; ***$P < 0.001$; ****$P < 0.0001$; not significant $P > 0.05$. No statistical method was used to predetermine sample sizes and no specific randomisation or blinding methods were used.

**Reporting summary**. Further information on research design is available in the Nature Research Reporting Summary linked to this article.

## Data availability
The bulk and single cell RNA-Seq data generated in this study have been deposited in the Gene Expression Omnibus database under accession codes GSE182727 and GSE182924, respectively. The published single cell RNA-Seq dataset of neonatal A$_{undiff}$ used in this study is available under accession code GSE109049[36]. Source data are provided with this paper.

## Code availability
Degust code for analysis of bulk RNA-Seq data is provided in a Zenodo repository (https://doi.org/10.5281/zenodo.3258932). Seurat for scRNA-Seq analysis is available from GitHub (https://github.com/satijalab/seurat/). SCENIC and scVelo are available from https://scenic.aertslab.org/ and https://scvelo.org respectively.

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

## Acknowledgements

We thank the Monash Animal Research Platform, Micromon, Monash FlowCore, Monash Bioinformatics Platform, Monash Micro Imaging and Monash Health Translation Precinct Medical (MHTP) Genomics, Microscopy, Histology and Flow Cytometry Platforms for technical support. This study was supported by NHMRC Project Grant APP1164019 to R.M.H. H.M.L. was supported by an Australian Government Research Training Program (RTP) Scholarship and A.P. by a Victorian Cancer Agency (VCA) Mid-Career Research Fellowship (MCRF20027). The Australian Regenerative Medicine Institute is supported by grants from the State Government of Victoria and Australian Government.

## Author contributions

H.M.L., J.L., T.L.L. and R.M.H. conceived and designed the study. H.M.L., J.L., J.M.D.L., A.-L.C., V.V., J.E.C. and R.M.H. performed experiments. H.M.L., J.L., J.M.D.L., F.J.R., A.P., T.L.L. and R.M.H. analysed and interpreted data. H.M.L., J.L., J.M.D.L., A.P., T.L.L. and R.M.H. wrote the paper.

## Competing interests

The authors declare no competing interests.
