## [Peer Review File · Nature Communications]

Distinctive molecular features of regenerative stem cells in the damaged male germlineReviewers' comments:

Reviewer #1 (Remarks to the Author):

Using mice as a model, Hue et al. report on dynamic changes undergone by undifferentiated spermatogonia (SPG) during germ cell regeneration. First, they verified earlier work by finding that a low dose of the alkylating agent, BU, efficiently depletes most SPG. Second, they identified a subset of highly undifferentiated SPG (Aundiff) that initiates a regenerative response 8 to 10 days after BU treatment. Third, RNAseq analysis of cells enriched for Aundiff (E-Cadherin+ α 6-integrin+ c-KIT-) at day 10 after BU treatment revealed a pronounced shift in gene expression. Comparison of this "regeneration RNAseq dataset" with a published steady-state RNAseq dataset revealed that the niche factor genes, *Gdnf* and *Fgf*, are increased in expression during the initial stage of regeneration. Fourth, to investigate the precise cell populations responsible, the authors performed single cell (sc) RNAseq analysis. This analysis suggested that spermatogonial stem cell (SSCs) are largely resistant to BU's toxic effects and that these SSCs, along with surviving progenitor cells, transition into a new cell state with rapidly self-renewing properties to drive regeneration. Fifth, to investigate mechanism, the authors followed up on their circumstantial evidence (described above) that GDNF signaling is involved in Aundiff regeneration. In support of a role for GDNF signaling, they found that AKT activation is increased during Aundiff regeneration. Sixth, since mTORC1 is a downstream effector, they tested the role of this pathway using the mTORC1 inhibitor, rapamycin, which indicated that indeed mTORC1 activation plays a role in the Aundiff regenerative response (but temporary mTORC1 inhibition did not compromise Aundiff regenerative capacity in the long-term). Seventh, they used conditional KO mice to test the role of a transcription factor, LHX1, which is upregulated during regeneration and is involved in SSC maintenance. This analysis showed that the *Lhx1* gene is dispensable for Aundiff regeneration. Finally, the authors found that, in contrast, another regeneration-regulated transcription factor gene—*Foxm1*—DOES have a role in Aundiff regeneration. This gene, which encodes a cell-cycle factor regulated by both the PI3K/AKT and mTORC1 pathways, promoted the proliferation of cells during the regeneration response.

This manuscript is potentially highly significant, as it provides evidence that growth factor-dependent signaling pathways—including the PI3K/AKT, mTORC1 and FOXM1 pathways—are likely important for SSC regeneration. In general, the experiments are well designed. However, there are also several concerns that must be addressed:

Concerns:

- 1) Organization and writing. This manuscript was difficult to read. In part, this may be unavoidable given journal space constraints, but nonetheless some improvements are needed. For example, some sections are so poorly introduced that it often took quite a while to absorb why the new set of experiments were being done. As just one example, the first few sentences introducing the section starting on line 327 are overly complicated and do not flow well together. It is suggested to instead state in the very first sentence what the broad question is and how it will be addressed; e.g., "To empirically assess the role of signaling during regeneration, we...". This is just one of many examples; all section introductions should be carefully inspected for this issue. Indeed, the entire manuscript should be inspected for improvements in clarity. While the sentences are typically well written, the flow of information from one sentence to another (or over a paragraph) is often hard to comprehend.
- 2) LHX1. Given that this factor was shown to NOT have a role in regeneration and this data interrupts the intellectual flow of the manuscript, it is suggested to omit this section. It could be written up as a separate short independent story elsewhere. If authors insist on keeping this section, consider moving to another position, such as at the end of this Results section, as a contrast with the *Foxm1* results.
- 3) scRNAseq analysis. Only 1 biological replicate was analyzed per genotype, which does not account for variability (even though the 1 replicate was a pool of 3 samples). This is a serious concern - at least 2 biological replicates should be analyzed by scRNAseq.
- 4) Fig. 2a. The legend indicates that *Ddx4*+ cells were selected for the analysis shown. Does this mean that the authors' purified cell populations containing other cell types? This crucial information must be added to the MS. It also pertains to the experiments in which BU-mediated regeneration is examined.
- 5) QC parameters. The authors should provide the quality matrix for the scRNAseq data, the normalization approaches used, and the cell numbers in each cluster.

- 6) Line 240. Why "As anticipated,..."? In general, not a good idea to use phrases like this and "as expected." Instead, state a hypothesis or give result and state why it fits a prediction.
- 7) Line 437. This conclusion is overstated. Is it not possible that the mTORC1 inhibitor, rapamycin, suppresses the increase of KI67+ cells by increasing the number of proliferating cells that undergo apoptosis?
- 8) Fig. 6h-i. This data showed that inhibitors could efficiently inhibit their known signaling pathway targets. All the inhibitors tested also reduce EPCAM+ cells. However, given that GDNF and FGF signaling pathways are well known to be important for Aundiff in vitro maintenance/culture, this mainly constitutes a verification experiment. To test the specific roles of AKT and mTORC1 during regeneration, the authors should also use AKT- and mTORC1-specific inhibitors and/or activators. The authors should also add the caveat that their in vitro culture system might not recapitulate signaling pathways used during regeneration.
- 9) In vivo relevance of in vitro experiments? The authors tested the role of signaling pathways in regeneration using inhibitors with an in vitro system. It is not clear whether the effects they observed are merely due to the known role of these signaling pathways (e.g., GDNF, AKT, and mTORC1) in steady-state spermatogenesis or whether the effects also pertain to regeneration.
- 10) Somatic cells. The authors often discuss growth factor-dependent signaling pathways involving secreted factors from somatic cells, but they do not experimentally address the effect of their BU/regeneration protocol on somatic cells. Some somatic cells may have been present in their enriched samples used for their scRNAseq analysis, which would allow the authors to report on the molecular characteristics of these somatic cell populations. At the least, the authors could comment on the effect of their BU/regeneration protocol on Sertoli cells, Leydig cells, and PTMs, based on standard histological analysis of testes sections (e.g., morphology, cell number, and cell death), based either their own data or data from the literature.
- 11) Other physiological factors. Likewise, information on regeneration-induced alterations in other factors, such as hormone levels, would enrich the authors' story.

Reviewer #2 (Remarks to the Author):

In this study the authors have made an important investigation toward a fuller understanding of regeneration process by spermatogonial stem cells (SSCs) after insult. In particular, they have revealed the molecular signatures of residual SSCs 10 days after busulfan treatment (BU), when regeneration is initiating, by bulk and single cell RNA-seq analyses of FACS-sorted undifferentiated spermatogonia population. They figured out 6 clusters of spermatogonia, whose balance is tilted toward more primitive states after BU, and revealed transcriptional evidence suggesting that, beside similar molecular signatures, the regenerative SC's are mitotically active due to increased growth factor signaling. They also found by in vitro and in vivo studies that mTOR pathways is activated transiently after BU. They further show the possible involvement of FOXM1 as a potential key regulator of activated cell cycle on regeneration. Overall the manuscript is well written in an evidence-based fashion and the conclusion is well supported by the data. I would point some issues that would further improve this informative and substantial study.

Comments:

1. Throughout, they frequently use the term "growth factors" that stimulate the mitotically active state of residual SSCs after insult. Although this is quite understandable, the described particular factors are not consistent- in many cases only GDNF is raised while in other parts GDNF plus FGFs are listed. In few cases and, importantly in summary scheme in Fig. 6g, Insulin is also listed. While it looks like that these factors work together to activate the mitotic state of SSCs after BU, they should also play distinct (or non-redundant) roles. So presumptive common and distinctive roles of these growth factor warrant more substantial statements throughout the manuscript.
2. In particular, based on the studies in homeostasis, the roles of insulin on SSCs have not been described compared with GDNF or FGF. For example, given the general knowledge it might also be a possibility that insulin supports differentiation-primed (and actively dividing) state by (moderately?) activating Foxo pathway. Insulin appears, generally, to be also more relevant to mTOR and Foxo pathways. So, it would be interesting, in regeneration, if Insulin act on SSCs toward the same direction with GDNF or FGF (maintenance of or transition to primitive states). If data-based discussion about this issue is not feasible, literature based discussion would

significantly increase the insight about this interesting issue.

3. Similarly, as the authors have stated, in many instances, the results of in vitro study may not be consistently applied to in vivo findings (in particular the roles of AKT). This is interesting and suggestive, warranting the interpretations by the authors.

4. The manuscript may be lengthy with too much information; some results may not be directly related to the conclusion or are not even mentioned at all in discussion. Such results are suggested to be deleted or largely trimmed so that the main conclusion can be stressed in a clearer manner. Such "branches" include 1) the cycloheximide treatment, 2) the roles of LHX1, and 3) repeated BU treatment.

5. Regarding the discussions based on the finding that, after BU, primitive states become more dominant while differentiation-tilted states shrink, in addition to their resistance against BU, the author is suggested to also place more emphasize on the mechanism of reversion. Reversion has been shown experimentally by classic lineage tracing study of Neurog3+ cells following busulfan treatment (Nakagawa et al., 2007, 2010). In Result (L226) and in Discussion paragraph starting from line 603, interpretations may better to be made in relation to these preceding studies. As is also the case for interpretation of the result of Velocity analysis in L617.

6. The result of repeated BU is confusing. While the EpCAM expression is diminished, the cellular response including the recovery to the homeostasis density appear not to be affected. This is reflected by the summary sentence of the related Result paragraph starting from L491. More confusingly, in Discussion section L664, the authors describe that "repeated gemline depletion blunted Aundiff regenerative responses", which I cannot agree with given the apparently unaffected cellular responses (despite the impaired EpCAM upregulation). I would advise the authors to provide more detailed explanations to this matter. Alternatively, given that this is not relevant to the main conclusion of this paper, related description could be deleted.

7. A minor comment. In L74, "committed" is too strong in this context since they are evidenced to be reversible to self-renewing state, and I would suggest "differentiation-destined progenitors". Otherwise, the authors may use other appropriate term here.

Reviewer #3 (Remarks to the Author):

The authors developed and characterized a model for spermatogonial stem cell regeneration based on chemotherapy. They define a unique regenerative state in repopulating cells, which is proposed to be driven by growth factors for the niche. They test the function of several potentially relevant genes that mediate regeneration. Transient signaling through mTORC1 was important for initiating regeneration. They found Lhx1 was dispensable for both normal spermatogenesis and regeneration. Foxm1 was identified as novel transcription factor driving the cycle response. I enjoyed reading this interesting manuscript and find that the methods are sound and the conclusions are generally justified by the data. It extends their previous finds nicely. It is likely to influence future studies of spermatogonial stem cells but would also be of interest to the general stem cell field. It could be strengthened a bit by some additional experiments and some editing.

Major points:

1. Many of the conclusions regarding the importance and function of Foxm1 are based on SSC culture (which I find convincing but slightly limited). However, transplantation analysis of Foxm1 mutant cells or equivalent genetic data would greatly strengthen these conclusions. Alternatively, can the FOXM1 inhibitor be used in vivo?
2. Although the replicates seem very similar, sample sizes for some experiments are a bit small, eg, Fig 6h and i, "n = 2 cultures, 2 independent experiments," What do the error bars on the graphs mean? It is not appropriate to display error bars for n=2.

Minor points:

1. Page 21, line 481, "similarly low numbers of E-cadherin+ cells". The Ecadherin flow data showing no difference is not convincing given the sample size. I would consider eliminating Supp fig 7.
2. The labeling of columns in the supp tables (Eg, supp Table 1) is not clear. I can't tell what they

all mean.

3. Fig 6: They should provide supp figures showing expanded images of FOXM1 staining and negative controls for primary antibody for Foxm1.
4. It would be helpful to provide a table with a list of the inhibitors, concentrations used and reported IC50 values for the specific targets (from the literature). Otherwise it is difficult for the reader to interpret the data without looking up the background literature for each inhibitor. The labeling of Supp Fig 8j is confusing because the names of the inhibitors are not given, except for torin, so one has to go to the methods section.
5. Page 10, DEGs in response to GDNF: it is not clear where the 189 in "54 out of 189" comes from or the 71 in 28 out of 71
6. Page 13: " Functional relevance of cells in cluster 3 was unclear and so were termed undefined Aundiff." Can the authors speculate at all? Are they even Aundiff or contaminants?
7. Page 22: Why doesn't FDI-6 reduce Foxm1 and targets?
8. Supp Fig 6b and legend – labeling/description regarding Epcam data is confusing
9. There are a number of minor word omissions/ typos (eg, Fig 6f graph axis).

RESPONSE TO THE REVIEWERS

We would like to thank all three reviewers for their constructive and positive comments that have allowed us to substantially improve the manuscript. As detailed below, we have addressed all concerns by generating significant new datasets, refining our previous analysis, and improving text and organisation of the manuscript. The revised manuscript now includes the following: (1) A more detailed and insightful analysis of single cell (sc) RNA-Seq data from regenerative undifferentiated spermatogonia that incorporates additional experimental replicates (**Fig. 2, 3** and **Supplementary Fig. 2-4**); (2) Direct *in vivo* evidence for the role of FOXM1 in germline regeneration (**Fig. 6g-i**); (3) *In vivo* demonstration of the importance of concerted activation of growth factor signalling pathways for function of regenerative/repopulating stem cells (RSCs) (**Fig. 7d-i** and **Supplementary Fig. 10**); (4) All additional experimental repeats and controls as requested. Our revised manuscript now presents a much more compelling body of work in support of our model of germline regeneration.

We apologise for the delay in resubmitting the manuscript, which was due to extended institute shutdowns last year caused by COVID-19 outbreaks and the demanding nature of requested *in vivo* analyses. However, we wanted to delay resubmission until we could provide a comprehensive response to all reviewer concerns. Point-by-point responses to each comment are provided below.

Changes to the text and citations to updated figure panels are indicated with yellow highlighting.

REVIEWER #1

Using mice as a model, Hue et al. report on dynamic changes undergone by undifferentiated spermatogonia (SPG) during germ cell regeneration. First, they verified earlier work by finding that a low dose of the alkylating agent, BU, efficiently depletes most SPG. Second, they identified a subset of highly undifferentiated SPG (Aundiff) that initiates a regenerative response 8 to 10 days after BU treatment. Third, RNAseq analysis of cells enriched for Aundiff (E-Cadherin+ a6-integrin+ c-KIT-) at day 10 after BU treatment revealed a pronounced shift in gene expression. Comparison of this "regeneration RNAseq dataset" with a published steady-state RNAseq dataset revealed that the niche factor genes, Gdnf and Fgf, are increased in expression during the initial stage of regeneration. Fourth, to investigate the precise cell populations responsible, the authors performed single cell (sc) RNAseq analysis. This analysis suggested that spermatogonial stem cell (SSCs) are largely resistant to BU's toxic effects and that these SSCs, along with surviving progenitor cells, transition into a new cell state with rapidly self-renewing properties to drive regeneration. Fifth, to investigate mechanism, the authors followed up on their circumstantial evidence (described above) that GDNF signaling is involved in Aundiff regeneration. In support of a role for GDNF signaling, they found that AKT activation is increased during Aundiff regeneration. Sixth, since mTORC1 is a downstream effector, they tested the role of this pathway using the mTORC1 inhibitor, rapamycin, which indicated that

indeed mTORC1 activation plays a role in the Aundiff regenerative response (but temporary mTORC1 inhibition did not compromise Aundiff regenerative capacity in the long-term). Seventh, they used conditional KO mice to test the role of a transcription factor, LHX1, which is upregulated during regeneration and is involved in SSC maintenance. This analysis showed that the Lhx1 gene is dispensable for Aundiff regeneration. Finally, the authors found that, in contrast, another regeneration-regulated transcription factor gene—Foxm1—DOES have a role in Aundiff regeneration. This gene, which encodes a cell-cycle factor regulated by both the PI3K/AKT and mTORC1 pathways, promoted the proliferation of cells during the regeneration response.

This manuscript is potentially highly significant, as it provides evidence that growth factor-dependent signaling pathways—including the PI3K/AKT, mTORC1 and FOXM1 pathways—are likely important for SSC regeneration. In general, the experiments are well designed. However, there are also several concerns that must be addressed:

We thank the reviewer for the positive comments and constructive suggestions that have allowed us to substantially improve our manuscript. Responses to specific concerns are detailed below:

Concerns:

1) Organization and writing. This manuscript was difficult to read. In part, this may be unavoidable given journal space constraints, but nonetheless some improvements are needed. For example, some sections are so poorly introduced that it often took quite a while to absorb why the new set of experiments were being done. As just one example, the first few sentences introducing the section starting on line 327 are overly complicated and do not flow well together. It is suggested to instead state in the very first sentence what the broad question is and how it will be addressed; e.g., “To empirically assess the role of signaling during regeneration, we....”. This is just one of many examples; all section introductions should be carefully inspected for this issue. Indeed, the entire manuscript should be inspected for improvements in clarity. While the sentences are typically well written, the flow of information from one sentence to another (or over a paragraph) is often hard to comprehend.

We apologise for the issues with manuscript clarity. We have now carefully reviewed and modified the text in multiple places to improve flow of the manuscript.

2) LHX1. Given that this factor was shown to NOT have a role in regeneration and this data interrupts the intellectual flow of the manuscript, it is suggested to omit this section. It could be written up as a

separate short independent story elsewhere. If authors insist on keeping this section, consider moving to another position, such as at the end of this Results section, as a contrast with the Foxm1 results.

In line with the reviewers' comment and to improve flow of the manuscript, we have now moved the LHX1 knockout section to after the FOXM1 results (**Supplementary Fig. 9**). Given the striking induction of *Lhx1* we observe in regenerative A_{undiff} and the published *in vitro* data supporting a key role for this transcription factor in SSC self-renewal (Oatley et al., 2007), we feel it is important to highlight redundancy of this gene in maintenance and regeneration of the germline.

3) scRNAseq analysis. Only 1 biological replicate was analyzed per genotype, which does not account for variability (even though the 1 replicate was a pool of 3 samples). This is a serious concern - at least 2 biological replicates should be analyzed by scRNAseq.

To better account for sample variability and improve significance of our single-cell RNA-Seq analysis we have repeated this analysis and now include 2 independent sets of experimental replicates (each replicate is pooled from 3 animals). After quality control steps, we now analyse a total of 3798 control and 2669 busulfan (BU)-treated A_{undiff} cells. Clustering analysis of individual samples illustrates the consistencies between experimental repeats (**Supplementary Fig. 12e, f**). Following addition of experimental repeats, we have completely revised our scRNA-Seq analysis using updated algorithms and the combined data from both experimental repeats (**Fig. 2, 3 and Supplementary Fig. 2-4**). This revised analysis has allowed us to provide more substantial insight into A_{undiff} heterogeneity and dynamics under steady state and regenerative conditions. The accompanying text has been changed to reflect the new analysis and insights obtained (**lines 225-371**).

While cell clustering is largely comparable to our original analysis, we now define a single quiescent "primitive" SSC state and two distinct proliferative SSC states (**Fig. 3a, b**). Consistent with our previous analysis, we also identify a predicted stem-progenitor transitional state, an undifferentiated state of unknown significance and the progenitor population (**Fig. 3a, b**). We now include a figure panel showing the top differentially expressed genes (DEGs) in each cluster (**Supplementary Fig. 3c**) and GO analysis of DEGs highlighting cellular pathways that distinguish the cell subsets (**Fig. 3c**). Interestingly, GO analysis reveals the importance of signalling pathway control in the primitive SSC state and RNA metabolism (processing and splicing) for the proliferative SSC states (**Fig. 3c**). DEG analysis also revealed known and novel marker genes of the distinct cell subsets (**Supplementary Fig. 3c**). Other analyses based on the scRNA-Seq data have been updated to take advantage of the larger dataset, but conclusions remain essentially unchanged (e.g., **Fig. 2**).

Importantly, we have revised our RNA velocity analysis using the scVelo algorithm (**Fig. 3f** and **lines 350-371**), which avoids assumptions on splicing rates incorporated into the original velocity algorithm and is more appropriate for systems with transient cell states, such as the dynamic A_{undiff} pool (Bergen et al., 2020; La et al., 2018b). While this analysis indicated that the A_{undiff} population was relative static in homeostatic tissue, long velocity streamlines between cell clusters suggested rapid and dynamic interconversion of cell states during regeneration (**Fig. 3f**). In particular, the velocity-inferred directionality indicated that quiescent “primitive” SSCs (cluster 0) actively transition into both proliferative SSC and progenitor states (clusters 4 and 1 respectively) under regenerative conditions (**Fig. 3f**). These results highlight the distinct fate options available to SSCs, i.e., self-renewal and differentiation-priming, and suggest that SSCs do not have to enter a proliferative state prior to forming progenitors. The scVelo algorithm also allows confirmation of cell cycle status of different clusters and predicted putative driver genes underlying cell transitions (**Supplementary Fig. 4a, b**). Notably, known regulators of spermatogonial differentiation (FOXO1 and DNMT3B) were identified as drivers of the primitive SSC to progenitor state, and specific cell cycle genes were predicted to underlie the primitive to proliferative SSC transition (**Supplementary Fig. 4b**) (Goertz et al., 2011; Shirakawa et al., 2013). Our revised velocity analysis of the A_{undiff} pool under regenerative conditions provides new insight into the dynamics and biology of SSCs.

Finally, to characterise potential regulators of the distinct A_{undiff} states and confirm identity of cell subsets, we analysed scRNA-Seq data of controls with the SCENIC algorithm (**Fig. 3d**, **Supplementary Fig. 3d**, **Supplementary Table 4** and **lines 331-344**). SCENIC predicts relative activity of regulons (transcription factors and downstream targets) in individual cells, which allows identification of transcription factors (TFs) controlling cellular states and helps resolves cell subsets (Aibar et al., 2017). The primitive SSC state was predicted to be regulated by TFs previously associated with SSC function (PDX1, EOMES and ETV5) (Ishii et al., 2012; La et al., 2018b; Sharma et al., 2019), together with other novel factors (e.g., HOXC10, ERF) (**Fig. 3d** and **Supplementary Table 4**). The progenitor population exhibited strong activity of RAR γ and SOX3, which play key roles in SSC differentiation, plus other novel factors (e.g., SALL1, IRF1) (**Fig. 3d**) (Ikami et al., 2015; McAninch et al., 2020). The proliferative SSC states were regulated by similar TFs (e.g., EGR3, HES1, HIF1A, DMRT3) while stem-progenitor transitional cells had strong activity of unique regulators including E2F7, E2F2 and EZH2 (**Fig. 3d**). Differential activity of selected regulons has also been overlaid onto UMAP plots to illustrate different cell states (**Supplementary Fig. 3d**).

4) Fig. 2a. The legend indicates that $Ddx4^+$ cells were selected for the analysis shown. Does this mean that the authors' purified cell populations containing other cell types? This crucial information

must be added to the MS. It also pertains to the experiments in which BU-mediated regeneration is examined.

In our original single cell analysis of **Fig. 2a** we selected *Ddx4*-expressing cells to ensure that we only included germ cells. However, as detailed below, given the purity of the sorted populations and filtering steps to remove the very low numbers of contaminating cells, selection of *Ddx4*⁺ cells is unnecessary. We have now repeated this analysis on all cells obtained after filtering and show expression of germ cell marker *Ddx4* alongside spermatogonial marker *Zbtb16/Plzf* to illustrate cell purity (**Fig. 2a**). Moreover, we now include details of the small clusters of contaminating populations in all samples in **Supplementary Fig. 12c, d** and **Supplementary Table 7** and highlight these in the Methods section on scRNA-Seq analysis alongside all applied filtering steps (**lines 901-959**).

Both our bulk and single cell RNA-Seq analysis were performed on A_{undiff} spermatogonia sorted according to the cell surface marker profile E-Cadherin⁺/c-KIT⁻/α6-integrin⁺. We have previously used this marker combination to isolate A_{undiff} cells from other mouse models (La et al., 2018a). E-Cadherin is a well-established marker of A_{undiff} spermatogonia including transplantable SSCs while exclusion of c-KIT⁺ cells removes those spermatogonia at early differentiation stages that have yet to downregulate E-Cadherin (La et al., 2018a; Tokuda et al., 2007). α6-integrin is also enriched on A_{undiff} cells, a known SSC marker gene and is used to maximise cell purity (Takubo et al., 2008).

Based on expression of characterised marker genes (Tan et al., 2020), our scRNA-Seq analysis demonstrates that the large majority of cells in the E-Cadherin⁺/c-KIT⁻/α6-integrin⁺ fraction from both control and busulfan (BU)-treated testes are A_{undiff} spermatogonia (information now contained in **Supplementary Fig. 12c, d** and **Supplementary Table 7**). Following standard quality control filtering, we identified two contaminating cell clusters negative for germ/spermatogonial markers (*Plzf/Zbtb16, Ddx4*) and positive for different somatic cell markers (e.g., *Cd74, Wt1, Aldh1a1, Acta2*) (**Supplementary Figure 12c, d**) (Tan et al., 2020). However, cells from these somatic clusters comprise only 0.8% and 2.5% of total cells from control and BU-treated samples respectively (**Supplementary Table 7**). Small clusters of cells positive for *Spem1*, representing contaminating spermatids (Zheng et al., 2007), are also found (**Supplementary Figure 12c, d**). Remaining cell clusters are positive for panels of spermatogonial and germ cell genes (e.g., *Plzf/Zbtb16, Sall4, Foxo1, Ddx4, Epcam*), confirming spermatogonial identity (**Supplementary Figure 3a**).

In total, **95.7%** and **94.5%** of cells from control and BU-treated samples respectively are within identified spermatogonial clusters, indicating the high degree of purity of sorted cell preparations and supporting the validity of bulk RNA-Seq analysis (**Fig. 1b, d** and **Supplementary Table 7**). Notably, one spermatogonial cluster was positive for *Kit* expression despite being negative for cell surface c-KIT protein by flow cytometry (**Fig. 1c** and **Supplementary Figure 12d**); presumably representing

A_{undiff} just initiating differentiation. We therefore excluded these *Kit+* spermatogonia from subsequent comparative analysis of control and BU-treated A_{undiff} scRNA-Seq data although these cells would still contribute to gene expression in the bulk analysis, highlighting the need for comparison of both bulk and single cell analysis.

5) *QC parameters. The authors should provide the quality matrix for the scRNAseq data, the normalization approaches used, and the cell numbers in each cluster.*

We have now extended the Methods section on scRNA-Seq to provide a more detailed description of our analysis pipeline (**lines 901-959**). This description includes filtering steps, normalisation approaches and cell numbers in each cluster. Further, we now include the following QC information: 1) Sequencing metrics from the Cell Ranger pipeline in **Supplementary Table 7**; 2) Violin plots showing QC features from the Seurat analysis for each of the experimental replicates (**Supplementary Fig. 12a**); 3) UMAP plots that illustrate the Seurat analysis pipeline including dataset integration plus alignment and initial clustering following filtering steps to remove low-quality cells (**Supplementary Fig. 12b, c**); 4) Marker gene expression used to define cell types in the initial clustering analysis (**Supplementary Fig. 12d**); 5) Cell numbers from the initial population clustering and re-clustering following exclusion of low numbers of contaminating cells (**Supplementary Table 7 and lines 932-935**); 6) Clustering analysis of individual samples to illustrate consistencies between experimental repeats, absence of significant batch effects and the relative proportions of cells in each population (**Supplementary Fig. 12e, f**).

6) *Line 240. Why “As anticipated,...”? In general, not a good idea to use phrases like this and “as expected.” Instead, state a hypothesis or give result and state why it fits a prediction*

We had used the term “as anticipated” in reference to our observation that *Gfra1*-expressing A_{undiff} downregulate *Pdx1* during regeneration and we had previously shown that PDX1 selectively marks homeostatic SSCs (La et al., 2018b). We have reworded the sentence to the following (**lines 255-256**): “We confirmed that regenerative *Gfra1*⁺ cells downregulated *Pdx1*, a selective marker of homeostatic SSCs⁵”.

7) *Line 437. This conclusion is overstated. Is it not possible that the mTORC1 inhibitor, rapamycin, suppresses the increase of KI67+ cells by increasing the number of proliferating cells that undergo apoptosis?*

It is plausible that rapamycin might reduce numbers of proliferating A_{undiff} by inducing apoptosis of cycling cells. From studies of cultured A_{undiff} , we have previously demonstrated that rapamycin induces cell cycle arrest (Hobbs et al., 2010). Further, we now provide data showing that prolonged rapamycin treatment *in vitro* does not increase numbers of apoptotic A_{undiff} by IF analysis for cleaved-caspase 3 and 9 (**Supplementary Fig. 6d**). We also note that treatment of neonatal mice with rapamycin does not result in increased apoptosis of germ cells (Busada et al., 2015). In response to the reviewers' comments, we have modified the relevant text to the following (**lines 454-457**):

“While the cytostatic effects of rapamycin on regenerative A_{undiff} could also be due to selective apoptosis of proliferating cells, the ability of rapamycin to induce cell cycle arrest of cultured A_{undiff} has previously been noted and prolonged *in vitro* treatment was not associated with elevated numbers of apoptotic A_{undiff} (Supplementary Fig. 6d)¹².”

8) *Fig. 6h-i. This data showed that inhibitors could efficiently inhibit their known signaling pathway targets. All the inhibitors tested also reduce EPCAM⁺ cells. However, given that GDNF and FGF signaling pathways are well known to be important for A_{undiff} in vitro maintenance/culture, this mainly constitutes a verification experiment. To test the specific roles of AKT and mTORC1 during regeneration, the authors should also use AKT- and mTORC1-specific inhibitors and/or activators. The authors should also add the caveat that their in vitro culture system might not recapitulate signaling pathways used during regeneration.*

In our original submission, we provided evidence that regenerative A_{undiff} *in vivo* have increased activity of AKT and mTORC1 signalling based on cytosolic FOXO1 localisation and P-S6 immunostaining (**Fig. 4b, f, g**). Consistent with a critical role of mTORC1 in the regenerative response *in vivo*, short-term treatment of mice with the mTORC1 inhibitor rapamycin suppressed A_{undiff} proliferation and substantially delayed germline recovery following BU (**Fig. 5**). However, rapamycin treatment did not affect germline recovery in the long term or abundance of EOMES⁺ regenerative SSCs at D10 after BU (**Fig. 5g** and **Supplementary Fig. 6c, e**). Moreover, while prolonged rapamycin treatment is linked to AKT suppression by effects on mTORC2 assembly (Sarbasov et al., 2006), we now confirm that rapamycin does not inhibit AKT activation in regenerative GFR α 1⁺ A_{undiff} as indicated by cytosolic FOXO1 (**Supplementary Fig. 10b**). Therefore, rapamycin does not suppress all cellular features or signalling activity associated with regeneration.

Importantly, our *in vitro* analysis indicated that concerted activation of multiple growth factor-dependent signalling pathways in A_{undiff} are involved in induction of FOXM1, as observed in regenerative A_{undiff} , and potentially the regenerative response (**Fig. 7a-c**). To test whether extensive inhibition of growth factor signalling in A_{undiff} *in vivo* leads to more pronounced effects on germline

regeneration than rapamycin, we treated mice for 5 days with the multikinase inhibitor AD80 or vehicle starting at D5 following BU (**Fig. 7d**). We had demonstrated in the original submission that treatment of cultured A_{undiff} with AD80 suppresses AKT, ERK MAPK and mTORC1 signalling and inhibits FOXM1 and EpCAM expression (now **Fig. 7b**) – key features associated with the regenerative state. This was linked to the ability of AD80 to suppress the GDNF response in cultured A_{undiff} by inhibiting RET, SRC family and other kinases (**Fig. 7a** and **Supplementary Fig. 8c**) (Dar et al., 2012). Strikingly, AD80 treatment *in vivo* suppressed expression of important markers of regenerative (and homeostatic) SSCs including $GFR\alpha 1$ and EOMES at D10 after BU (**Fig. 7f** and **Supplementary Fig. 10a**). E-Cadherin⁺ A_s and A_{pr} were still found in AD80-treated samples at this timepoint, indicating that A_{undiff} were present (**Fig. 7d, e**). This was confirmed by flow cytometry analysis for PLZF⁺ c-KIT⁻ cells (**Supplementary Fig. 10c**).

In contrast to the effects of rapamycin, treatment with AD80 suppressed both AKT and mTORC1 signalling in regenerative A_{undiff} as indicated by a switch in FOXO1 localisation to the nucleus and loss of P-S6 immunostaining in E-Cadherin⁺ A_s and A_{pr} cells from D10 samples (**Fig. 7d, e**). The proportion of regenerative A_{undiff} that were KI67⁺ was also reduced by AD80, consistent with suppressed proliferation (**Fig. 7f**). Importantly, we found by flow cytometry analysis of A_{undiff} at D10 after BU that expression of uPAR and EpCAM, key features of regenerative SSCs (RSCs), was substantially inhibited by AD80 (**Fig. 7g**). While this regimen of AD80 treatment did not appear to disrupt FOXM1 expression as effectively as in cultured A_{undiff} , the reduction in expression of multiple FOXM1 targets, e.g., CCND1 and uPAR, suggested that FOXM1 activity was disrupted by AD80 *in vivo* (**Fig. 7g, h** and **Supplementary Fig. 10d**) (Li et al., 2013; Wang et al., 2001). This new dataset indicated that AD80 treatment strongly suppressed signalling activity and characteristic features of RSCs *in vivo*, consistent with our *in vitro* data.

Given the effectiveness of AD80 in inhibiting the regenerative SSC state, we also assessed what the long-term impact of transient AD80 treatment would be on germline recovery following BU. As a testament to the robustness of the rodent germline, spermatogonial recovery was still evident in seminiferous tubules of AD80-treated mice at D30 after BU (**Fig. 7i**). Flow cytometry analysis for PLZF⁺ c-KIT⁻ cells also confirmed recovery of A_{undiff} (**Supplementary Fig. 10e**). However, while almost 90% of the tubule length in vehicle controls contained SALL4⁺ spermatogonia and active spermatogenesis at D30, only ~60% of the tubule length did in AD80-treated samples (**Fig. 7i**). Evidently, transient AD80 treatment after BU resulted in significant disruption of the RSC population. Interestingly, within the spermatogenic patches of AD80-treated mice at D30, $GFR\alpha 1$ ⁺ SSCs were more abundant than in controls and a higher proportion were KI67⁺ (**Fig. 7i** and **Supplementary Fig. 10f**), suggesting a delayed and on-going regenerative response as we found after rapamycin treatment (**Fig. 5d, e**). From flow cytometry analysis of D30 samples, an increased proportion of PLZF⁺ c-

KIT⁻ and EOMES⁺ A_{undiff} were KI67⁺, consistent with continuing regeneration (**Supplementary Fig. 10g, h**). Notably, while AD80 treatment resulted in depletion of GFR α 1⁺ and EOMES⁺ A_{undiff} at D10, these populations recovered at D30 once treatment had ceased (**Supplementary Fig. 10f, g**) illustrating the plasticity of A_{undiff} transcriptional states. Interestingly, a notable fraction of the GFR α 1 population in AD80 samples at D30 were found to co-express the progenitor marker RAR γ (**Supplementary Fig. 10i**), suggesting an accumulation of stem-progenitor transitional states and a delay in restoration of spermatogonial homeostasis.

Together with our original observations, these new datasets confirm the critical roles played by growth factor-dependent signalling, in particular the PI3K/AKT and mTORC1 pathways, in driving the regenerative capabilities of A_{undiff} *in vivo* and support the validity of our *in vitro* experiments on cultured A_{undiff}. However, in response to the reviewers' comments about the A_{undiff} culture system, we now add the following statement when describing the *in vitro* data in Fig. 7 (**lines 607-609**):

“We note that differences may exist in the role of signalling pathways in cultured A_{undiff} versus regenerative A_{undiff} *in vivo*, limiting the ability to extrapolate *in vitro* results to the regenerating testis.”

9) In vivo relevance of in vitro experiments? The authors tested the role of signaling pathways in regeneration using inhibitors with an in vitro system. It is not clear whether the effects they observed are merely due to the known role of these signaling pathways (e.g., GDNF, AKT, and mTORC1) in steady-state spermatogenesis or whether the effects also pertain to regeneration

As discussed in the response to point #8 above, we have confirmed the importance of mTORC1 in the regenerative response *in vivo* by treating mice with rapamycin from D3 to D10 following BU and analysing spermatogonial recovery over time (**Fig. 5** and **Supplementary Fig. 6**). Importantly, our results indicate that mTORC1 plays distinct roles in A_{undiff} under homeostatic and regenerative conditions (discussed in the manuscript **lines 788-818**). Multiple studies have found that mTORC1 signalling is associated with SSC differentiation and that aberrant mTORC1 activation through deletion of upstream negative regulators, e.g., TSC2, PTEN, GILZ, results in SSC exhaustion and germline degeneration (Busada et al., 2015; Goertz et al., 2011; Hobbs et al., 2015; La et al., 2018a; La and Hobbs, 2019; Suzuki et al., 2021). These reports are consistent with known roles of mTORC1 in other stem cell systems, e.g., haematopoietic (Lee et al., 2010), and the low numbers of GFR α 1⁺ SSCs that stain positive for markers of mTORC1 signalling (P-S6⁺) in steady-state testis (**Fig. 4f, g**).

In contrast, mTORC1 signalling is activated in regenerative GFR α 1⁺ SSCs as indicated by increased numbers of P-S6⁺ cells at D10 after BU (**Fig. 4f, g**). Moreover, transient treatment of mice with the mTORC1 inhibitor rapamycin suppresses proliferation and expansion of GFR α 1⁺ regenerative SSCs following BU and delays spermatogonial recovery (**Fig. 5d-g**). Our results

therefore indicate that while chronic elevation of mTORC1 signalling is detrimental for homeostatic SSC maintenance, temporary activation of mTORC1 is essential for SSC-dependent regeneration. This conclusion is supported by studies of other tissues including muscle and the haematopoietic system in which mTORC1 activation is required for the regenerative response (Kalaitzidis et al., 2012; Rodgers et al., 2014). Interestingly, repeated regenerative events drive stem cell exhaustion in epithelia due to continual cycles of mTORC1 activation (Haller et al., 2017). Similarly distinctive roles for AKT in homeostatic vs. regenerative SSCs can be proposed given that AKT signalling is restricted in homeostatic GFR α 1+ SSCs but activated in regenerative SSCs as indicated by a switch in predominant FOXO1 localisation from nucleus to cytosol (**Fig. 4b**) (discussed in **lines 804-818**). Further, chronic AKT activation and FOXO inactivation in SSCs, e.g., through deletion of upstream inhibitor PTEN or by FOXO deletion, are associated with SSC exhaustion (Goertz et al., 2011).

Given that regenerative A_{undiff} *in vivo* show activation of other signalling pathways besides mTORC1, e.g., PI3K/AKT, we have now also treated mice with the multikinase inhibitor AD80 from D5 to D10 after BU and demonstrated pronounced effects on cellular signalling, molecular features, and function of regenerative SSCs (RSCs) *in vivo* (**Fig. 7** and **Supplementary Fig. 10**). See response to point #8 above. Our analysis of cultured A_{undiff} demonstrated that multikinase inhibitors such as AD80 and ponatinib are extremely effective at preventing growth factor responses of these cells and limiting expression of pro-proliferative genes upregulated during regeneration, including the transcription factor FOXM1 (**Fig. 7b** and **Supplementary Fig. 8c**). Accordingly, the *in vivo* impact of AD80 on the regenerative response following BU is more pronounced than that of rapamycin, which only targets mTORC1. For instance, AD80 blocks both mTORC1 and AKT signalling in regenerative A_{undiff} (indicated by reduced P-S6 and nuclear FOXO1), suppresses key markers of the regenerative state including EOMES and uPAR, and impacts regenerative capacity of the germline (**Fig. 7d-i**). In contrast, rapamycin inhibits mTORC1 but not AKT in regenerative A_{undiff} , has no effect on SSC/RSC markers such as GFR α 1 and EOMES, and no evident long-term impact on germline recovery (**Fig. 5b, d** and **Supplementary Figs. 6c, e and 10b**). Our data therefore supports the model that concerted activation of growth factor-dependent signalling pathways in A_{undiff} is essential for induction of the regenerative response *in vivo*. It is interesting to note that AD80 is known to be a direct and potent inhibitor of RET and would be expected to block GDNF responses (Dar et al., 2012; Plenker et al., 2017), as we confirm using cultured A_{undiff} (**Supplementary Fig. 8c**). Our new dataset therefore provides additional support for a key role of GDNF in induction of regeneration.

In our original submission, we provided evidence that regenerative A_{undiff} had increased activity of the transcription factor and cell cycle regulator FOXM1 (**Fig. 6a-c**). Studies of cultured A_{undiff} confirmed that FOXM1 inhibition resulted in cell cycle arrest and reduced growth (**Fig. 6d-f**), linking FOXM1 to enhanced proliferation of regenerative A_{undiff} and consistent with known roles of FOXM1

in regeneration of other tissues (Wang et al., 2002). To demonstrate relevance of our findings for the A_{undiff} regenerative response *in vivo*, we assessed effects of injecting mice with an inhibitor to FOXM1 after BU treatment. Mice were therefore treated with thiostrepton from D3 to D9 following BU according to an established protocol for FOXM1 inhibition and analysed on D10 (**Fig. 6g**) (Buchner et al., 2015; Sheng et al., 2020). Thiostrepton directly binds and inhibits FOXM1 (Hegde et al., 2011), and we had previously found that treating cultured A_{undiff} with thiostrepton reduced expression of FOXM1 plus target genes and blocked proliferation (**Fig. 6d-e** and **Supplementary Fig. 7e**).

When used *in vivo*, thiostrepton modestly but significantly inhibited regenerative A_{undiff} proliferation as indicated by a lower proportion of KI67+ cells (**Fig. 6g**). Strikingly, thiostrepton treatment also significantly reduced the fraction of GFR α 1+ regenerative SSCs that were P-S6+, indicating that FOXM1 inhibition suppressed mTORC1 activation, a core feature of the regenerative response. This effect of thiostrepton was also associated with reduction in KI67 expression in these cells (**Fig. 6h, i**). While relative abundance of regenerative A_{undiff} and GFR α 1+ cells were not substantially altered by thiostrepton treatment at this timepoint (**Supplementary Fig. 7h, i**), our data indicated that FOXM1 is involved in induction of the regenerative response *in vivo*. Given the inhibitory effect of thiostrepton on mTORC1 in regenerative A_{undiff} and the positive role of signalling pathways including mTORC1 in FOXM1 expression, a positive feedback loop between FOXM1 and upstream growth factor-dependent signalling can exist in A_{undiff} . Notably, FOXM1 forms a positive feedback loop with HGF/c-MET signalling in pancreatic adenocarcinoma cells (Cui et al., 2016).

10) Somatic cells. The authors often discuss growth factor-dependent signaling pathways involving secreted factors from somatic cells, but they do not experimentally address the effect of their BU/regeneration protocol on somatic cells. Some somatic cells may have been present in their enriched samples used for their scRNAseq analysis, which would allow the authors to report on the molecular characteristics of these somatic cell populations. At the least, the authors could comment on the effect of their BU/regeneration protocol on Sertoli cells, Leydig cells, and PTMs, based on standard histological analysis of testes sections (e.g., morphology, cell number, and cell death), based either their own data or data from the literature.

While the primary focus of our manuscript concerns effects of BU treatment on A_{undiff} spermatogonia, we have now explored potential effects of BU on somatic populations and discussed key studies from the literature (**lines 212-223**). As we discuss, it has been documented that Sertoli and Leydig cell populations of mice appear minimally affected by BU treatment, even when used at higher doses than in our study (30-40mg/kg vs. 10mg/kg) (Bucci and Meistrich, 1987; O'Shaughnessy et al., 2008). Moreover, steroidogenic activity of murine Leydig cells, as judged by intratesticular testosterone

levels and expression of enzymes within the steroidogenic pathway, are unaffected by BU (O'Shaughnessy et al., 2008). In agreement with these studies, we have now confirmed that abundance of SOX9+ Sertoli cells within the seminiferous tubules is unaffected at D10 following BU treatment (**Supplementary Fig. 1d**). Moreover, Leydig populations, identified as c-KIT+ interstitial cells (Rothschild et al., 2003), appeared essentially normal after BU (**Supplementary Fig. 1d**). Immunostaining for smooth muscle actin also revealed an intact layer of peritubular myoid cells surrounding the seminiferous tubules after BU (**Supplementary Fig. 1e**) (Maekawa et al., 1996).

Our analysis suggested that key testis somatic populations were unaffected by a low dose of BU. However, as we now comment, functional activity of these supporting cells may be affected. Expression of many Sertoli cell-specific genes is reportedly sensitive to the germ cell loss accompanying BU treatment (O'Shaughnessy et al., 2008). BU may also affect testis endothelial cells and disrupt interstitial microvessels (Bhang et al., 2018). Unfortunately, as we performed scRNA-Seq on sorted A_{undiff} populations, our genomics analysis provided little insight into functional changes of the somatic cell populations. As detailed in the response to point #4 above, a very limited number of contaminating somatic cells were present in the scRNA-Seq datasets. Specifically, two distinct somatic cell clusters comprising 0.8% and 2.5% of total cells from control and BU-treated samples were identified respectively (**Supplementary Fig. 12c, d** and **Supplementary Table 7**). Based on marker gene expression of these contaminating clusters, they represented peritubular myoid cells (*Acta2+*) and innate lymphoid cells (*Cd74+*, *Cnmd+*) (**Supplementary Fig. 12d**) (Tan et al., 2020). However, given that so few cells in our dataset were present in these clusters and that these contaminating cells may not be entirely representative of the populations within the testis, we have not performed a detailed analysis of these cells. A comprehensive assessment of the effects of BU on distinct testis somatic populations is beyond the scope of our current study.

11) Other physiological factors. Likewise, information on regeneration-induced alterations in other factors, such as hormone levels, would enrich the authors' story.

As discussed above and referenced in the revised text (**lines 212-223**), levels of circulating FSH and intratesticular testosterone in the mouse are reportedly unaffected by 30mg/kg BU treatment (O'Shaughnessy et al., 2008). Levels of reproductive hormones are therefore unlikely to be changed by the 10mg/kg low dose BU we use in our study.

REVIEWER #2

In this study the authors have made an important investigation toward a fuller understanding of regeneration process by spermatogonial stem cells (SSCs) after insult. In particular, they have revealed the molecular signatures of residual SSCs 10 days after busulfan treatment (BU), when regeneration is initiating, by bulk and single cell RNA-seq analyses of FACS-sorted undifferentiated spermatogonia population. They figured out 6 clusters of spermatogonia, whose balance is tilted toward more primitive states after BU, and revealed transcriptional evidence suggesting that, beside similar molecular signatures, the regenerative SC's are mitotically active due to increased growth factor signaling. They also found by in vitro and in vivo studies that mTOR pathways is activated transiently after BU. They further show the possible involvement of FOXM1 as a potential key regulator of activated cell cycle on regeneration. Overall the manuscript is well written in an evidence-based fashion and the conclusion is well supported by the data. I would point some issues that would further improve this informative and substantial study.

We thank the reviewer for providing detailed comments that have helped us improve the manuscript and for highlighting the importance of our study. In our revised manuscript we now provide a more compelling dataset in support of our model of germline regeneration (**Fig. 8**).

Importantly, we have revised and extended our analysis of single cell (sc)RNA-Seq data from regenerative A_{undiff} spermatogonia and incorporated additional experimental replicates (**Fig. 2, 3** and **Supplementary Fig. 2-4**). While the conclusions are broadly unchanged, the analysis of a larger set of cells now provides a more detailed and valuable resource for the field (described in **lines 225-371**). By analysing our scRNA-Seq dataset with the updated RNA velocity algorithm scVelo we now provide evidence for dynamic transition of quiescent “primitive” SSCs into both proliferative SSCs and differentiation-primed progenitor states in regenerative samples and predict genes that promote these transitions (**Fig. 3f** and **Supplementary Fig. 4b**) (Bergen et al., 2020). This revised analysis highlights the distinct fates available to SSCs and associated regulatory mechanisms. For example, transition from the primitive SSC to progenitor state is predicted to be driven by FOXO1 and DNMT3B (**Supplementary Fig. 4b**), transcriptional modulators involved in A_{undiff} differentiation (Goertz et al., 2011; Shirakawa et al., 2013). The primitive to proliferative SSC transition is predicted to be driven by specific cell cycle regulators and other factors including CDK1 and USP26.

Analysis of our scRNA-Seq data with the SCENIC algorithm predicted transcription factors (TFs) that functionally regulate each of the identified A_{undiff} cell clusters (**Fig. 3d**, **Supplementary Fig. 3d** and **Table 4**). SCENIC calculates activity of regulons (TFs and predicted targets) in individual cells, allowing identification of TFs controlling cell states and resolving cell subsets (Aibar et al., 2017). For example, the primitive SSC state is predicted to be driven by TFs including PDX1, EOMES and

ETV5, consistent with previous studies (La et al., 2018b; Oatley et al., 2007; Sharma et al., 2019). In contrast, progenitors were predicted to have high activity of RAR γ and SOX3, in agreement with *in vivo* studies (Gely-Pernot et al., 2012; Ikami et al., 2015; McAninch et al., 2020). The identification of known regulators of A_{undiff} function from analysis of scRNA-Seq data with SCENIC and scVelo then supports the study of genes predicted to be involved in A_{undiff} fate transitions and function that currently have poorly defined roles in the germline (**Fig. 3d, Supplementary Fig. 4b and Table 4**).

We also provide new *in vivo* evidence in support of the cellular pathways involved in germline regeneration. For instance, treatment of mice with the FOXM1 inhibitor thiostrepton after busulfan (BU) suppresses proliferation of regenerative A_{undiff} and reduces mTORC1 activation in regenerative GFR α 1+ cells (**Fig. 6g-i**). These data confirm that FOXM1 plays important roles in the regenerative response and reveal positive feedback pathways from FOXM1 to upstream growth factor signalling in A_{undiff}, as in other cell types (Cui et al., 2016). Given our data suggested involvement of distinct signalling pathways including PI3K/AKT and mTORC1 in the A_{undiff} regenerative response, we now test effects of the multikinase inhibitor AD80 on germline regeneration following BU treatment (**Fig. 7d-i and Supplementary Fig. 10**). Strikingly, AD80 treatment blocked both mTORC1 and AKT signalling in regenerative A_{undiff} (indicated by reduced P-S6 and nuclear FOXO1), suppressed expression of key markers of the regenerative state including EOMES and uPAR, and impacted germline recovery (**Fig. 7d-i**). In contrast, rapamycin inhibited mTORC1 but not AKT in regenerative A_{undiff}, had no effect on regenerative SSC markers such as GFR α 1 and EOMES, and no long-term impact on germline regeneration (**Fig. 5b, d and Supplementary Figs. 6c, e and 10b**). This new data supports our model that the concerted activation of growth factor signalling pathways in A_{undiff} is essential for induction of the regenerative response (**Fig. 8**). Given the ability of AD80 to inhibit directly and potently c-RET (Plenker et al., 2017), our data provides additional support for a key role of GDNF during the initiation of regeneration. This new dataset is described in **lines 650-727**.

The responses to specific reviewer comments are listed below.

Comments:

1. Throughout, they frequently use the term “growth factors” that stimulate the mitotically active state of residual SSCs after insult. Although this is quite understandable, the described particular factors are not consistent- in many cases only GDNF is raised while in other parts GDNF plus FGFs are listed. In few cases and, importantly in summary scheme in Fig. 6g, Insulin is also listed. While it looks like that these factors work together to activate the mitotic state of SSCs after BU, they should also play distinct (or non-redundant) roles. So presumptive common and distinctive roles of these growth factor warrant more substantial statements throughout the manuscript

We agree that different microenvironmental growth factors likely play both overlapping and non-redundant roles in germline regeneration. For example, both GDNF and bFGF are linked to induction of SSC-associated genes including *Lhx1*, *Bcl6b* and *Etv5* but can have distinct effects on A_{undiff} fate (Ishii et al., 2012; La et al., 2018b; Masaki et al., 2018; Oatley et al., 2006). While GDNF and bFGF activate similar downstream pathways in cultured A_{undiff} , e.g., AKT, mTORC1 and ERK MAPK, they do so to differing extents and, presumably, with different kinetics (**Supplementary Fig. 8c**) (Hobbs et al., 2010; Ishii et al., 2012; Lee et al., 2007). They may also activate unique signalling components in A_{undiff} although the identity of such pathways is unclear. Differences in signalling outputs are predicted to underlie the distinct responses of A_{undiff} to GDNF and bFGF. We note, however, that different FGFs have been demonstrated to play redundant roles in regulating SSC density within seminiferous tubules (Kitadate et al., 2019). While our primary focus concerns the role of specific signalling pathways in the A_{undiff} regenerative response, we have revised the text wherever possible to highlight potentially distinct roles of different growth factors in regeneration, e.g., **lines 202-204** and **377-379**. Moreover, our new dataset characterising effects of the multikinase inhibitor AD80 on the regenerative response supports a central role for GDNF in this process (see below).

Based on analysis of our RNA-Seq dataset, we provide evidence to indicate that multiple growth factors, receptors, and downstream pathways are involved in the regenerative response. We show that A_{undiff} of regenerating testis typically upregulate genes known to be stimulated by GDNF in cultured A_{undiff} and downregulate genes suppressed by GDNF (**Fig. 1h** and **Supplementary Table 1**) (Oatley et al., 2006). While FGF target genes in spermatogonia are poorly defined, genes induced by FGF in developing *Xenopus laevis* (*Spry4*, *Dusp6*, *Fgfr2*, *Spry1*) and FGF-responsive SSC-associated genes (*Lhx1*, *Bcl6b*) were upregulated in regenerative A_{undiff} (**Fig. 1h**) (Branney et al., 2009; Ishii et al., 2012). In line with the well-appreciated and synergistic roles of GDNF and FGFs in promoting A_{undiff} self-renewal and proliferation (Kitadate et al., 2019; La and Hobbs, 2019; Meng et al., 2000) this analysis supports roles for both GDNF and FGF in promoting the regenerative A_{undiff} state.

Pathway analysis of our RNA-Seq data predicted that additional growth factors and signalling components positively regulated the regenerative A_{undiff} state, e.g., bFGF, HGF, IGF1, TGF β 1, ERBB2, PI3K, HRAS and ERK MAPK (**Fig. 6a** and **Supplementary Table 5**). Many of these predicted stimuli would feed into common downstream signalling pathways including mTORC1, such that mTORC1 inhibition would be expected to partially abrogate response to multiple growth factors and signalling proteins (Laplante and Sabatini, 2009). This is consistent with ability of the mTORC1 inhibitor rapamycin to suppress the regenerative response (**Fig. 5**). However, characterising distinct contributions of the various growth factors and signalling components in germline regeneration requires substantial additional studies and is beyond the scope of the manuscript.

Growth factor stimulation of A_{undiff} is expected to activate multiple signalling components and pathways besides mTORC1 that can be required for regeneration. To test whether extensive inhibition of growth factor signalling in A_{undiff} leads to more pronounced effects on germline regeneration than rapamycin, we treated mice for 5 days with the multikinase inhibitor AD80 or vehicle starting at D5 following BU (**Fig. 7d**). We had demonstrated in our original submission that treatment of cultured A_{undiff} with AD80 suppressed AKT, ERK MAPK and mTORC1 signalling and inhibited FOXM1 and EpCAM expression (now **Fig. 7b**) – key features associated with the regenerative state. This was linked to the ability of AD80 to suppress the GDNF response in cultured A_{undiff} by inhibiting RET, SRC family and other kinases (**Fig. 7a** and **Supplementary Fig. 8c**) (Dar et al., 2012).

Strikingly, AD80 treatment *in vivo* suppressed expression of important markers of regenerative (and homeostatic) SSCs including GFR α 1 and EOMES at D10 after BU (**Fig. 7f** and **Supplementary Fig. 10a**). E-Cadherin⁺ A_s and A_{pr} were still found in AD80-treated samples at this timepoint, indicating that A_{undiff} were present (**Fig. 7d, e**). This was confirmed by flow cytometry analysis for PLZF⁺ c-KIT⁻ cells (**Supplementary Fig. 10c**). In contrast to the effects of rapamycin, both AKT and mTORC1 signalling were suppressed in regenerative A_{undiff} by AD80 as indicated by a switch in FOXO1 localisation to the nucleus and loss of P-S6 immunostaining in E-Cadherin⁺ A_s and A_{pr} cells from D10 samples (**Fig. 7d, e**). The proportion of regenerative A_{undiff} that were KI67⁺ was also reduced by AD80, consistent with suppressed proliferation (**Fig. 7f**). Importantly, we found by flow cytometry analysis of A_{undiff} at D10 after BU that expression of uPAR and EpCAM, key features of regenerative SSCs (RSCs), was substantially inhibited by AD80 (**Fig. 7g**). While this regimen of AD80 treatment did not appear to disrupt FOXM1 expression as effectively as in cultured A_{undiff} , the reduction in expression of multiple FOXM1 targets, e.g., CCND1 and uPAR, suggested that FOXM1 activity was disrupted by AD80 *in vivo* (**Fig. 7g, h** and **Supplementary Fig. 10d**) (Li et al., 2013; Wang et al., 2001). These new data indicated that AD80 strongly suppressed the characteristic features and signalling activity of RSCs *in vivo*, consistent with our *in vitro* data.

Given the effectiveness of AD80 in inhibiting the regenerative SSC state, we also assessed what the long-term impact of transient AD80 treatment was on germline recovery following BU. As a testament to the robustness of the rodent germline, spermatogonial recovery was still evident in seminiferous tubules of AD80-treated mice at D30 after BU (**Fig. 7i**). Flow cytometry analysis for PLZF⁺ c-KIT⁻ cells also confirmed recovery of A_{undiff} (**Supplementary Fig. 10e**). However, while almost 90% of the tubule length from BU-treated control mice contained SALL4⁺ spermatogonia and active spermatogenesis at D30, only ~60% of the tubule length did from AD80-treated samples (**Fig. 7i**). Transient AD80 treatment after BU therefore resulted in a significant disruption to the repopulating stem cell (RSC) population. Interestingly, within the spermatogenic patches of AD80-treated mice at D30, GFR α 1⁺ SSCs were more abundant than in controls and a higher proportion

were KI67+ (**Fig. 7i** and **Supplementary Fig. 10f**), suggesting a delayed and on-going regenerative response as we found after rapamycin (**Fig. 5d, e**). From flow cytometry analysis of D30 samples, an increased proportion of PLZF+ c-KIT⁻ and EOMES⁺ A_{undiff} were KI67+, consistent with continuing regeneration (**Supplementary Fig. 10g, h**). Notably, while AD80 treatment resulted in depletion of GFR α 1+ and EOMES⁺ A_{undiff} at D10, these populations had recovered at D30 after treatment had ceased (**Supplementary Fig. 10f, g**) illustrating the plasticity of A_{undiff} transcriptional states.

Together with our original observations, these new datasets confirm the critical roles played by growth factor-dependent signalling in driving the regenerative capabilities of A_{undiff} *in vivo*. Moreover, given that AD80 is known to be a potent inhibitor of RET signalling and can block GDNF responses in A_{undiff} (**Supplementary Fig. 8c**) (Dar et al., 2012; Plenker et al., 2017), these data provide additional support for the importance of GDNF in the regenerative response.

2. In particular, based on the studies in homeostasis, the roles of insulin on SSCs have not been described compared with GDNF or FGF. For example, given the general knowledge it might also be a possibility that insulin supports differentiation-primed (and actively dividing) state by (moderately?) activating Foxo pathway. Insulin appears, generally, to be also more relevant to mTOR and Foxo pathways. So, it would be interesting, in regeneration, if Insulin act on SSCs toward the same direction with GDNF or FGF (maintenance of or transition to primitive states). If data-based discussion about this issue is not feasible, literature based discussion would significantly increase the insight about this interesting issue.

As highlighted by the reviewer, the effects of key growth factors including GDNF, FGFs and insulin or insulin-like growth factors (IGFs) on A_{undiff} fate and the regenerative response are of great interest although not a primary focus of the current study. The bulk of our data supports central roles for GDNF and FGFs in regeneration (see above) and it is challenging to make conclusions regarding roles of insulin or IGFs on regenerative A_{undiff} function given the limited amount of relevant literature. For instance, in contrast to GDNF, specific genes regulated by insulin or IGFs in A_{undiff} are uncharacterised. However, we note that insulin is routinely used in A_{undiff} culture medium and supplementation of cultures with IGF1 significantly enhances transplantation capacity (Kanatsu-Shinohara et al., 2003; Kubota et al., 2004; La et al., 2018b). Selective roles of IGF1 in A_{undiff} G2/M cell cycle transition and SSC activity have also been reported based on *in vitro* studies (Wang et al., 2015). Consistent with these reports, pathway analysis of our RNA-Seq data predicted that IGF1 promotes the A_{undiff} regenerative state alongside other growth factors (**Fig. 6a** and **lines 486-493**).

Based on this limited literature and our pathway analysis, we tested effects of the insulin and IGF1 receptor inhibitor (linsitinib/OSI-906) on signalling pathway activity and FOXM1 expression in

cultured A_{undiff} in our original submission (now **Fig. 7b**) (Mulvihill et al., 2009). Linsitinib treatment generally had modest effects on mTORC1 and ERK MAPK signalling plus FOXM1 levels in cultured A_{undiff} arguing against a dominant role of insulin or IGF in the regenerative response (**Fig. 7b**). However, levels of P-AKT were consistently reduced by linsitinib, indicating that insulin and/or IGF1 play roles in regulating the PI3K/AKT pathway in A_{undiff} . In response to the reviewers comments we confirmed that linsitinib increased the proportion of cultured A_{undiff} with nuclear FOXO1, demonstrating that insulin/IGF1 regulates the AKT-FOXO1 pathway in these cells (**Supplementary Fig. 8b** and **lines 576-580**). We note that effects of signalling inhibitors on cultured A_{undiff} will be influenced by composition of the medium and may not be entirely reflective of the *in vivo* situation.

Our *in vitro* analysis suggested that growth factors other than insulin or IGFs would play central roles in activating PI3K/AKT and mTORC1 pathways in A_{undiff} , including GDNF and FGFs (**Fig. 7b** and **Supplementary Fig. 8c**). Essentially, any growth factor that promotes activity of PI3K/AKT would be expected to inhibit FOXO TFs and stimulate mTORC1 signalling in A_{undiff} (**Fig. 7a**) (Eijkelenboom and Burgering, 2013; Hobbs et al., 2010; Laplante and Sabatini, 2012). For instance, the multikinase inhibitor AD80 effectively suppressed AKT plus mTORC1 signalling in cultured A_{undiff} maintained in standard medium (**Fig. 7b**) and promoted nuclear accumulation of FOXO1 (now shown in **Supplementary Fig. 8b**). However, AD80 poorly suppressed the ability of insulin to stimulate AKT in growth factor starved A_{undiff} while effectively blocked the signalling response to GDNF and partially inhibited the bFGF response (**Supplementary Fig. 8c**). Our results are in line with development of AD80 as an inhibitor of RET-based signalling and the ability of AD80 to directly bind and inhibit RET and hence block GDNF responses (Dar et al., 2012; Plenker et al., 2017). Results from use of AD80 therefore support GDNF and potentially FGF as key regulators of FOXO TFs and mTORC1 in A_{undiff} . Note, however, that selective inhibition of FGF receptors with AZD4547 had limited effects on AKT activity and FOXO1 localisation in A_{undiff} cultured in standard media (**Fig. 7b** and **Supplementary Fig. 8b**). Critically, treatment of mice with AD80 lead to a profound suppression of the A_{undiff} regenerative response associated with inhibition of both PI3K/AKT and mTORC1 (see above) (**Fig 7d-i**), linking GDNF signalling and regeneration *in vivo*.

Together with our RNA-Seq analysis, these results support roles for GDNF and FGF in germline regeneration as highlighted in the manuscript. As our evidence for a specific role of insulin or IGF in regeneration is very limited, we have not specifically discussed this outside of commenting on effects of linsitinib of cultured A_{undiff} . We note however that regenerative SSCs strongly upregulate *Igf2* (**Fig. 2b**), which is known to play autocrine/paracrine roles in regeneration of other tissues, and we are currently following up on this observation (Modi et al., 2015; Wang et al., 2018).

3. Similarly, as the authors have stated, in many instances, the results of *in vitro* study may not be consistently applied to *in vivo* findings (in particular the roles of AKT). This is interesting and suggestive, warranting the interpretations by the authors.

The differences between *in vitro* and *in vivo* regulation of A_{undiff} function are of great interest, as is the exact developmental relationship between cultured cells and their *in vivo* counterparts. The A_{undiff} culture system has been pivotal in dissecting many aspects of SSC biology (La and Hobbs, 2019). However, as we discuss in the Introduction (**lines 110-128**), pathways such as PI3K-AKT can apparently have distinct roles in A_{undiff} *in vivo* and *in vitro*. Specifically, AKT activation was shown to be essential downstream GDNF for A_{undiff} self-renewal and expansion *in vitro* but aberrant activation of AKT *in vivo* is associated with SSC exhaustion (Goertz et al., 2011; Lee et al., 2007; Oatley et al., 2007). Notably, we observed increased AKT activation in regenerative GFR α 1 SSCs following BU, suggesting that cultured A_{undiff} may be in a constant regenerative state and hence reliant on AKT. Consistent with this idea, mTORC1 activity is low in homeostatic SSCs but activated during regeneration *in vivo* and required for A_{undiff} proliferation and growth *in vitro* (Hobbs et al., 2015; Hobbs et al., 2010). Moreover, both cultured and regenerating SSCs are actively expanding in response to microenvironmental cues and might be expected to be in similar functional states. Data generated using cultured cells may therefore be more relevant for our understanding of SSCs under regenerative than homeostatic conditions, supporting use of the culture system in our manuscript. We note that key observations made *in vitro* with cultured A_{undiff} have been confirmed with the regenerating germline *in vivo* through treatment with pathway inhibitors, e.g., AD80, thiostrepton.

Given that our focus is primarily on differences between SSCs of homeostatic and regenerative tissue, we have dedicated a significant proportion of the Discussion to considering roles and regulation of mTORC1 and PI3K/AKT pathways in germline regeneration vs. long-term tissue maintenance (**lines 788-818**). However, these comments may apply equally to functional distinctions between homeostatic SSCs *in vivo* and cultured SSCs. As there are some uncertainties regarding physiological relevance of cultured A_{undiff} , we have added the following statement when describing the *in vitro* data of Fig. 7 (**lines 607-609**): “We note that differences may exist in the role of signalling pathways in cultured A_{undiff} versus regenerative A_{undiff} *in vivo*, limiting the ability to extrapolate *in vitro* results to the regenerating testis.”

4. The manuscript may be lengthy with too much information; some results may not be directly related to the conclusion or are not even mentioned at all in discussion. Such results are suggested to be deleted or largely trimmed so that the main conclusion can be stressed in a clearer manner. Such

“branches” include 1) the cycloheximide treatment, 2) the roles of LHX1, and 3) repeated BU treatment.

We have made substantial revisions to the text to improve flow and clarity of the manuscript. In particular, the repeated regeneration section has been removed from the Results section and this dataset now briefly referred to in the Discussion (**Supplementary Fig. 11 and lines 819-832**). The repeated regeneration dataset had somewhat limited conclusions but is highly relevant to our discussion of effects of repeated damage on integrity of stem cells and can prompt future studies.

The LHX1 knockout section has now been moved to after the FOXM1 results (**Supplementary Fig. 9**). Given the striking induction of *Lhx1* we observe in regenerative A_{undiff} and the published *in vitro* data supporting a key role for this TF in SSC self-renewal (Oatley et al., 2007), we feel it is important to highlight redundancy of this gene in maintenance and regeneration of the germline. This section now provides a contrast to the positive role uncovered for the TF FOXM1 in regeneration.

Our regeneration model is dependent on treatment with the single chemotherapeutic drug BU. Therefore, we think it is important to include data that illustrates effects of a distinct genotoxic drug on the germline to characterise common features of the A_{undiff} regenerative state. We chose to use cyclophosphamide as is damaging to the male germline, in regular use in the clinic and appropriate doses for mouse models have previously been defined (Aloisio et al., 2014; Meistrich, 2013).

5. Regarding the discussions based on the finding that, after BU, primitive states become more dominant while differentiation-tilted states shrink, in addition to their resistance against BU, the author is suggested to also place more emphasize on the mechanism of reversion. Reversion has been shown experimentally by classic lineage tracing study of Neurog3+ cells following busulfan treatment (Nakagawa et al., 2007, 2010). In Result (L226) and in Discussion paragraph starting from line 603, interpretations may better to be made in relation to these preceding studies. As is also the case for interpretation of the result of Velocity analysis in L617.

As highlighted by the reviewer, reversion of progenitors to SSCs has been demonstrated to be an important mechanism underlying efficient germline recovery following damage (Carrieri et al., 2017; Nakagawa et al., 2007; Nakagawa et al., 2010). While our manuscript does not directly address this phenomenon, the RNA-Seq and IF data indicating a switch in predominant fate within the A_{undiff} pool to an SSC state following damage is entirely consistent with this concept (**Fig. 1d-f and 2a**). We have discussed progenitor-to-SSC reversion during regeneration in the introduction and referenced these key studies (**lines 74-80**). We have also emphasised this mechanism in the Results section following discussion of bulk and single cell RNA-Seq data (**lines 240-243**). Interestingly, studies of PAX7+

and EOMES+ A_{undiff} have suggested that subsets of SSCs might be selectively resistant to BU treatment (Aloisio et al., 2014; Sharma et al., 2019). The observed switch in predominant fate of A_{undiff} to an SSC state may therefore be due to both progenitor-to-SSC reversion and selective resistance of SSCs to BU. As we do not provide direct evidence to address relative contributions of these two mechanisms in the switch of A_{undiff} fate during regeneration, and given evidence supporting progenitor reversion, we have re-worded the relevant section in the Results section to the following and included appropriate references (**lines 240-243**):

“Based on bulk and single cell RNA-Seq data we concluded that A_{undiff} at initial regeneration stages primarily adopted an SSC-like state (Figs. 1d and 2a). This implies that surviving progenitor cells revert into a self-renewing state during initiation of regeneration and/or the progenitor population is selectively sensitive to BU and SSCs are resistant^{11,20-22}.”

Regarding RNA velocity analysis, our revised analysis using the more robust scVelo algorithm and a larger dataset (see above), provides limited evidence for progenitor-to-SSC reversion (**Fig. 3f**). Importantly, scVelo avoids key assumptions on splicing rates and dynamics incorporated into the original velocity algorithm and is more appropriate for systems with transient cell states, such as the A_{undiff} pool (Bergen et al., 2020; La et al., 2018b). While our analysis indicated that the A_{undiff} population was relative static in homeostatic tissue, long velocity streamlines between clusters suggested rapid and dynamic interconversion of cell states during regeneration (**Fig. 3f**). In particular, the velocity-inferred directionality indicated that quiescent “primitive” SSCs (cluster 0) transition into both proliferative SSC and progenitor states (clusters 4 and 1 respectively) under regenerative conditions. These results suggest that during germline regeneration, SSCs are triggered to both expand the SSC pool through increased proliferation and generate progenitors for spermatogenic recovery. Further, that SSCs do not have to enter a proliferative state prior to forming progenitors.

However, evidence for active reversion of progenitors (cluster 1) or stem/progenitor transitional cells (cluster 2) back to SSC states (clusters 0, 4, 5) during regeneration was less evident (**Fig. 3f**). As we isolated A_{undiff} at D10 after BU for analysis, cell reversion may already have occurred and A_{undiff} at earlier timepoints may need to be analysed to characterise this process. Interestingly, scVelo indicated that reversion of stem/progenitor transitional cells to proliferative SSCs occurred under homeostatic conditions (**Fig. 3f**), consistent with the dynamic nature of A_{undiff} (La et al., 2018b; Nakagawa et al., 2010). However, in the absence of lineage-tracing or similar functional data, we cannot make definitive conclusions concerning cell interconversion. We have updated the text to reflect our revised RNA velocity analysis (**lines 350-371 and 753-758**).

6. *The result of repeated BU is confusing. While the EpCAM expression is diminished, the cellular response including the recovery to the homeostasis density appear not to be affected. This is reflected by the summary sentence of the related Result paragraph starting from L491. More confusingly, in Discussion section L664, the authors describe that “repeated germline depletion blunted Aundiff regenerative responses”, which I cannot agree with given the apparently unaffected cellular responses (despite the impaired EpCAM upregulation). I would advise the authors to provide more detailed explanations to this matter. Alternatively, given that this is not relevant to the main conclusion of this paper, related description could be deleted.*

We agree with the reviewer that the repeated BU dataset is in some respects inconclusive. While reduced induction of EpCAM in A_{undiff} following repeated BU suggested a disrupted regenerative response, other markers of the regenerative response (enhanced abundance of P-S6⁺ and KI67⁺ cells within the GFR α 1⁺ SSC pool), and spermatogonial recovery appeared unaffected. We have therefore removed this section from the Results and now briefly refer to this dataset in the Discussion as is relevant for our discussion of the effects of repeated tissue damage on tissue stem cell integrity and role of mTORC1 in this process (**lines 819-832** and **Supplementary Fig. 11**). We have removed suggestions that the A_{undiff} regenerative response was blunted by repeated damage in this setting.

Interestingly, Trp63⁺ stem cells of mouse tracheal epithelium are significantly depleted in response to repeated (3x) rounds of damage induced by SO₂ exposure; an effect linked to repeated activation of mTORC1 during epithelial regeneration (Haller et al., 2017). This study therefore prompted us to assess whether multiple (3x) rounds of BU treatment would similarly result in SSC depletion and disrupt germline recovery. Given that germline recovery following repeated BU treatment appeared comparable to that of mice receiving a single BU challenge (**Supplementary Fig. 11**), our results highlight the relative robustness of the rodent germline to repeated damage. Future studies are therefore needed to test the exact limits of SSC regenerative capacity.

7. *A minor comment. In L74, “committed” is too strong in this context since they are evidenced to be reversible to self-renewing state, and I would suggest “differentiation-destined progenitors”. Otherwise, the authors may use other appropriate term here.*

We have changed the term “committed” to “differentiation-destined” as suggested. The revised sentence now reads as follows (**lines 74-75**):

“In contrast, A_{undiff} marked by NGN3 or MIWI2 mostly act as differentiation-destined progenitors in undisturbed tissue”

REVIEWER #3

The authors developed and characterized a model for spermatogonial stem cell regeneration based on chemotherapy. They define a unique regenerative state in repopulating cells, which is proposed to be driven by growth factors for the niche. They test the function of several potentially relevant genes that mediate regeneration. Transient signaling through mTORC1 was important for initiating regeneration. They found Lhx1 was dispensable for both normal spermatogenesis and regeneration. Foxm1 was identified as novel transcription factor driving the cycle response. I enjoyed reading this interesting manuscript and find that the methods are sound and the conclusions are generally justified by the data. It extends their previous finds nicely. It is likely to influence future studies of spermatogonial stem cells but would also be of interest to the general stem cell field. It could be strengthened a bit by some additional experiments and some editing.

We thank the reviewer for their positive comments and the constructive and detailed points to help improve the manuscript. We note that besides providing substantial additional data demonstrating the *in vivo* relevance of our observations and providing extra experimental replicates (see below), we have revised and extended our single cell (sc)RNA-Seq analysis to incorporate a larger dataset and take advantage of new algorithms, e.g., scVelo and SCENIC. We therefore provide a more insightful analysis of A_{undiff} heterogeneity under homeostatic and regenerative conditions and a better resource for the field (discussed in **lines 225-371**). Responses to specific queries are listed below.

Major points:

Many of the conclusions regarding the importance and function of Foxm1 are based on SSC culture (which I find convincing but slightly limited). However, transplantation analysis of Foxm1 mutant cells or equivalent genetic data would greatly strengthen these conclusions. Alternatively, can the FOXM1 inhibitor be used in vivo?

In our original submission, we provided evidence based on RNA-Seq analysis and immunostaining that regenerative A_{undiff} *in vivo* displayed increased activity of the transcription factor and cell cycle regulator FOXM1 (**Fig. 6a-c**). Studies of cultured A_{undiff} confirmed that FOXM1 inhibition resulted in cell cycle arrest and reduced cell growth (**Fig. 6d-f**), linking FOXM1 to the enhanced proliferation of regenerative A_{undiff} and consistent with known role of FOXM1 in regeneration of other tissues (Wang et al., 2002). To confirm *in vivo* relevance of our observations, we have now assessed effects of treating mice with an inhibitor to FOXM1 on the A_{undiff} regenerative response. Mice were treated with thiostrepton from D3 to D9 following BU according to an established protocol for FOXM1 inhibition and analysed on D10 (**Fig. 6g**) (Buchner et al., 2015; Sheng et al., 2020). Thiostrepton

directly binds and inhibits FOXM1 (Hegde et al., 2011), and we had previously found that treating cultured A_{undiff} with thiostrepton reduced expression of FOXM1 plus target genes and blocked proliferation (**Fig. 6d-e** and **Supplementary Fig. 7e**).

When used *in vivo*, thiostrepton modestly but significantly inhibited the proliferation of regenerative A_{undiff} (PLZF⁺ c-KIT⁻) as indicated by a lower proportion of KI67⁺ cells by flow cytometry (**Fig. 6g**). Strikingly, thiostrepton treatment also significantly reduced the fraction of GFR α 1⁺ regenerative SSCs that were P-S6⁺, indicating that FOXM1 inhibition suppressed mTORC1 activation, a core feature of the regenerative response. This effect of thiostrepton was associated with reduction in KI67 expression in these cells (**Fig. 6h, i**). While relative abundance of regenerative A_{undiff} and GFR α 1⁺ cells were not substantially altered by thiostrepton treatment at this timepoint (**Supplementary Fig. 7h, i**), our data indicate that FOXM1 is involved in induction of the regenerative response *in vivo*. Given the effect of thiostrepton on mTORC1 activation in regenerative A_{undiff} and the positive role of signalling pathways including mTORC1 in FOXM1 expression, our data suggests that a positive feedback loop between FOXM1 and upstream growth factor-dependent signalling exists in A_{undiff} . Notably, FOXM1 forms a positive feedback loop with HGF/c-MET signalling in pancreatic adenocarcinoma cells (Cui et al., 2016).

Our *in vitro* analysis indicated that concerted activation of multiple growth factor-dependent signalling pathways was required for efficient induction of FOXM1 in A_{undiff} and the regenerative response (**Fig. 7a-c**). Accordingly, regenerative A_{undiff} *in vivo* have increased activity of both AKT and mTORC1 based on cytosolic FOXO1 localisation and P-S6 immunostaining (**Fig. 4b, f, g**). When a single signalling component such as mTORC1 is inhibited by rapamycin in cultured A_{undiff} , only modest effects on FOXM1 levels are observed (**Supplementary Fig. 8f**). Consistent with this observation, short-term treatment of mice with rapamycin limited A_{undiff} proliferation and delayed spermatogonial recovery after BU but did not ultimately disrupt the regenerative SSC pool or germline recovery (**Fig. 5c-g** and **Supplementary Fig. 6e**). Note that prolonged rapamycin treatment is linked to AKT suppression by effects on mTORC2 assembly (Sarbasov et al., 2006). However, we confirmed that rapamycin does not suppress AKT in regenerative GFR α 1⁺ A_{undiff} as indicated by cytosolic FOXO1 (**Supplementary Fig. 10b**). Therefore, rapamycin does not suppress all signalling activity associated with regenerative A_{undiff} and their long-term regenerative capacity.

To test whether extensive inhibition of growth factor signalling in A_{undiff} *in vivo* leads to more pronounced effects on germline regeneration than rapamycin, we treated mice for 5 days with the multikinase inhibitor AD80 or vehicle starting at D5 following BU (**Fig. 7d**). In our original submission we demonstrated that treatment of cultured A_{undiff} with AD80 suppressed AKT, ERK MAPK and mTORC1 signalling and inhibited FOXM1 and EpCAM expression (now **Fig. 7b**) – key features associated with the regenerative state. This was linked to the ability of AD80 to suppress the

GDNF response in cultured A_{undiff} by inhibiting RET and other kinases (**Fig. 7a** and **Supplementary Fig. 8c**) (Dar et al., 2012). Strikingly, AD80 treatment *in vivo* suppressed expression of important markers of regenerative (and homeostatic) SSCs including $GFR\alpha 1$ and EOMES at D10 after BU (**Fig. 7f** and **Supplementary Fig. 10a**). Note that $GFR\alpha 1^+$ and EOMES⁺ populations were still present after rapamycin treatment (**Fig. 5d, e** and **Supplementary Fig. 6c**). However, E-Cadherin⁺ A_s and A_{pr} were found in AD80-treated samples at D10 after BU, indicating that A_{undiff} were present (**Fig. 7d, e**). This was confirmed by flow cytometry for PLZF⁺ c-KIT⁻ cells (**Supplementary Fig. 10c**).

In contrast to effects of rapamycin, AD80 treatment inhibited both AKT and mTORC1 signalling in regenerative A_{undiff} as indicated by a switch in FOXO1 localisation to the nucleus and loss of P-S6 immunostaining in E-Cadherin⁺ A_s and A_{pr} cells from D10 samples (**Fig. 7d, e**). The proportion of regenerative A_{undiff} that were KI67⁺ was also reduced by AD80, consistent with suppressed proliferation (**Fig. 7f**). Importantly, we found by flow cytometry analysis of A_{undiff} at D10 after BU that expression of uPAR and EpCAM, key features of regenerative SSCs (RSCs), were substantially inhibited by AD80 (**Fig. 7g**). While this regimen of AD80 treatment did not appear to disrupt FOXM1 expression as effectively as in cultured A_{undiff} (**Fig. 7b** and not shown), the reduction in expression of FOXM1 targets in AD80-treated samples, e.g., CCND1 and uPAR, suggested that FOXM1 activity was disrupted by AD80 *in vivo* (**Fig. 7g, h** and **Supplementary Fig. 10d**) (discussed in **lines 691-701**) (Li et al., 2013; Wang et al., 2001). These data indicate that AD80 strongly suppressed the characteristic features and signalling activity of RSCs *in vivo*, consistent with our *in vitro* data.

Given effectiveness of AD80 in inhibiting the regenerative SSC state, we also assessed what the long-term impact of transient AD80 treatment would be on germline recovery following BU. As a testament to robustness of the rodent germline, spermatogonial recovery was still evident in seminiferous tubules of AD80-treated mice at D30 after BU (**Fig. 7i**). However, while almost 90% of the tubule length in vehicle controls contained SALL4⁺ spermatogonia and active spermatogenesis at D30, only ~60% of the tubule length did in AD80-treated samples (**Fig. 7i**). Therefore, transient AD80 treatment after BU disrupted the RSC population. Interestingly, within spermatogenic patches of AD80-treated mice at D30, $GFR\alpha 1^+$ SSCs were more abundant than in controls and a higher proportion were KI67⁺ (**Fig. 7i** and **Supplementary Fig. 10f**), suggesting a delayed and on-going regenerative response as we found after rapamycin treatment (**Fig. 5d, e**). Further, while AD80 depleted $GFR\alpha 1^+$ and EOMES⁺ A_{undiff} at D10, these populations recovered at D30 once treatment had ceased (**Supplementary Fig. 10f, g**) illustrating plasticity of A_{undiff} transcriptional states.

Together with our original observations, these new datasets confirm the critical roles played by growth factor-dependent signalling and downstream target FOXM1 in driving the regenerative capabilities of A_{undiff} *in vivo* and support the validity of our *in vitro* experiments on cultured A_{undiff} .

However, in response to comments about the A_{undiff} culture system from the reviewers, we have added the following statement when describing *in vitro* data of **Fig. 7 (lines 607-609)**:

“We note that differences may exist in the role of signalling pathways in cultured A_{undiff} versus regenerative A_{undiff} *in vivo*, limiting the ability to extrapolate *in vitro* results to the regenerating testis.”

2. Although the replicates seem very similar, sample sizes for some experiments are a bit small, eg, Fig 6h and i, “n = 2 cultures, 2 independent experiments,” What do the error bars on the graphs mean? It is not appropriate to display error bars for n=2.

We thank the reviewer for highlighting these issues with the *in vitro* data. We have now repeated these experiments with 3 independent lines of cultured A_{undiff} (each derived from a different adult mouse; see Methods **lines 985-987**). Graphs show mean values of these biological replicates \pm SEM and significance determined by one-way ANOVA followed by Tukey’s multiple comparisons test. Conclusions from these datasets are unchanged. Additional repeats have been performed for the following experiments: 1) Analysis of effects of different inhibitors on signalling pathway activity and expression of FOXM1 plus downstream targets (CCNB1, BIRC5) in cultured A_{undiff} by western blotting (now **Fig. 7b**); 2) Effects of signalling inhibitors on levels of EpCAM and E-Cadherin cell surface markers on cultured A_{undiff} by flow cytometry (now **Fig. 7c**); 3) Effects of different FOXM1 inhibitors at multiple concentrations on levels of FOXM1 plus downstream targets (CCNB1, CCND1, BIRC5) and PLZF in cultured A_{undiff} by western blotting (now **Supplementary Fig. 7e**).

Previously, we had used 2 independent lines of cultured A_{undiff} and shown results from 1 of 2 rounds of experiments together with mean values and erroneously shown SEM. Statistical analysis had not been performed. All quantified *in vitro* datasets now include 3 biological replicates and graphs show mean values \pm SEM and associated *P* values. We note that all quantified *in vivo* datasets include a minimum of 3 biological replicates per group and mean values \pm SEM are shown.

Minor points:

1. Page 21, line 481, “similarly low numbers of E-cadherin+ cells”. The Ecadherin flow data showing no difference is not convincing given the sample size. I would consider eliminating Supp fig 7.

We agree with the reviewer that the repeated BU dataset is in some respects inconclusive and specific results variable. The reduction in induction of EpCAM in A_{undiff} following repeated vs. single dose BU was consistent between samples and suggested a disrupted regenerative response. However, other

markers of the regenerative response (enhanced abundance of P-S6+ and KI67+ cells within the GFR α 1+ SSC pool), and spermatogonial recovery appeared unaffected.

Considering these comments, we have now removed this section from Results and briefly refer to this dataset in the Discussion as it is still relevant for our discussion of effects of repeated tissue damage on stem cell integrity and role of mTORC1 in this process (**lines 819-832 and Supplementary Fig. 11**). Specifically, Trp63+ stem cells of mouse tracheal epithelium are significantly depleted in response to repeated (3x) rounds of damage induced by SO₂ exposure; an effect linked to repeated activation of mTORC1 during epithelial regeneration (Haller et al., 2017). Given the conserved role of mTORC1 in the A_{undiff} regenerative response, this previous study provided a basis for us to assess whether multiple (3x) rounds of BU treatment would similarly result in SSC depletion and disrupt germline recovery. However, besides reduced induction of EpCAM on regenerative A_{undiff}, SSC depletion was not evident after repeated damage and germline recovery appeared comparable to that of mice receiving a single BU challenge (**Supplementary Fig. 11**). These results highlight relative robustness of the rodent germline to repeated damage and can form the basis for future studies that test the limits of SSC regenerative capacity.

2. The labelling of columns in the supp tables (Eg, supp Table 1) is not clear. I can't tell what they all mean.

We have revised column labelling in all Supplementary Tables to improve clarity.

3. Fig 6: They should provide supp figures showing expanded images of FOXM1 staining and negative controls for primary antibody for Foxm1.

We now show expanded images of control and D10 BU-treated testis sections immunostained (IF) for FOXM1 plus PLZF to identify spermatogonial populations (**Supplementary Fig. 7b**). Sections stained with secondary antibody alone generated no specific staining (not shown). Based on our IF analysis, nuclear FOXM1 is detected at multiple stages of spermatogenesis including in sets of mitotic, meiotic, and post-meiotic germ cells (**Supplementary Fig. 7b**). This pattern is expected given its role as a key cell cycle regulator (presumably mitotic and meiotic) (Laoukili et al., 2005), and agrees with the detection of *Foxm1* transcript at multiple spermatogenic stages in published single cell datasets from mouse testis (**Rebuttal Fig. 1**) (Hermann et al., 2018). FOXM1 protein stability is also controlled through distinct mechanisms, e.g., by interaction with deubiquitinase USP21 and phosphorylation by cyclin D-CDK4/6 (Anders et al., 2011; Arceci et al., 2019), indicating additional levels of control of FOXM1 expression besides transcriptional.

Importantly, subsets of FOXM1 positive and negative spermatogonia were evident within the PLZF+ populations of control and BU-treated mice (**Supplementary Fig. 7b**), supporting antibody specificity in this application and our data demonstrating that an increased proportion of GFR α 1+ SSCs are FOXM1+ during regeneration (**Fig. 6c**). We note that this increase in FOXM1 during regeneration correlated with pathway analysis of RNA-Seq data and increased expression of a panel of FOXM1 target genes in isolated regenerative A_{undiff} (**Fig. 6a, b**). Interestingly,

Foxm1 is expressed as distinct isoforms through alternative splicing (Ma et al., 2005), and while our semi-quantitative RT-PCR analysis indicated that both predominant isoforms are detected in adult testis (*Foxm1b* and *Foxm1c*), cultured A_{undiff} predominantly expressed *Foxm1c* (**Supplementary Fig. 7f**). Germ cells at different spermatogenic stages may therefore rely on different FOXM1 isoforms, which are functionally distinct (Ma et al., 2005). The FOXM1 antibody we use is raised against a C-terminal fusion protein and is predicted to react with both b and c isoforms (Proteintech 13147-1-AP).

4. It would be helpful to provide a table with a list of the inhibitors, concentrations used and reported IC₅₀ values for the specific targets (from the literature). Otherwise it is difficult for the reader to interpret the data without looking up the background literature for each inhibitor. The labelling of Supp Fig 8j is confusing because the names of the inhibitors are not given, except for torin, so one has to go to the methods section.

We now provide a list of inhibitors we have used in the second sheet of **Supplementary Table 6** alongside antibodies. This table includes compound and brand names of inhibitors (if applicable), known targets, reported IC₅₀ values, concentrations used for *in vitro* studies (dose for *in vivo* studies is included in the Methods **lines 863-877**), commercial source and catalogue number. Further, we have re-labelled **Supplementary Fig. 8f** (previously 8j) to include both specific target and name of the inhibitor used.

5. Page 10, DEGs in response to GDNF: it is not clear where the 189 in “54 out of 189” comes from or the 71 in 28 out of 71

We apologise for this confusion and have re-worded the relevant sentence for clarity. GDNF responsive genes in A_{undiff} were previously identified from microarray analysis of cultured A_{undiff} subjected to GDNF withdrawal (Oatley et al., 2006). 189 unique genes were identified in this study that were downregulated in cultured A_{undiff} deprived of GDNF (i.e., stimulated by GDNF), while 71 genes were upregulated (repressed by GDNF). Note that the microarray results table in the original publication lists some genes under multiple probe sets (Oatley et al., 2006). Non-validated mRNA sequences listed in this table were also excluded from the comparison. Additional details of this analysis are now included in the Methods **lines 898-899**. We assessed how many of these GDNF responsive genes were significantly altered in regenerative A_{undiff} from our RNA-Seq analysis to provide evidence for stimulation by GDNF (**Fig. 1h**). The sentence now reads (**lines 196-199**): “Importantly, comparison of identified DEGs to genes reported to be responsive to GDNF in cultured A_{undiff} ⁴⁰ revealed that regenerative A_{undiff} typically upregulated genes stimulated by GDNF (54 out of 189, 29% of GDNF-responsive; $P < 2.02E-13$) and downregulated genes suppressed by GDNF (28 out of 71, 39% of GDNF-repressive; $P < 2.99E-11$) (Fig. 1h and Supplementary Table 1).”

6. Page 13:” Functional relevance of cells in cluster 3 was unclear and so were termed undefined A_{undiff} .” Can the authors speculate at all? Are they even A_{undiff} or contaminants?

We have investigated identity of the cluster 3 population from our scRNA-Seq analysis in more detail. Importantly, cells of this cluster expressed multiple markers of A_{undiff} , e.g., *Zbtb16/Plzf*, *Sall4*, *Foxo1*, plus germ cell marker *Ddx4* at comparable levels as the other clusters (**Supplementary Fig. 3a**); supporting an A_{undiff} identity and consistent with the fact that cells for analysis were sorted according to A_{undiff} markers (E-Cadherin⁺ α 6-integrin⁺ c-KIT⁻) (La et al., 2018a). Interestingly, this cluster was positive for expression of *Ldhc* and *Meig1*, which are predominantly expressed at meiotic and post-meiotic spermatogenic stages (**Supplementary Fig. 3c**) (Goldberg et al., 2010; Li et al., 2015). However, according to published single cell datasets from mouse testis (Hermann et al., 2018), the transcript of these genes is detected in spermatogonia (**Rebuttal Fig. 2**). Gene ontology analysis of cluster 3 generated terms associated with spermatid development and SCENIC analysis identified regulons linked with spermiogenesis, e.g., CREM (**Fig. 3c, d**) (Nantel et al., 1996). However, low numbers of contaminating spermatids were identified and removed from our datasets based on *Spem1* expression (**Supplementary Fig. 12c, d and lines 923-927**) (Zheng et al., 2007). Transcripts of spermiogenesis regulators such as CREM are also detectable in spermatogonia (**Rebuttal Fig. 2**).

Given ambiguities regarding identity of cluster 3 but multiple lines of evidence indicating they are of A_{undiff} origin, we have retained the cells in our analysis and refer to them as “undefined A_{undiff} ”. Description of this cluster in the Results has now been modified to the following (lines 304-309):

“Surprisingly, cluster 3 cells were characterised by detectable expression of genes involved at late spermatogenic stages, including *Ldhc* and *Meig1* (Supplementary Fig. 3c)⁵. The physiological relevance of cells in cluster 3 was unclear although they expressed undifferentiated spermatogonial markers (*Zbtb16*, *Sall4*, *Foxo1*) and germ cell marker *Ddx4* as expected and were therefore termed undefined A_{undiff} (Supplementary Fig. 3a).”

Further, we now include additional detailed information on the quality control, normalisation, and filtering steps of our scRNA-Seq analysis in the Methods (lines 901-959). The following QC information is now included:

1) Sequencing metrics from the Cell Ranger pipeline in **Supplementary Table 7**; 2) Violin plots showing QC features from the Seurat analysis for each of the experimental replicates (**Supplementary Fig. 12a**); 3) UMAP plots that illustrate the Seurat analysis pipeline including dataset integration plus alignment and initial clustering following filtering steps to remove low-quality cells (**Supplementary Fig. 12b, c**); 4) Marker gene expression used to define cell types in the initial clustering analysis (**Supplementary Fig. 12d**); 5) Cell numbers from the initial population clustering and re-clustering following exclusion of low numbers of contaminating cells (**Supplementary Table 7** and lines 932-935); 6) Clustering analysis of individual samples to illustrate consistencies between experimental repeats, absence of significant batch effects and the relative proportions of cells in each population (**Supplementary Fig. 12e, f**).

Note that **95.7%** and **94.5%** of cells from control and BU-treated samples respectively are within identified spermatogonial clusters, indicating a high degree of purity of sorted cell preparations and supporting validity of bulk RNA-Seq (**Fig. 1 and Supplementary Table 7**). Based on gene expression, the small clusters of contaminating cells were spermatids (*Spem1+*), peritubular myoid (*Acta2+*) and innate lymphoid cells (*Cd74+*, *Cnmd+*) (**Supplementary Fig. 12d**) (Tan et al., 2020). Note that one spermatogonial cluster was positive for *Kit* expression despite cells being negative for c-KIT by flow cytometry (**Fig. 1c and Supplementary Figure 12d**), presumably representing A_{undiff} just initiating differentiation. *Kit+* spermatogonia were therefore excluded from subsequent analysis of scRNA-Seq datasets (see **lines 923-926**) although these cells would contribute to gene expression in the bulk analysis, highlighting the need for comparison of both bulk and single cell analysis.

7. Page 22: Why doesn't FDI-6 reduce *Foxm1* and targets?

FDI-6 was identified from a screen of compounds that selectively blocked the binding of FOXM1 to DNA response elements and inhibited expression of FOXM1 targets in MCF-7 and other cancer cell lines (Gormally et al., 2014). Despite evidence that FOXM1 binds its own promoter and auto-induces transcription (Halasi and Gartel, 2009), FDI-6 was reported to reduce chromatin-associated FOXM1 without affecting global FOXM1 levels in MCF-7 cells (Gormally et al., 2014). Others have demonstrated that FOXM1 inhibition with thiostrepton reduces FOXM1 mRNA and protein but questioned whether FOXM1 directly regulates its own promoter (Hegde et al., 2011). Importantly, treating the MCF-7 breast cancer line with FDI-6 at 20 μ M resulted in widespread disruption of FOXM1 target gene expression while avoiding overt toxicity and cell death (Ziegler et al., 2019).

When we treated cultured A_{undiff} with comparable concentrations of FDI-6 (5–20 μ M), expression of the majority of FOXM1 targets and *Foxm1* itself were mostly unaltered (**Fig. 6d and Supplementary Fig. 7e**). However, levels of CCND1, a direct FOXM1 target identified from studies in hepatocytes, were significantly reduced (**Supplementary Fig. 7e**). Our results indicated that FDI-6 treatment resulted in some modulation of FOXM1 function but that inhibitory effects on FOXM1 targets might be cell type-dependent. In contrast, thiostrepton inhibited expression of multiple FOXM1 targets in A_{undiff} and disrupted features of the regenerative response *in vivo* (**Fig. 6d, g-i**).

Given that FDI-6 displayed limited efficacy in inhibiting FOXM1 in A_{undiff} , we also tested an additional small molecule inhibitor, RCM-1, which reportedly blocks FOXM1 effectively *in vitro* and *in vivo* by preventing FOXM1 nuclear localisation and promoting its degradation (Shukla et al., 2019; Sun et al., 2017). However, RCM-1 failed to reduce expression of *Foxm1* or key targets in cultured A_{undiff} when used within previously reported concentration ranges (**Supplementary Fig. 7d**).

(Sun et al., 2017). We concluded that the efficacy of distinct FOXM1 inhibitors is cell-type dependent (**lines 521-525**), and our data supports use of thiostrepton for FOXM1 inhibition in A_{undiff} .

8. Supp Fig 6b and legend – labelling/description regarding Epcam data is confusing

We have revised layout of this Supplementary Figure and checked the legend to improve clarity.

9. There are a number of minor word omissions/ typos (eg, Fig 6f graph axis).

We have carefully checked the revised manuscript text and figures for spelling and other errors.

REFERENCES

- Aibar, S., Gonzalez-Blas, C.B., Moerman, T., Huynh-Thu, V.A., Imrichova, H., Hulselmans, G., Rambow, F., Marine, J.C., Geurts, P., Aerts, J., *et al.* (2017). SCENIC: single-cell regulatory network inference and clustering. *Nat Methods* *14*, 1083-1086.
- Aloisio, G.M., Nakada, Y., Saatcioglu, H.D., Pena, C.G., Baker, M.D., Tarnawa, E.D., Mukherjee, J., Manjunath, H., Bugde, A., Sengupta, A.L., *et al.* (2014). PAX7 expression defines germline stem cells in the adult testis. *J Clin Invest* *124*, 3929-3944.
- Anders, L., Ke, N., Hydbring, P., Choi, Y.J., Widlund, H.R., Chick, J.M., Zhai, H., Vidal, M., Gygi, S.P., Braun, P., *et al.* (2011). A systematic screen for CDK4/6 substrates links FOXM1 phosphorylation to senescence suppression in cancer cells. *Cancer Cell* *20*, 620-634.
- Arceci, A., Bonacci, T., Wang, X., Stewart, K., Damrauer, J.S., Hoadley, K.A., and Emanuele, M.J. (2019). FOXM1 Deubiquitination by USP21 Regulates Cell Cycle Progression and Paclitaxel Sensitivity in Basal-like Breast Cancer. *Cell Rep* *26*, 3076-3086 e3076.
- Bergen, V., Lange, M., Peidli, S., Wolf, F.A., and Theis, F.J. (2020). Generalizing RNA velocity to transient cell states through dynamical modeling. *Nat Biotechnol.*
- Bhang, D.H., Kim, B.J., Kim, B.G., Schadler, K., Baek, K.H., Kim, Y.H., Hsiao, W., Ding, B.S., Rafii, S., Weiss, M.J., *et al.* (2018). Testicular endothelial cells are a critical population in the germline stem cell niche. *Nat Commun* *9*, 4379.
- Branney, P.A., Faas, L., Steane, S.E., Pownall, M.E., and Isaacs, H.V. (2009). Characterisation of the fibroblast growth factor dependent transcriptome in early development. *PLoS One* *4*, e4951.
- Bucci, L.R., and Meistrich, M.L. (1987). Effects of busulfan on murine spermatogenesis: cytotoxicity, sterility, sperm abnormalities, and dominant lethal mutations. *Mutat Res* *176*, 259-268.
- Buchner, M., Park, E., Geng, H., Klemm, L., Flach, J., Passegue, E., Schjerven, H., Melnick, A., Paietta, E., Kopanja, D., *et al.* (2015). Identification of FOXM1 as a therapeutic target in B-cell lineage acute lymphoblastic leukaemia. *Nat Commun* *6*, 6471.
- Busada, J.T., Niedenberger, B.A., Velte, E.K., Keiper, B.D., and Geyer, C.B. (2015). Mammalian target of rapamycin complex 1 (mTORC1) Is required for mouse spermatogonial differentiation in vivo. *Dev Biol* *407*, 90-102.
- Carrieri, C., Comazzetto, S., Grover, A., Morgan, M., Bunes, A., Nerlov, C., and O'Carroll, D. (2017). A transit-amplifying population underpins the efficient regenerative capacity of the testis. *J Exp Med* *214*, 1631-1641.
- Cui, J., Xia, T., Xie, D., Gao, Y., Jia, Z., Wei, D., Wang, L., Huang, S., Quan, M., and Xie, K. (2016). HGF/Met and FOXM1 form a positive feedback loop and render pancreatic cancer cells resistance to Met inhibition and aggressive phenotypes. *Oncogene* *35*, 4708-4718.
- Dar, A.C., Das, T.K., Shokat, K.M., and Cagan, R.L. (2012). Chemical genetic discovery of targets and anti-targets for cancer polypharmacology. *Nature* *486*, 80-84.
- Eijkelenboom, A., and Burgering, B.M. (2013). FOXOs: signalling integrators for homeostasis maintenance. *Nat Rev Mol Cell Biol* *14*, 83-97.
- Gely-Pernot, A., Raverdeau, M., Celebi, C., Dennefeld, C., Feret, B., Klopfenstein, M., Yoshida, S., Ghyselinck, N.B., and Mark, M. (2012). Spermatogonia differentiation requires retinoic acid receptor gamma. *Endocrinology* *153*, 438-449.
- Goertz, M.J., Wu, Z., Gallardo, T.D., Hamra, F.K., and Castrillon, D.H. (2011). Foxo1 is required in mouse spermatogonial stem cells for their maintenance and the initiation of spermatogenesis. *J Clin Invest* *121*, 3456-3466.

- Goldberg, E., Eddy, E.M., Duan, C., and Odet, F. (2010). LDHC: the ultimate testis-specific gene. *J Androl* *31*, 86-94.
- Gormally, M.V., Dexheimer, T.S., Marsico, G., Sanders, D.A., Lowe, C., Matak-Vinkovic, D., Michael, S., Jadhav, A., Rai, G., Maloney, D.J., *et al.* (2014). Suppression of the FOXM1 transcriptional programme via novel small molecule inhibition. *Nat Commun* *5*, 5165.
- Halasi, M., and Gartel, A.L. (2009). A novel mode of FoxM1 regulation: positive auto-regulatory loop. *Cell Cycle* *8*, 1966-1967.
- Haller, S., Kapuria, S., Riley, R.R., O'Leary, M.N., Schreiber, K.H., Andersen, J.K., Melov, S., Que, J., Rando, T.A., Rock, J., *et al.* (2017). mTORC1 Activation during Repeated Regeneration Impairs Somatic Stem Cell Maintenance. *Cell Stem Cell* *21*, 806-818 e805.
- Hegde, N.S., Sanders, D.A., Rodriguez, R., and Balasubramanian, S. (2011). The transcription factor FOXM1 is a cellular target of the natural product thioestron. *Nat Chem* *3*, 725-731.
- Hermann, B.P., Cheng, K., Singh, A., Roa-De La Cruz, L., Mutoji, K.N., Chen, I.C., Gildersleeve, H., Lehle, J.D., Mayo, M., Westernstroer, B., *et al.* (2018). The Mammalian Spermatogenesis Single-Cell Transcriptome, from Spermatogonial Stem Cells to Spermatids. *Cell Rep* *25*, 1650-1667 e1658.
- Hobbs, R.M., La, H.M., Makela, J.A., Kobayashi, T., Noda, T., and Pandolfi, P.P. (2015). Distinct germline progenitor subsets defined through Tsc2-mTORC1 signaling. *EMBO Rep* *16*, 467-480.
- Hobbs, R.M., Seandel, M., Falcatori, I., Rafii, S., and Pandolfi, P.P. (2010). Plzf regulates germline progenitor self-renewal by opposing mTORC1. *Cell* *142*, 468-479.
- Ikami, K., Tokue, M., Sugimoto, R., Noda, C., Kobayashi, S., Hara, K., and Yoshida, S. (2015). Hierarchical differentiation competence in response to retinoic acid ensures stem cell maintenance during mouse spermatogenesis. *Development* *142*, 1582-1592.
- Ishii, K., Kanatsu-Shinohara, M., Toyokuni, S., and Shinohara, T. (2012). FGF2 mediates mouse spermatogonial stem cell self-renewal via upregulation of Etv5 and Bcl6b through MAP2K1 activation. *Development* *139*, 1734-1743.
- Kalaitzidis, D., Sykes, S.M., Wang, Z., Punt, N., Tang, Y., Ragu, C., Sinha, A.U., Lane, S.W., Souza, A.L., Clish, C.B., *et al.* (2012). mTOR complex 1 plays critical roles in hematopoiesis and Pten-loss-evoked leukemogenesis. *Cell Stem Cell* *11*, 429-439.
- Kanatsu-Shinohara, M., Ogonuki, N., Inoue, K., Miki, H., Ogura, A., Toyokuni, S., and Shinohara, T. (2003). Long-term proliferation in culture and germline transmission of mouse male germline stem cells. *Biol Reprod* *69*, 612-616.
- Kitadate, Y., Jorg, D.J., Tokue, M., Maruyama, A., Ichikawa, R., Tsuchiya, S., Segi-Nishida, E., Nakagawa, T., Uchida, A., Kimura-Yoshida, C., *et al.* (2019). Competition for Mitogens Regulates Spermatogenic Stem Cell Homeostasis in an Open Niche. *Cell Stem Cell* *24*, 79-92 e76.
- Kubota, H., Avarbock, M.R., and Brinster, R.L. (2004). Growth factors essential for self-renewal and expansion of mouse spermatogonial stem cells. *Proc Natl Acad Sci U S A* *101*, 16489-16494.
- La, H.M., Chan, A.L., Legrand, J.M.D., Rossello, F.J., Gangemi, C.G., Papa, A., Cheng, Q., Morand, E.F., and Hobbs, R.M. (2018a). GILZ-dependent modulation of mTORC1 regulates spermatogonial maintenance. *Development* *145*.
- La, H.M., and Hobbs, R.M. (2019). Mechanisms regulating mammalian spermatogenesis and fertility recovery following germ cell depletion. *Cell Mol Life Sci*.
- La, H.M., Makela, J.A., Chan, A.L., Rossello, F.J., Nefzger, C.M., Legrand, J.M.D., De Seram, M., Polo, J.M., and Hobbs, R.M. (2018b). Identification of dynamic undifferentiated cell states within the male germline. *Nat Commun* *9*, 2819.

- Laoukili, J., Kooistra, M.R., Bras, A., Kauw, J., Kerkhoven, R.M., Morrison, A., Clevers, H., and Medema, R.H. (2005). FoxM1 is required for execution of the mitotic programme and chromosome stability. *Nat Cell Biol* 7, 126-136.
- Laplane, M., and Sabatini, D.M. (2009). mTOR signaling at a glance. *J Cell Sci* 122, 3589-3594.
- Laplane, M., and Sabatini, D.M. (2012). mTOR signaling in growth control and disease. *Cell* 149, 274-293.
- Lee, J., Kanatsu-Shinohara, M., Inoue, K., Ogonuki, N., Miki, H., Toyokuni, S., Kimura, T., Nakano, T., Ogura, A., and Shinohara, T. (2007). Akt mediates self-renewal division of mouse spermatogonial stem cells. *Development* 134, 1853-1859.
- Lee, J.Y., Nakada, D., Yilmaz, O.H., Tothova, Z., Joseph, N.M., Lim, M.S., Gilliland, D.G., and Morrison, S.J. (2010). mTOR activation induces tumor suppressors that inhibit leukemogenesis and deplete hematopoietic stem cells after Pten deletion. *Cell Stem Cell* 7, 593-605.
- Li, D., Wei, P., Peng, Z., Huang, C., Tang, H., Jia, Z., Cui, J., Le, X., Huang, S., and Xie, K. (2013). The critical role of dysregulated FOXM1-PLAUR signaling in human colon cancer progression and metastasis. *Clin Cancer Res* 19, 62-72.
- Li, W., Tang, W., Teves, M.E., Zhang, Z., Zhang, L., Li, H., Archer, K.J., Peterson, D.L., Williams, D.C., Jr., Strauss, J.F., 3rd, *et al.* (2015). A MEIG1/PACRG complex in the manchette is essential for building the sperm flagella. *Development* 142, 921-930.
- Ma, R.Y., Tong, T.H., Cheung, A.M., Tsang, A.C., Leung, W.Y., and Yao, K.M. (2005). Raf/MEK/MAPK signaling stimulates the nuclear translocation and transactivating activity of FOXM1c. *J Cell Sci* 118, 795-806.
- Maekawa, M., Kamimura, K., and Nagano, T. (1996). Peritubular myoid cells in the testis: their structure and function. *Arch Histol Cytol* 59, 1-13.
- Masaki, K., Sakai, M., Kuroki, S., Jo, J.I., Hoshina, K., Fujimori, Y., Oka, K., Amano, T., Yamanaka, T., Tachibana, M., *et al.* (2018). FGF2 Has Distinct Molecular Functions from GDNF in the Mouse Germline Niche. *Stem Cell Reports* 10, 1782-1792.
- McAninch, D., Makela, J.A., La, H.M., Hughes, J.N., Lovell-Badge, R., Hobbs, R.M., and Thomas, P.Q. (2020). SOX3 promotes generation of committed spermatogonia in postnatal mouse testes. *Sci Rep* 10, 6751.
- Meistrich, M.L. (2013). Effects of chemotherapy and radiotherapy on spermatogenesis in humans. *Fertil Steril* 100, 1180-1186.
- Meng, X., Lindahl, M., Hyvonen, M.E., Parvinen, M., de Rooij, D.G., Hess, M.W., Raatikainen-Ahokas, A., Sainio, K., Rauvala, H., Lakso, M., *et al.* (2000). Regulation of cell fate decision of undifferentiated spermatogonia by GDNF. *Science* 287, 1489-1493.
- Modi, H., Jacovetti, C., Tarussio, D., Metref, S., Madsen, O.D., Zhang, F.P., Rantakari, P., Poutanen, M., Nef, S., Gorman, T., *et al.* (2015). Autocrine Action of IGF2 Regulates Adult beta-Cell Mass and Function. *Diabetes* 64, 4148-4157.
- Mulvihill, M.J., Cooke, A., Rosenfeld-Franklin, M., Buck, E., Foreman, K., Landfair, D., O'Connor, M., Pirritt, C., Sun, Y., Yao, Y., *et al.* (2009). Discovery of OSI-906: a selective and orally efficacious dual inhibitor of the IGF-1 receptor and insulin receptor. *Future Med Chem* 1, 1153-1171.
- Nakagawa, T., Nabeshima, Y., and Yoshida, S. (2007). Functional identification of the actual and potential stem cell compartments in mouse spermatogenesis. *Dev Cell* 12, 195-206.
- Nakagawa, T., Sharma, M., Nabeshima, Y., Braun, R.E., and Yoshida, S. (2010). Functional hierarchy and reversibility within the murine spermatogenic stem cell compartment. *Science* 328, 62-67.

- Nantel, F., Monaco, L., Foulkes, N.S., Masquilier, D., LeMeur, M., Henriksen, K., Dierich, A., Parvinen, M., and Sassone-Corsi, P. (1996). Spermiogenesis deficiency and germ-cell apoptosis in CREM-mutant mice. *Nature* *380*, 159-162.
- O'Shaughnessy, P.J., Hu, L., and Baker, P.J. (2008). Effect of germ cell depletion on levels of specific mRNA transcripts in mouse Sertoli cells and Leydig cells. *Reproduction* *135*, 839-850.
- Oatley, J.M., Avarbock, M.R., and Brinster, R.L. (2007). Glial cell line-derived neurotrophic factor regulation of genes essential for self-renewal of mouse spermatogonial stem cells is dependent on Src family kinase signaling. *J Biol Chem* *282*, 25842-25851.
- Oatley, J.M., Avarbock, M.R., Telaranta, A.I., Fearon, D.T., and Brinster, R.L. (2006). Identifying genes important for spermatogonial stem cell self-renewal and survival. *Proc Natl Acad Sci U S A* *103*, 9524-9529.
- Plenker, D., Riedel, M., Bragelmann, J., Dammert, M.A., Chauhan, R., Knowles, P.P., Lorenz, C., Keul, M., Buhrmann, M., Pagel, O., *et al.* (2017). Drugging the catalytically inactive state of RET kinase in RET-rearranged tumors. *Sci Transl Med* *9*.
- Rodgers, J.T., King, K.Y., Brett, J.O., Cromie, M.J., Charville, G.W., Maguire, K.K., Brunson, C., Mastey, N., Liu, L., Tsai, C.R., *et al.* (2014). mTORC1 controls the adaptive transition of quiescent stem cells from G0 to G(Alert). *Nature* *510*, 393-396.
- Rothschild, G., Sottas, C.M., Kissel, H., Agosti, V., Manova, K., Hardy, M.P., and Besmer, P. (2003). A role for kit receptor signaling in Leydig cell steroidogenesis. *Biol Reprod* *69*, 925-932.
- Sarbassov, D.D., Ali, S.M., Sengupta, S., Sheen, J.H., Hsu, P.P., Bagley, A.F., Markhard, A.L., and Sabatini, D.M. (2006). Prolonged rapamycin treatment inhibits mTORC2 assembly and Akt/PKB. *Mol Cell* *22*, 159-168.
- Sharma, M., Srivastava, A., Fairfield, H.E., Bergstrom, D., Flynn, W.F., and Braun, R.E. (2019). Identification of EOMES-expressing spermatogonial stem cells and their regulation by PLZF. *Elife* *8*.
- Sheng, Y., Yu, C., Liu, Y., Hu, C., Ma, R., Lu, X., Ji, P., Chen, J., Mizukawa, B., Huang, Y., *et al.* (2020). FOXM1 regulates leukemia stem cell quiescence and survival in MLL-rearranged AML. *Nat Commun* *11*, 928.
- Shirakawa, T., Yaman-Deveci, R., Tomizawa, S., Kamizato, Y., Nakajima, K., Sone, H., Sato, Y., Sharif, J., Yamashita, A., Takada-Horisawa, Y., *et al.* (2013). An epigenetic switch is crucial for spermatogonia to exit the undifferentiated state toward a Kit-positive identity. *Development* *140*, 3565-3576.
- Shukla, S., Milewski, D., Pradhan, A., Rama, N., Rice, K., Le, T., Flick, M.J., Vaz, S., Zhao, X., Setchell, K.D., *et al.* (2019). The FOXM1 Inhibitor RCM-1 Decreases Carcinogenesis and Nuclear beta-Catenin. *Mol Cancer Ther* *18*, 1217-1229.
- Sun, L., Ren, X., Wang, I.C., Pradhan, A., Zhang, Y., Flood, H.M., Han, B., Whitsett, J.A., Kalin, T.V., and Kalinichenko, V.V. (2017). The FOXM1 inhibitor RCM-1 suppresses goblet cell metaplasia and prevents IL-13 and STAT6 signaling in allergen-exposed mice. *Sci Signal* *10*.
- Suzuki, S., McCarrey, J.R., and Hermann, B.P. (2021). An mTORC1-dependent switch orchestrates the transition between mouse spermatogonial stem cells and clones of progenitor spermatogonia. *Cell Rep* *34*, 108752.
- Takubo, K., Ohmura, M., Azuma, M., Nagamatsu, G., Yamada, W., Arai, F., Hirao, A., and Suda, T. (2008). Stem cell defects in ATM-deficient undifferentiated spermatogonia through DNA damage-induced cell-cycle arrest. *Cell Stem Cell* *2*, 170-182.
- Tan, K., Song, H.W., and Wilkinson, M.F. (2020). Single-cell RNAseq analysis of testicular germ and somatic cell development during the perinatal period. *Development* *147*.

- Tokuda, M., Kadokawa, Y., Kurahashi, H., and Marunouchi, T. (2007). CDH1 is a specific marker for undifferentiated spermatogonia in mouse testes. *Biol Reprod* 76, 130-141.
- Wang, M.J., Chen, F., Liu, Q.G., Liu, C.C., Yao, H., Yu, B., Zhang, H.B., Yan, H.X., Ye, Y., Chen, T., *et al.* (2018). Insulin-like growth factor 2 is a key mitogen driving liver repopulation in mice. *Cell Death Dis* 9, 26.
- Wang, S., Wang, X., Wu, Y., and Han, C. (2015). IGF-1R signaling is essential for the proliferation of cultured mouse spermatogonial stem cells by promoting the G2/M progression of the cell cycle. *Stem Cells Dev* 24, 471-483.
- Wang, X., Kiyokawa, H., Dennewitz, M.B., and Costa, R.H. (2002). The Forkhead Box m1b transcription factor is essential for hepatocyte DNA replication and mitosis during mouse liver regeneration. *Proc Natl Acad Sci U S A* 99, 16881-16886.
- Wang, X., Quail, E., Hung, N.J., Tan, Y., Ye, H., and Costa, R.H. (2001). Increased levels of forkhead box M1B transcription factor in transgenic mouse hepatocytes prevent age-related proliferation defects in regenerating liver. *Proc Natl Acad Sci U S A* 98, 11468-11473.
- Zheng, H., Stratton, C.J., Morozumi, K., Jin, J., Yanagimachi, R., and Yan, W. (2007). Lack of Spem1 causes aberrant cytoplasm removal, sperm deformation, and male infertility. *Proc Natl Acad Sci U S A* 104, 6852-6857.
- Ziegler, Y., Laws, M.J., Sanabria Guillen, V., Kim, S.H., Dey, P., Smith, B.P., Gong, P., Bindman, N., Zhao, Y., Carlson, K., *et al.* (2019). Suppression of FOXM1 activities and breast cancer growth in vitro and in vivo by a new class of compounds. *NPJ Breast Cancer* 5, 45.

REVIEWER COMMENTS

Reviewer #1 (Remarks to the Author):

The authors have done an excellent job of revising their MS in terms of new data. This is now a very comprehensive and compelling piece of work. The remaining concerns have to do with writing. First, some of the sections are still not very introduced, particularly for non-SSC aficionados. Second, it is suggested to make some organizational changes. For example, in this Reviewer's opinion, it is better to describe the scRNAseq-defined cell clusters before discussing the BU-altered cell states (in other words, largely swap the "Heterogeneity and cellular dynamics of regenerating Aundiff" section with the "Distinct cellular state of regenerative GFRa1 + spermatogonia" section). Also, the "Multikinase inhibitor AD80 suppresses molecular features and function of RSCs" seems out of place at the end of the Results section. The subject relates to growth factors, so this section might go better after the "Enhanced growth factor-dependent signaling in regenerative Aundiff" section. A final comment: the rebuttal was far too long and was also internally redundant!

Reviewer #2 (Remarks to the Author):

During this round of revision, I acknowledge that the authors have made huge efforts in conducting additional set of experiments, as well as rephrasing the documents. Although they have responded the points that I raised in the initial review, these revisions might not necessarily lead to, or help for, clarification of the issues and achievement of unambiguous conclusion. Indeed, the revised version still include a great amount of data and relevant information (even much more than the original version), many of which appear not to have direct relationship with, or do not provide unambiguous supports for, their main conclusion. I think that this is a critical drawback of this manuscript, and, honestly it is very difficult for me to follow the descriptions and to evaluate if the conclusions are supported appropriately. In particular, the current manuscript includes 23 page-long result section; to my view, this may be 2.5 to 3 times longer than average full paper format. Before considering publication, I think that it is essential for the authors to select really essential data set to present in front, i.e., in the main text, and I would like to see which data the author consider to be essential to support what conclusion. In the following, I will provide some comments, but please forgive me if I have not captured the data or descriptions appropriately, due to the narrative that appears quite intricate to me.

- 1) In response to my original comments on the presumptively differential effects of growth factors including insulin, I acknowledge that the newly added data are informative. However, the descriptions are complicated and lengthy, very hard to follow. Please digest the results and describe the conclusions in a concise fashion.
- 2) In relation with this, the revised version now includes many in vitro data (which increases the total length greatly). Though in vitro system is effective to investigate the effect of extracellular growth factors, it is confusing that the authors can (do) not conclude about the rationale between in vitro situation and regeneration in vivo. Given that this study focuses on the difference between homeostasis and regeneration, the readers including me will be confused whether in vitro study is a mimic of regeneration or homeostasis, of whether this is unrelated to these situations. Though they have pointed several common features between in vitro and regeneration, they are not conclusive, leaving the readers confused. One (extreme) idea is to move the in vitro data to supplementary, mentioned only briefly in the main text.
- 3) Throughout the paper, the authors use the term "RSCs" very often, However, this is really confusing for me, given the heterogeneous nature of undifferentiated spermatogonia both in homeostasis and regeneration. On the scRNA-seq clustering, they define some states/clusters such as primitive or proliferative SSCs, and they showed that the difference between homeostasis and regeneration is represented by the change of the balance between such clusters. Therefore, I think that "RSCs" do not define a particular cell state revealed by clustering analyses. Then, are the "RSCs" heterogeneous population with different proportion of subclusters compared with homeostasis? If so, is the use of "regenerative Stem Cells" for a mixed population consistent with the use of SSCs for particular states? Further, the author also use "regenerative Aundiff", making me more confused. To me "regenerative Aundiff" sounds more comfortable, given the inclusion of

the concept of heterogeneity in this term.

4) Although it may not be discussed so clearly related to the use of terms like RSCs, I think that it is an essential question if the DEGs and differentially activated pathways which have been detected between homeostasis and regeneration represent the different proportions of Aundiff clusters, or their expression levels are altered within a particular cluster. However, it looks like that the functional analysis have been conducted without considering the subclusters (again, I apologize if I did not capture correctly). For example, there must be multiple state transition events within Aundiff population, including those going on opposite directions, it should be dangerous if the authors simply discuss the activation or suppression of particular pathways without considering the relevant cell states. I hope that the roles of genes (and pathways) of interest warrant discussions considering such heterogeneous composition of spermatogonia.

5) It is unfortunate that my suggestion to remove some parts unrelated (or least related) to the main conclusions from the front could not convince the authors (e.g., Lhx1, repeated BU), although they have shortened the relevant statements. Remaining these less important information (though I would not say they are not important and I agree with the explanations by the authors) makes the revised version more complicated, with the addition of much information from new experiments/analyses.

6) In addition to the aforementioned points, I would also suggest to remove the entire section of EpCAM-related issue from the front.

7) Considering the seemingly inconsistent results obtained in revision for multiple inhibitors of FoxM1, the observations obtained from the use of essentially a single inhibitor, thiostrepton, became not fully reliable to draw a conclusion about the roles of FoxM1 in vivo. I think that the roles of FoxM1 may better to be investigated in in vivo and regenerative context in a more reliable (specific) manner, ideally using gene knock out or knock down.

Reviewer #3 (Remarks to the Author):

All the issues raised in the original review have been adequately addressed.

RESPONSE TO THE REVIEWERS

We thank the reviewers for the positive comments on our revised manuscript. To address remaining concerns, we have restructured the manuscript, removed less critical datasets to improve clarity, substantially reduced the length, and included new analysis of scRNA-Seq datasets. A point-by-point response detailing these changes is below. Changes to text are indicated with **yellow highlighting**.

REVIEWER #1

The authors have done an excellent job of revising their MS in terms of new data. This is now a very comprehensive and compelling piece of work. The remaining concerns have to do with writing. First, some of the sections are still not very introduced, particularly for non-SSC aficionados. Second, it is suggested to make some organizational changes. For example, in this Reviewer's opinion, it is better to describe the scRNAseq-defined cell clusters before discussing the BU-altered cell states (in other words, largely swap the "Heterogeneity and cellular dynamics of regenerating Aundiff" section with the "Distinct cellular state of regenerative GFRa1+ spermatogonia" section). Also, the "Multikinase inhibitor AD80 suppresses molecular features and function of RSCs" seems out of place at the end of the Results section. The subject relates to growth factors, so this section might go better after the "Enhanced growth factor-dependent signaling in regenerative Aundiff" section. A final comment: the rebuttal was far too long and was also internally redundant!

We thank the Reviewer for the positive comments and suggestions for improvement. We have changed the structure and formatting of the manuscript in line with these comments as follows:

- a) The clustering analysis of scRNA-Seq data is now described before the BU-altered cell state section. This re-ordering required some adjustment of individual figure panels and text. Clustering analysis is now shown in **Figs. 2, 3** and **Supplementary Figs. 2, 3** while the altered cell state section is described in **Fig. 3** and **Supplementary Figs. 4, 5**. We have also confirmed that similar sets of genes are differentially expressed in the SSC clusters (#0 primitive and #4 proliferative) under regenerative conditions as in the *Gfra1*⁺ and *Eomes*⁺ SSC-enriched fractions, validating our approach (**lines 231-240**). This new analysis is included in **Supplementary Fig. 4b, c**.
- b) The section describing effects of multikinase inhibitor AD80 on regeneration has been moved to after the sections describing growth factor signalling and role of mTORC1 in regeneration. This improves flow of the manuscript and required some changes to text and figure panels. The AD80 dataset is now shown in **Fig. 5** and **Suppl. Fig. 7**. Note that figures describing growth factor signalling and role of mTORC1 in regeneration have been combined into **Fig. 4** in response to Reviewer #2.
- c) We have revised the text extensively to reduce the manuscript length and improve clarity of the Results section and introductory sentences.

REVIEWER #2

During this round of revision, I acknowledge that the authors have made huge efforts in conducting additional set of experiments, as well as rephrasing the documents. Although they have responded the points that I raised in the initial review, these revisions might not necessarily lead to, or help for, clarification of the issues and achievement of unambiguous conclusion. Indeed, the revised version still include a great amount of data and relevant information (even much more than the original version), many of which appear not to have direct relationship with, or do not provide unambiguous supports for, their main conclusion. I think that this is a critical drawback of this manuscript, and, honestly it is very difficult for me to follow the descriptions and to evaluate if the conclusions are supported appropriately. In particular, the current manuscript includes 23 page-long result section; to my view, this may be 2.5 to 3 times longer than average full paper format. Before considering publication, I think that it is essential for the authors to select really essential data set to present in front, i.e., in the main text, and I would like to see which data the author consider to be essential to support what conclusion. In the following, I will provide some comments, but please forgive me if I have not captured the data or descriptions appropriately, due to the narrative that appears quite intricate to me.

We thank the reviewer for suggestions to improve the manuscript. As detailed below, we have now greatly streamlined the manuscript to focus on key results, substantially reduced the text length and clarified the conclusions. The Results section has been reduced from 7700 words in the previous version to just 4494 in the current version. The Introduction and Discussion sections have also been reduced and simplified. The main text has been reduced from almost 11,000 words to 6750. We feel that these changes have improved clarity of the manuscript and strengthened our conclusions.

1) In response to my original comments on the presumptively differential effects of growth factors including insulin, I acknowledge that the newly added data are informative. However, the descriptions are complicated and lengthy, very hard to follow. Please digest the results and describe the conclusions in a concise fashion.

We have streamlined our discussion of the predicted roles of different growth factors in regeneration and have focused on the roles of downstream signalling pathways. Detailed discussion on the potential roles of different growth factors including insulin/IGFs are no longer included. Our discussion is primarily limited to the following observations: **a)** Gene expression changes in regenerative A_{undiff} indicate stimulation by both GDNF and FGFs and literature suggests these factors may regulate overlapping targets (**lines 125-138**); **b)** *In vivo* treatment with AD80 leads to

pronounced disruption of the regenerative response, suggesting a role for GDNF in regeneration given that AD80 is a potent inhibitor of GDNF-dependent signalling (**lines 332-334** and **489-490**). Comparison of the effects of receptor inhibitors on signalling and gene expression in cultured A_{undiff} have been removed from the main figures (now **Supplementary Fig. 9a-e, h, i**).

2) In relation with this, the revised version now includes many in vitro data (which increases the total length greatly). Though in vitro system is effective to investigate the effect of extracellular growth factors, it is confusing that the authors can (do) not conclude about the rationale between in vitro situation and regeneration in vivo. Given that this study focuses on the difference between homeostasis and regeneration, the readers including me will be confused whether in vitro study is a mimic of regeneration or homeostasis, of whether this is unrelated to these situations. Though they have pointed several common features between in vitro and regeneration, they are not conclusive, leaving the readers confused. One (extreme) idea is to move the in vitro data to supplementary, mentioned only briefly in the main text.

As mentioned by the reviewer, we have used the culture system to gain mechanistic insight into growth factor signalling in A_{undiff} and the effects of inhibitors on A_{undiff} function as these analyses are challenging to perform *in vivo*. Key observations have then been confirmed through inhibitor studies *in vivo*. The culture system is commonly used to characterise mechanisms regulating SSC function and a detailed comparison of homeostatic and regenerative A_{undiff} with cultured cells is beyond the study scope. We have briefly highlighted some interesting similarities between regenerative A_{undiff} and cultured cells in a short section in the Discussion (**lines 468-476**), which seems entirely valid. For instance, cytosolic FOXO1 and EpCAM induction are common features and are associated with increased growth factor stimulation as occurs during regeneration and *in vitro* culture. We have not made broader conclusions about this topic. To minimise confusion, we have now moved most of the *in vitro* results to Supplementary (**Supplementary Figs. 6 and 9**) and focused on *in vivo* analyses in the main figures. Out of all the main figures only **Fig. 6d-f** contain data from cultured cells.

3) Throughout the paper, the authors use the term “RSCs” very often, However, this is really confusing for me, given the heterogeneous nature of undifferentiated spermatogonia both in homeostasis and regeneration. On the scRNA-seq clustering, they define some states/clusters such as primitive or proliferative SSCs, and they showed that the difference between homeostasis and regeneration is represented by the change of the balance between such clusters. Therefore, I think that “RSCs” do not define a particular cell state revealed by clustering analyses. Then, are the “RSCs” heterogeneous population with different proportion of subclusters compared with

homeostasis? If so, is the use of “regenerative Stem Cells” for a mixed population consistent with the use of SSCs for particular states? Further, the author also use “regenerative Aundiff”, making me more confused. To me “regenerative Aundiff” sounds more comfortable, given the inclusion of the concept of heterogeneity in this term.

We apologise for the confusing use of different terms for cells mediating regeneration. We now use the term “regenerative A_{undiff}” in most of the manuscript to reflect heterogeneity within this population, particularly when these cells are identified through simple IF analysis, e.g., GFR α 1+. However, when performing scRNA-Seq analysis, we can distinguish SSC subsets present within the heterogenous regenerative A_{undiff} population and in this context refer to “regenerative SSCs”. The term RSC or “repopulating stem cell”, was coined by Dirk de Rooij based on morphological studies to describe A_{undiff} remaining after BU treatment that mediate germline recovery and that resembled A_s and A_{pr}¹. We briefly refer to this study in the Discussion (**lines 452-455**) but have now removed the use of RSC in the manuscript. As discussed in the response below, we provide substantial evidence that regenerative SSCs have a unique functional status compared to homeostatic SSCs.

4) Although it may not be discussed so clearly related to the use of terms like RSCs, I think that it is an essential question if the DEGs and differentially activated pathways which have been detected between homeostasis and regeneration represent the different proportions of Aundiff clusters, or their expression levels are altered within a particular cluster. However, it looks like that the functional analysis have been conducted without considering the subclusters (again, I apologize if I did not capture correctly). For example, there must be multiple state transition events within Aundiff population, including those going on opposite directions, it should be dangerous if the authors simply discuss the activation or suppression of particular pathways without considering the relevant cell states. I hope that the roles of genes (and pathways) of interest warrant discussions considering such heterogeneous composition of spermatogonia.

Analysis of isolated A_{undiff} by bulk RNA-Seq provided insight into changes in composition of the A_{undiff} pool during regeneration, and suggested cellular pathways and factors involved in regeneration (**Fig. 1 and 6a, b**). Our scRNA-Seq data confirmed that composition of the A_{undiff} pool is substantially altered during regeneration and that proliferative SSC states become more abundant while progenitor populations are depleted (**Fig. 2**). To assess whether regenerative SSCs have a distinct functional state, we defined differentially expressed genes (DEGs) from scRNA-Seq datasets in A_{undiff} expressing the SSC-associated genes *Gfra1* and *Eomes*. To support this analysis, we now provide analysis of DEGs in the primitive and proliferative SSC clusters from Seurat analysis from

regenerative and homeostatic A_{undiff} (**Supplementary Fig. 4b and Table 3**). We find substantial overlap in identified DEGs between these various SSC fractions, including induction of cell cycle genes and other genes of interest e.g., *Plaur* and *Igf2* (**Supplementary Fig. 4c and lines 231-240**).

Importantly, pathway analysis on bulk RNA-Seq data suggested that FOXM1 was activated in regenerative A_{undiff} , and we now show that this FOXM1 signature is also found in the SSC-enriched *Gfra1*⁺ fraction and SSC clusters from scRNA-Seq data (**Supplementary Fig. 8d, Table 3 and lines 379-381**). We therefore validate key DEGs and pathways affected in SSCs during regeneration.

From inspection of scRNA-Seq clustering analysis, it might appear that composition of the A_{undiff} pool is changed during regeneration and that functional properties of the various subpopulations are unaffected. However, cells within the same SSC clusters from regenerative and homeostatic samples display changes in gene expression indicating the presence of a unique regenerative state, while transcriptional similarities between these samples are still sufficiently strong to group cells within the same cluster².

5) It is unfortunate that my suggestion to remove some parts unrelated (or least related) to the main conclusions from the front could not convince the authors (e.g., Lhx1, repeated BU), although they have shortened the relevant statements. Remaining these less important information (though I would not say they are not important and I agree with the explanations by the authors) makes the revised version more complicated, with the addition of much information from new experiments/analyses.

We have now removed the *Lhx1* knockout and repeated BU datasets as suggested. In addition, we have removed all data concerning the response of A_{undiff} to the alternative chemotherapeutic agent, cyclophosphamide. As a result, the figures describing growth factor signalling and role of mTORC1 in regeneration have now been combined into a single figure (**Fig. 4**).

6) In addition to the aforementioned points, I would also suggest to remove the entire section of EpCAM-related issue from the front.

The observation that EpCAM is induced in regenerative A_{undiff} is highly relevant for the manuscript given that we identify markers of the regenerative state. Our scRNA-Seq confirms *Epcam* induction in identified SSC clusters plus SSC-enriched *Gfra1*⁺ and *Eomes*⁺ subsets (**Fig. 3a and Supplementary Fig. 4a, b**). EpCAM expression has previously been associated with growth factor stimulation of A_{undiff} and further supports our model of regeneration³. We have streamlined accompanying text describing our *in vivo* observations (**lines 132-138**) but have opted to keep the data in **Fig. 1i**. EpCAM levels are also used to confirm disruption of the regenerative state by AD80

in vivo (**Fig. 5d**), proving an informative marker in this context. However, the regulation of EpCAM expression by growth factor signalling in cultured A_{undiff} has been moved to **Supplementary Fig. 9e**.

7) *Considering the seemingly inconsistent results obtained in revision for multiple inhibitors of FoxM1, the observations obtained from the use of essentially a single inhibitor, thiostrepton, became not fully reliable to draw a conclusion about the roles of FoxM1 in vivo. I think that the roles of FoxM1 may better to be investigated in in vivo and regenerative context in a more reliable (specific) manner, ideally using gene knock out or knock down.*

Thiostrepton has been used in multiple studies to specifically suppress FOXM1 function because it directly binds FOXM1 and inhibits its ability to interact with an array of target genes⁴⁻¹⁰. For example, thiostrepton treatment inhibits leukemia development in mouse models in a manner that recapitulates *Foxm1* deletion^{5,10}. Accordingly, treatment of cultured A_{undiff} with thiostrepton resulted in substantially reduced expression of a panel of FOXM1 targets and FOXM1 itself (**Fig. 6d** and **Supplementary Fig. 8e**). Thiostrepton also efficiently inhibited cell cycle of A_{undiff} (**Fig. 6e**), consistent with the role of FOXM1 in cell cycle progression¹¹ and effects of overexpressing wildtype and dominant negative *Foxm1* constructs in cultured cells (**Fig. 6f**). When used *in vivo* according to an established protocol¹⁰, thiostrepton reduced proliferation of regenerative A_{undiff} as anticipated and inhibited mTORC1 (**Fig. 6g-i**). Positive feedback effects of FOXM1 on growth factor signalling have been described¹², supporting the ability of thiostrepton to inhibit mTORC1 in regenerative A_{undiff} .

We tested additional FOXM1 inhibitors on cultured A_{undiff} , including FDI-6, which is less well-characterised and not recommended for *in vivo* studies¹³. Treatment of A_{undiff} with FDI-6 resulted in more modest effects on FOXM1 function although key downstream targets, including CCND1¹⁴, were substantially reduced (**Supplementary Fig. 8e**). Therefore, the effects of thiostrepton and FDI-6 are not inconsistent, but their efficacy towards FOXM1 in A_{undiff} varies. Thiostrepton and FDI-6 are proposed to bind a common drug-binding pocket in the FOXM1 DNA binding domain¹⁵. For clarity, we have removed data related to FDI-6 from the main figures and only retained some in **Supplementary Fig. 8e**. We also tested an additional FOXM1 inhibitor (RCM1)¹⁶, but this compound had negligible effects on FOXM1 function, and this dataset has now been removed.

Combined, we provide a highly coherent dataset supporting a role for FOXM1 in regeneration, in line with RNA-Seq analysis that predicted FOXM1 activation during regeneration (**Fig. 6a, b** and **Supplementary Fig. 8d**). We have therefore kept the thiostrepton dataset but have reduced length of this section and included the following statement in the revised manuscript (**lines 414-415**):

“However, genetic studies will be required to fully characterise the role of FOXM1 in germline regeneration.”

We note that our analysis of the regulation of FOXM1 expression by growth factor signalling in A_{undiff} is also very relevant for our manuscript although accompanying text has now been substantially reduced (**lines 417-440**) and data moved to **Supplementary Fig. 9**. This section links together the growth factor stimulation of A_{undiff} occurring during regeneration and changes in cellular behaviour. Our data indicates that FOXM1 expression is tightly controlled at both transcriptional and post-transcriptional levels in A_{undiff} and that coordinated activation of multiple signalling pathways, e.g., PI3K/AKT and mTORC1, is required for efficient induction (**Supplementary Fig. 9b, h**). Thus, observed activation of FOXM1 in regenerative A_{undiff} provides key insight into cellular pathways activated in response to germline damage, as well downstream effectors involved in regeneration.

REVIEWER #3

All the issues raised in the original review have been adequately addressed.

References

1. van Keulen CJ, de Rooij DG. Spermatogenic clones developing from repopulating stem cells surviving a high dose of an alkylating agent. *Cell Tissue Kinet* **8**, 543-551 (1975).
2. Butler A, Hoffman P, Smibert P, Papalexi E, Satija R. Integrating single-cell transcriptomic data across different conditions, technologies, and species. *Nat Biotechnol* **36**, 411-420 (2018).
3. Kanatsu-Shinohara M, Takashima S, Ishii K, Shinohara T. Dynamic changes in EPCAM expression during spermatogonial stem cell differentiation in the mouse testis. *PLoS One* **6**, e23663 (2011).
4. Hegde NS, Sanders DA, Rodriguez R, Balasubramanian S. The transcription factor FOXM1 is a cellular target of the natural product thiostrepton. *Nat Chem* **3**, 725-731 (2011).
5. Buchner M, *et al.* Identification of FOXM1 as a therapeutic target in B-cell lineage acute lymphoblastic leukaemia. *Nat Commun* **6**, 6471 (2015).
6. Kwok JM, Myatt SS, Marson CM, Coombes RC, Constantinidou D, Lam EW. Thiostrepton selectively targets breast cancer cells through inhibition of forkhead box M1 expression. *Mol Cancer Ther* **7**, 2022-2032 (2008).
7. Hasegawa T, *et al.* Identification of a novel arthritis-associated osteoclast precursor macrophage regulated by FoxM1. *Nat Immunol* **20**, 1631-1643 (2019).

8. Sinha S, *et al.* Glycogen synthase kinase-3 β inhibits tubular regeneration in acute kidney injury by a FoxM1-dependent mechanism. *FASEB J* **34**, 13597-13608 (2020).
9. Yang J, *et al.* Pathological Ace2-to-Ace enzyme switch in the stressed heart is transcriptionally controlled by the endothelial Brg1-FoxM1 complex. *Proc Natl Acad Sci U S A* **113**, E5628-5635 (2016).
10. Sheng Y, *et al.* FOXM1 regulates leukemia stem cell quiescence and survival in MLL-rearranged AML. *Nat Commun* **11**, 928 (2020).
11. Laoukili J, *et al.* FoxM1 is required for execution of the mitotic programme and chromosome stability. *Nat Cell Biol* **7**, 126-136 (2005).
12. Cui J, *et al.* HGF/Met and FOXM1 form a positive feedback loop and render pancreatic cancer cells resistance to Met inhibition and aggressive phenotypes. *Oncogene* **35**, 4708-4718 (2016).
13. Gormally MV, *et al.* Suppression of the FOXM1 transcriptional programme via novel small molecule inhibition. *Nat Commun* **5**, 5165 (2014).
14. Wang X, Quail E, Hung NJ, Tan Y, Ye H, Costa RH. Increased levels of forkhead box M1B transcription factor in transgenic mouse hepatocytes prevent age-related proliferation defects in regenerating liver. *Proc Natl Acad Sci U S A* **98**, 11468-11473 (2001).
15. Tabatabaei-Dakhili SA, Aguayo-Ortiz R, Dominguez L, Velazquez-Martinez CA. Untying the knot of transcription factor druggability: Molecular modeling study of FOXM1 inhibitors. *J Mol Graph Model* **80**, 197-210 (2018).
16. Sun L, *et al.* The FOXM1 inhibitor RCM-1 suppresses goblet cell metaplasia and prevents IL-13 and STAT6 signaling in allergen-exposed mice. *Sci Signal* **10**, (2017).

REVIEWERS' COMMENTS

Reviewer #2 (Remarks to the Author):

I have no additional comments or suggestions. I appreciate that the manuscript has been very much clarified and focused. My previous suggestion to shorten the mns might have been an irregular one (though I believe it was essential). However, I acknowledge that the authors made great efforts to rearrange it in a really effective manner. I hope that this will be an important piece of work regarding the regenerating properties of SSCs.

RESPONSE TO THE REVIEWERS

REVIEWER #2

I have no additional comments or suggestions. I appreciate that the manuscript has been very much clarified and focused. My previous suggestion to shorten the mns might have been an irregular one (though I believe it was essential). However, I acknowledge that the authors made great efforts to rearrange it in a really effective manner. I hope that this will be an important piece of work regarding the regenerating properties of SSCs.

We thank the Reviewer for the positive comments and previous suggestions for improvement.